# The exceptionally high diversity of small carnivorans from the Late Miocene hominid locality of Hammerschmiede (Bavaria, Germany)

Nikolaos Kargopoulos[1]*, Alberto Valenciano[2,3]*, Juan Abella[4,5], Panagiotis Kampouridis[1], Thomas Lechner[1,6], Madelaine Böhme[1,6]

1 Department of Geosciences, Eberhard Karls University of Tübingen, Tübingen, Baden-Württemberg, Germany, 2 Departamento de Ciencias de la Tierra, Universidad de Zaragoza and Instituto Universitario de Investigación en Ciencias Ambientales de Aragón (IUCA), Zaragoza, Zaragoza, Spain, 3 Research and Exhibitions Department, Iziko Museums of South Africa, Cape Town, Western Cape, South Africa, 4 Institut Català de Paleontologia Miquel Crusafont, Universitat Autònoma de Barcelona, Barcelona, Catalonia, Spain, 5 Instituto Nacional de Biodiversidad, Quito, Pichincha, Ecuador, 6 Senckenberg Centre for Human Evolution and Paleoenvironment (HEP), Tübingen, Baden-Württemberg, Germany

* nikoskargopoulos@gmail.com (NK); alb3rtovv@gmail.com (AV)

**Data Availability Statement:** All relevant data are within the manuscript.

## Abstract

The present study deals with new material of carnivorans (Mustelidae, Mephitidae, Ailuridae, Potamotheriinae and Viverridae) from the basal Tortonian (Late Miocene, late Astaracian) hominid-bearing locality of Hammerschmiede (Bavaria, Germany). The small carnivoran fauna includes 20 species belonging to nine different subfamilies (Guloninae, Lutrinae, Mellivorinae, Potamotheriinae, Leptarctinae, Mephitinae, Simocyoninae, Genettinae and Viverrinae). The identified forms include: "*Martes*" *sansaniensis*, "*Martes*" cf. *munki*, "*Martes*" sp., *Circamustela hartmanni* n. sp., *Laphyctis mustelinus*, Guloninae indet., *Eomellivora moralesi*, *Vishnuonyx neptuni*, *Paralutra jaegeri*, *Lartetictis* cf. *dubia*, *Trocharion albanense*, *Palaeomeles pachecoi*, *Proputorius sansaniensis*, *Proputorius pusillus*, *Alopecocyon goeriachensis*, Simocyoninae indet., *Potamotherium* sp., *Semigenetta sansaniensis*, *Semigenetta grandis* and *Viverrictis modica*. The new species *Circamustela hartmanni* n. sp. is differentiated from the other members of the genus by its small size and the morphology of its dental cusps in the upper and lower carnassials. This is one of the highest reported taxonomic diversities for fossil small carnivorans in the Miocene of Europe, including also first and last occurrences for several genera and species. Additionally, the assemblage comprises some rare taxa such as *Palaeomeles pachecoi* and *Eomellivora moralesi*. An ecomorphological comparison of the discovered taxa reveals possible cases of competition and niche partitioning.

## Introduction

The locality of Hammerschmiede, situated near the small town of Pforzen (southwest Bavaria, Germany), has been known for its Miocene fluvio-alluvial fossiliferous sediments for nearly

**Funding:** The research was supported by the Bavarian State Ministry of Research and the Arts, and by the Bavarian Natural History Collections (SNSB). The first author (N. K.) would like to thank DAAD for financially supporting the present project. This study was also supported by the Government of Aragon (Group ref. E33_20R), the Spanish Ministry of Economy and Competitiveness and FEDER funds (Research Projects PGC2018-094122-B-100 and PID2020-116220GB-I00), the Research Group UCM 910607, and the Spanish Ministry of Science, Innovation, and Universities ("Juan de la Cierva Formación", ref. FJC2018-036669-I for A.V.). This study was supported by the Generalitat de Catalunya (CERCA Programme, and Beatriu de Pinós contract 2017 BP 00223 from AGAUR to J. A.). The funders had no role in study design, data collection and analysis, decision to publish, or preparation of the manuscript.

**Competing interests:** The authors have declared that no competing interests exist.

half a century. At least six different fossiliferous levels have been found in the clay pit, with the majority of fossils being found at the fluvial channels HAM 4 and HAM 5. These two channel fillings have been dated to 11.44 and 11.62 Mya respectively [1]. During these fifty years of studies, several publications have been conducted, revealing an extraordinary faunal assemblage of mammals, birds, reptiles, amphibians, fish, molluscs, and plants [1–23]. The species *Danuvius guggenmosi* Böhme et al., 2019 [16], a great ape that is suggested to have practiced bipedalism in its locomotion, has brought the locality in the spotlight [16, 24, 25].

The datum of the carnivorans of the locality was firstly investigated by [4], who reported the presence of *Proputorius sansaniensis* Filhol, 1890 [26] and *Proputorius pusillus* (Viret 1951) [27] (as "*Martes pusillus*") in the HAM 1 layer. Later, [1] and [16] published a preliminary faunal list for the locality, reporting carnivorans of several families. The present article is a part of a detailed review of the carnivoran fauna of the locality that has started recently with the publications of [21–23], reporting the discovery of the viverrids *Semigenetta sansaniensis* [28] and *Semigenetta grandis* Crusafont Pairó & Golpe Posse, 1981 [29]; the new otter species *Vishnuonyx neptuni* Kargopoulos et al., 2021 [22]; the ictithere *Thalassictis montadai* Villalta Comella & Crusafont Pairó, 1943 [30] and a large hyaenid.

The aim of this article is to present new material of the groups Mustelidae Batsch, 1788 [31], Mephitidae Bonaparte, 1845 [32], Ailuridae Gray, 1843 [33], Potamotheriinae Willemsen, 1992 [34] and Viverridae Gray, 1821 [35] from the locality of Hammerschmiede. The specimens belonging to the genus *Circamustela* Petter, 1967 [36] are here attributed to a new species. An ecomorphological comparison between the discussed forms is conducted, in order to reveal the possible intraspecific interactions, such as competition or niche partitioning.

## Material and methods

### Material

The specimens studied herein come from the fluvial channels HAM 1, HAM 4 (11.44 Ma), and HAM 5 (11.62 Ma) of the locality of Hammerschmiede (Bavaria, Germany). The material from HAM 1 corresponds to the material published by [4]. This material has been reviewed and some specimens were attributed to different taxa. The incisors published by [4] were not included in the present manuscript, because determination on species level was not possible, due to the lack of diagnostic characters. The exact age of HAM 1 is not known, but based on the details given by [4] [the sediment description as greenish-grey marl with aquatic gastropods, the given thickness of the horizon (50 cm) and the topographic height (ca. 680 m a.s.l.)] a lateral correlation to HAM 5 can be assumed. The material from HAM 4 and HAM 5 has been unearthed during the excavations held by the Eberhard-Karls University of Tübingen between 2011 and 2021. All the material is currently stored in the Palaeontological Collection of the University of Tübingen, Germany (GPIT), and is inventoried with numbers of both GPIT (for excavations from 2011 to 2019) and SNSB-BSPG (for excavations of 2020 and 2021). No permits were required for the described study, which complied with all relevant regulations. More information about the geographic position and the stratigraphy of the locality can be found in [1, 16].

### Methods

The term "small carnivorans" is used in the sense given by [37]. Dental nomenclature follows [38] and [39]. All measurements were taken with a digital caliper and rounded to the first decimal point. In cases of multiple skeletal specimens per element for a single species, the descriptions and comparisons concern all the available specimens. In cases of multiple data for a form in the tables, the range, average value and number of specimens are mentioned. Single

measurements in square brackets indicate that they have been taken at the alveolus, whereas single measurements in a parenthesis indicate that they were taken in approximation due to specimen damage.

Institutional Abbreviations: **GPIT:** Palaeontological collection of the University of Tübingen, Tübingen, Germany; **ICP:** Institut Català de Paleontologia, Barcelona, Spain; **MHNL:** Muséum d'histoire naturelle de Lyon, Lyon, France; **MNHN:** Muséum national d'Histoire naturelle, Paris, France; **NHMUK:** Natural History Museum, London, United Kingdom; **NMA:** Naturmuseum der Stadt Augsburg, Augsburg, Germany; **NHMW:** Naturhistorisches Museum Wien, Vienna, Austria; **SMNS:** Staatliches Museum für Naturkunde Stuttgart, Stuttgart, Germany; **SNSB-BSPG:** Staatliche Naturwissenschaftliche Sammlungen Bayerns-Bayerische Staatssammlung für Paläontologie und Geologie, Munich, Germany; **UCBL:** Université Claude-Bernard Lyon I, Villeurbanne, France.

### Nomenclatural acts

The electronic edition of this article conforms to the requirements of the amended International Code of Zoological Nomenclature, and hence the new names contained herein are available under that Code from the electronic edition of this article. This published work and the nomenclatural acts it contains have been registered in ZooBank, the online registration system for the ICZN. The ZooBank LSIDs (Life Science Identifiers) can be resolved and the associated information viewed through any standard web browser by appending the LSID to the prefix "http://zoobank.org/". The LSID for this publication is: urn:lsid:zoobank.org:pub:B09DB8CD-3CA3-48F9-ACAA-B6AB27001A44. The electronic edition of this work was published in a journal with an ISSN, and has been archived and is available from the following digital repositories: PubMed Central and LOCKSS.

### Results

Order Carnivora Bowdich, 1821 [40]

 Suborder Caniformia Kretzoi, 1943 [41]

 Family Mustelidae Batsch, 1788 [31]

 Subfamily Guloninae Gray, 1825 [42]

 Genus *Martes* Pinel, 1792 [43]

 **Type species:** *Martes foina* (Linnaeus 1758) [44]

 **Remarks:** The genus *Martes* in its traditional sense includes several dozens of extant and fossil species (e.g., [38]). Recently, [45] demostrated that small marten-like mustelids from the Early and Middle Miocene of Eurasia show a dissimilar morphology with *Martes* and adscribed them to "*Martes*". These mustelids are: "*Martes*" *laevidens* Dehm, 1950 [46], "*Martes*" *sainjoni* (Mayet, 1908) [47], "*Martes*" *munki* Roger, 1900 [48], "*Martes*" *delphinensis* Depéret, 1892 [49], "*Martes*" *burdigaliensis* de Beaumont, 1974 [50], "*Martes*" *collongensis* Roth and Mein, 1987 [51], "*Martes*" *cadeoti* Mein, 1958 [52], "*Martes*" *sansaniensis* (Lartet, 1851) [28], "*Martes*" *filholi* (Depéret, 1887) [53], *Aragonictis araid* Valenciano et al., 2022 [45], "*Martes*" *woodwardi* Pilgrim, 1931 [54], "*Martes*" *jaegeri* (Schlosser, 1902) [55], "*Martes*" *lefkonensis* Schmidt-Kittler, 1995 [56], "*Martes*" *anderssoni* Schlosser, 1924 [57], "*Martes*" *melibulla* Petter, 1963 [58], "*Martes*" *basilii* Petter, 1964 [59], "and "*Martes*" *leporinum* (Khomenko, 1914) [60]. These forms are in need of a thorough taxonomical revision, which is beyond the scope of the present article. For practical issues we refer to all these Miocene forms as "*Martes*", following [61], although [45, 62] suggested that some Late Miocene forms can be classified as *Martes*. The species *Mustela transitoria* Gaillard, 1899 [63] has been considered as related to the aforementioned forms, but it has now been transferred to its own genus: *Gaillardina*

Ginsburg, 1999 [38]. The two most commonly discussed extant species of the genus are *Martes martes* (Linnaeus, 1758) [44] and *Martes foina* (Erxleben, 1777) [64].

"*Martes*" *sansaniensis* (Lartet, 1851) [28]

**Lectotype:** MNHN Sa 755, a left hemimandible with the roots of p3 and the p4 & m1.

**Type Locality:** Sansan (France).

**Referred Specimens:** HAM 4: GPIT/MA/10959, skull; SNSB-BSPG-2020 XCIV-4065, right P4; GPIT/MA/16963, right P4. HAM 5: GPIT/MA/16349, left P4; GPIT/MA/09882, left M1; GPIT/MA/12308, right M1.

**Description:** The specimen GPIT/MA/10959 (Figs 1A and 2A) is an almost complete skull, lacking only parts of the zygomatic arches and being dorsoventrally compressed at the palate and slightly damaged at the right auditory region and the postorbital processes. The skull is relatively long and narrow. The external narial aperture is deformed, but it seems to be high, wide and M-shaped. The anterior palatine foramina are short, extending posteriorly at the middle of the canines' plane. The mesial border of the orbit ends at the plane of the connection between P3 and P4. The infraorbital foramen is relatively large. The palate extends far beyond the plane of M1, ending to a relatively narrow choana. The postorbital processes are moderately developed and a strong postorbital constriction is present. Two faint temporal lines start at the postorbital processes and merge approximately at the level of postorbital constriction. A faint sagittal crest starts to develop at this point, ending at the nuchal crest, which is also not very robust. Most probably, the restricted size of these crests and the absence of teeth wear indicate that the skull belongs to a young adult individual. The braincase is relatively low and wide. The remaining parts of the zygomatic arch are relatively thin and the glenoid cavity is

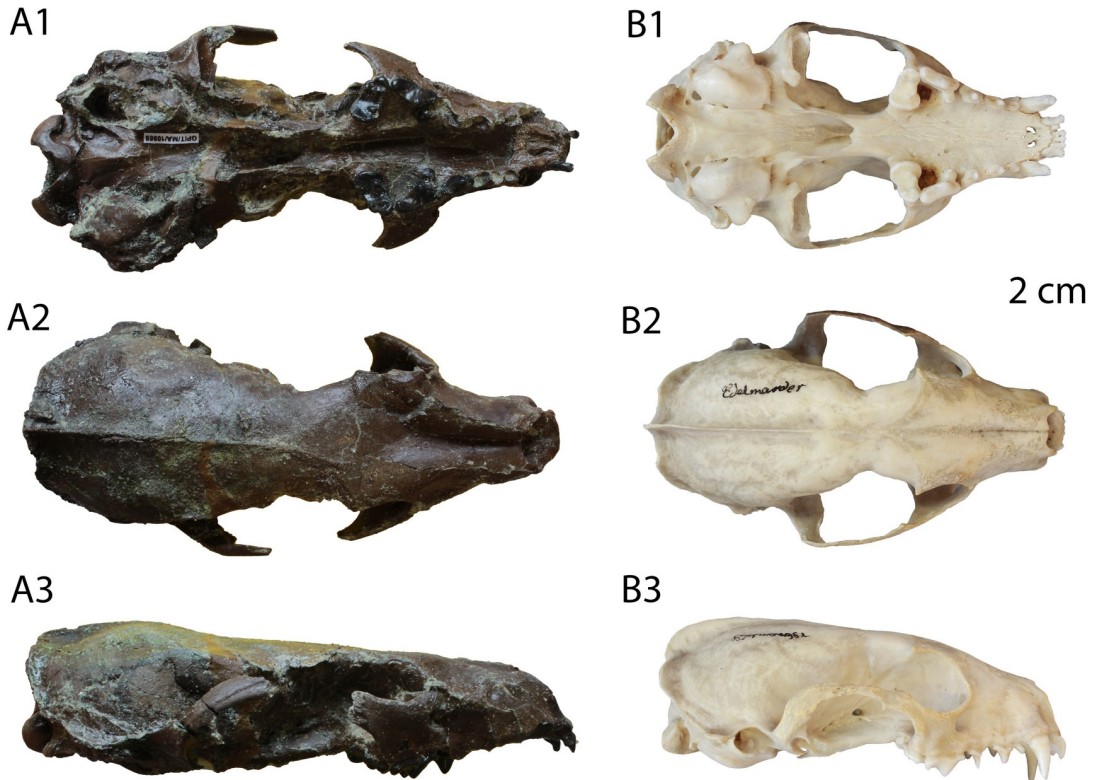

**Fig 1.** Skull of "*Martes*" *sansaniensis* from Hammerschmiede (A; GPIT/MA/10959) in comparison to that of the extant *Martes martes* (B; GPIT/MA/18609).

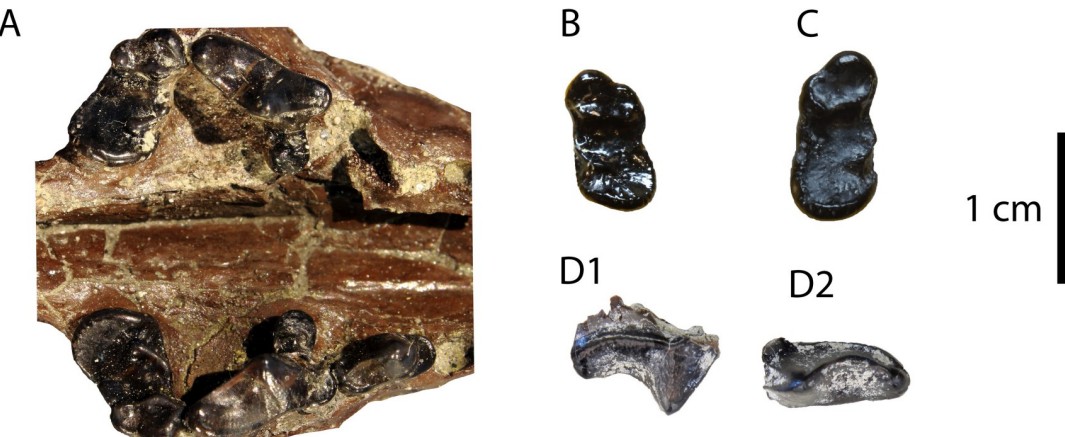

**Fig 2.** Upper dentition of "*Martes*" *sansaniensis* from Hammerschmiede: (A) GPIT/MA/10959 skull; (B) GPIT/MA/09882 left M1; (C) GPIT/MA/12308 right M1 (D) SNSB-BSPG-2020 XCIV-4065 right P4 in lingual (D1) and occlusal view (D2).

anteriorly-oriented having a strong postglenoid process. The auditory region is damaged, not enabling a detailed description. However, the entotympanic is relatively large and elongated, chaperoned by a narrow and mesiolaterally placed ectotympanic. The mastoid process is small, while the paraoccipital process is developed. A hypoglossal foramen is present near the foramen magnum. The posterior lacerate and jugular foramina seem to be fused in one opening. The foramen magnum is oval in shape and the occipital condyles are laterally bent. The occiput is moderately high and U-shaped.

The first and second incisors seem to have approximately the same size, judging from their alveoli, while I3 is significantly larger. The preserved I2 and I3 have similar morphology, being peg-shaped and having a marked cingulum in their distal border. This cingulum covers only the distolingual part of I3, which also has a slight distal heel-like enhancement. Their measurements can be found in Table 1. The alveoli of C are large and oval-shaped. The alveolus of P1 is relatively large and in close contact to that of P2. The alveolus of P2 is two-rooted, large, in-line and in close contact with P3 and its distal root is significantly longer than the mesial one. The P3 is formed by a high main cusp (P3H = 4.1 mm) with no accessory cusps and a smooth cingulum in its perimeter. It is slightly asymmetrical with the main cusp being faintly distally bent and the distal part of the base of the tooth creating a small heel valley. The upper carnassial has a cingulum, which is more developed at its lingual-distal part. There are no signs of wear. The protocone is large, with a long and slightly compressed neck, situated between the planes of the paracone and the minute parastyle. The paracone is the largest cusp (P4H = 5.0 mm) and it hosts two fine crests, towards the low and robust parastyle and the mesial border of protocone's neck respectively. The metastyle is low, blunt and buccally bent. There is no carnassial notch between the paracone and the metastyle. The upper molar is large and oval-shaped, without signs of wear. The cingulum is faint. The buccal border over the paracone is distinctly more developed than over the metacone. The paracone and the metacone are of

**Table 1. Dimensions (in mm) of the upper incisors of the "*Martes*" *sansaniensis* skull from Hammerschmiede (GPIT/MA/10959).**

|    | L   | W   | H   |
|----|-----|-----|-----|
| I2 | 3.0 | 1.8 | 4.2 |
| I3 | 3.9 | 2.8 | 5.2 |

similar height and they are connected by the postparacrista and the premetacrista. The metacone is slightly sharper than the paracone. A preparacrista is also present, but there are no signs of a parastyle or a metastyle. A preparaconular crista connects the mesial cingulum with the low and blunt paraconule. There are no signs of other cusps or cristae. A similar morphology is exhibited at the two isolated M1.

**Comparison**: Size has been one of the main differentiating factors between the Miocene marten-like species. Tables 2 and 3 summarize the metrical comparison between all the forms in concern, based on upper and lower teeth respectively, demonstrating that there is a notable size difference between some groups.

Three main size groups can be seen when observing Tables 2 and 3 in comparison to the extant *Martes martes* and *Martes foina* (m1L≈8.5–10.5 mm; P4L≈7.5–9.0 mm; M1L≈3.5–4.5 mm). The small-sized forms include: "*M*". *delphinensis*, "*M*". *cadeoti*, "*M*". *laevidens*, "*M*". *jaegeri* and "*M*". *lefkonensis* (m1L≈5.0–7.5 mm; P4L≈6.5–7.5 mm; M1L≈3.0–4.0 mm). The species "*M*". *anderssoni*, "*M*". *collongensis*, "*M*". *burdigaliensis*, "*M*". *munki*, "*M*". *melibulla*, "*M*". *basilii* and "*M*". *sainjoni* occupy and intermediate position (m1L≈8.0–11.0 mm; P4L≈9.5–10.5 mm; M1L≈4.0–5.0 mm), whereas the species "*M*". *filholi*, "*M*". *sansaniensis*, "*M*". *woodwardi* and "*M*". *leporinum* are relatively large (m1L≈11.0–14.0 mm; P4L≈10.5–11.5 mm; M1L≈4.5–6.5 mm). The material from Rudabánya published by [65] as "*M*. cf. *filholi*" represents a smaller form with morphological differences from the type material of "*M*". *filholi* (e.g. the lower M1L/M1W ratio). These specimens most probably cannot be attributed to "*M*". *filholi*, but in the present paper the name "*M*. cf. *filholi*" is retained until more material can clarify its taxonomy.

The specimens GPIT/MA/10959, GPIT/MA/12308, GPIT/MA/09882 from Hammerschmiede belong to the large-sized species group, whereas the specimens GPIT/MA/10666, GPIT/MA/10636, GPIT/MA/18606 and GPIT/MA/16924 are clearly smaller, fitting between the small- and medium-sized forms. The latter material is discussed in detail further below.

**Table 2. Comparison of the upper teeth of the "*Martes*" material from Hammerschmiede and the marten-like species from the Middle/Late Miocene indicating the source of data.** In the "*M*". *filholi* specimen from La Grive-Saint-Alban, measurements were taken from a cast of the holotype and based on the figures of [27].

| Species | Code/Locality | P3L | P3W | P4L | P4W | M1L | M1W |
|---|---|---|---|---|---|---|---|
| "*M*". *sansaniensis* | GPIT/MA/10959 | 7.0 | 3.4 | 10.2 | 7.3 | 5.0 | 10.2 |
| | GPIT/MA/16349 | | | 9.8 | | | |
| | GPIT/MA/16963 | | | 9.7 | | | |
| | GPIT/MA/12308 | | | | | 5.1 | 10.6 |
| | GPIT/MA/09882 | | | | | 5.3 | 9.4 |
| "*M*". cf. *munki* | GPIT/MA/10666 | | | | | 4.5 | 7.3 |
| | GPIT/MA/18606 | | | | | 4.4 | 7.4 |
| "*Martes*" sp. | SNSB-BSPG-1973-XIX-34 | | | 4.1 | 2.2 | | |
| "*M*". *sansaniensis* | Sansan [72] | 6.7 | 3.2 | 11.2 | 8.5 | 4.7–5.4 4.9 (5) | 9.3–11.0 10.0 (5) |
| "*M*". *filholi* | La Grive [53], Vieux Collonges [52] | 6.8 | 3.8 | 10.4–11.0 10.8 (3) | 7.4–8.0 7.7 (3) | 6.6–7.7 7.2 (2) | 9.7–11.0 10.4 (2) |
| "*M*. cf. *filholi*" | Rudabánya [65] | | | | | 4.4 | 8.0 |
| "*M*". *munki* | Sandelzhausen [73], Sant Quirico [58], Vieux-Collonges [52] | 5.0–7.4 6.2 (2) | 2.1–3.7 2.9 (2) | 9.6–10.0 9.8 (5) | 5.0–6.2 5.8 (5) | 3.9–5.0 4.4 (10) | 7.8–10.2 9.2 (10) |
| "*M*". aff. *anderssoni* | Can Ponsic [36] | | | | | 4.0 | 8.0 |
| "*M*". *cadeoti* | Vieux-Collonges [52] | | | 7.1–7.5 7.3 (3) | 3.9–4.3 4.0 (3) | 3.0–3.8 3.4 (4) | 6.6–7.2 6.9 (4) |
| "*M*". *lefkonensis* | Maramena [56] | | | 6.3 | | 3.0 | |

**Table 3. Comparison of the lower teeth of the "*Martes*" material from Hammerschmiede and the members of the genus "*Martes*" in the Middle/Late Miocene indicating the source of data.** The measurements of the material of "*M*". *filholi* from La Grive-Saint-Alban were taken based on the figures of [27].

| Species | Code/Locality | p2L | p2W | p3L | p3W | p4L | p4W | m1L | m1W | m2L | m2W |
|---|---|---|---|---|---|---|---|---|---|---|---|
| "*M*". cf. *munki* | GPIT/MA/10636 | | | | | | | 8.3 | 3.2 | | |
| | GPIT/MA/16924 | | | | | 4.6 | 2.2 | 7.6 | 3.1 | 3.3 | 2.7 |
| "*M*". *sansaniensis* | Sansan [72] | 4.4–5.9 5.4 (8) | 2.5–3.3 2.9 (8) | 5.5–7.0 6.6 (10) | 2.7–3.4 3.2 (8) | 8.0–9.0 8.4 (10) | 3.6–4.2 3.9 (9) | 11.8–13.8 12.9 (10) | 4.8–5.9 5.3 (11) | 4.3–5.6 5.1 (3) | 4.1–4.5 4.3 (3) |
| "*M*". *woodwardi* | Pikermi [66] | | | | | | | 11.4–12.0 11.7 (2) | 4.6–5.0 4.8 (2) | | |
| "*M*". *leporinum* | Taraklia [77] | 5.0 | | 6.5 | | 8.8 | | 13.5 | | | |
| "*M*". *filholi* | La Grive & Vieux-Collonges [52] | 6.0 | 3.0 | 7.0 | 3.3 | 6.0–8.0 7.0 (2) | 3.0–4.0 3.5 (2) | 10.0–11.5 10.8 (2) | 4.4–5.3 4.9 (2) | 4.6 (2) | 3.8–3.9 3.9 (2) |
| "*M. cf. filholi*" | Rudabánya [65] | | | | | 5.6 | 3.0 | 9.5–10.3 9.9 (2) | 3.4–3.9 3.6 (3) | | |
| "*M*". *munki* | Sandelzhausen [73], Erketshofen 2 [76] & Vieux-Collonges [52] | 3.8–5.1 4.2 (5) | 2.0–2.5 2.2 (5) | 4.5–5.7 5.1 (9) | 2.2–2.39 2.4 (9) | 5.2–8.4 6.3 (11) | 2.6–3.8 3.0 (11) | 8.3–9.5 9.0 (10) | 3.8–4.5 4.1 (10) | 3.3 (2) | 3.1–3.2 3.2 (2) |
| "*M*". *basilii* | Los Algezares [59] | 4.1 | 2.7 | 5.8 | 3.0 | | | 11.0 | 4.0 | | |
| "*M*". cf. *basilii* | Can Ponsic [78] | | | | | | | 9.8 | 4.5 | | |
| "*M*". *sainjoni* | Chilleurs-aux-Bois [47] & Artenay [47] | | | 5.5 | | 7.0 (3) | | 11.0 | 4.5 | | |
| "*M*". *melibulla* | Can Llobateres [58] | | | | | 7.0 | 3.2 | 10.5 | 4.5 | | |
| "*M*". *anderssoni* | Can Ponsic [36] | | | | | 5.5 | | 9.0 | 3.4 | | |
| "*M*". *collongensis* | Vieux-Collonges [51] | | | 4.4 | 2.2 | 5.3 | 2.5 | 8.3 | 3.5 | 2.2 | 1.6 |
| "*M*". *burdigaliensis* | Vieux-Collonges [50] | | | | | | | 7.4–8.3 (4) | 3.3 | | |
| "*M*". *cadeoti* | Vieux-Collonges [52] | | | 3.3 (2) | 1.9–2.2 2.1 (2) | 4.9–5.4 5.2 (2) | 2.2–2.9 2.6 (2) | 6.8–7.3 7.0 (4) | 3.0–3.2 3.1 (4) | | |
| "*M*". *jaegeri* | Salmendingen [55] | | | [2.5] | | [3.0] | | 5.5 | | | |
| "*M*". *laevidens* | Wintershof-West [46] | 3.3–3.7 3.5 (2) | | 4.5–4.7 4.6 (2) | | 4.8–5.8 5.3 (4) | | 6.8–7.5 7.1 (5) | | 3.4 | |
| "*M*". *delphinensis* | Hostalets de Pierola [30], Vieux-Collonges [52], Manchones [78] & La Grive [78] | | | 3.5 | | 3.7–4.5 4.1 (2) | 1.7–1.9 1.8 (2) | 5.3–6.7 5.9 (10) | 2.1–2.6 2.4 (7) | | |
| "*M*". *lefkonensis* | Maramena [56] | | | | | 4.9 | 2.4 | 7.2–7.6 7.4 (4) | 2.8–3.2 3.0 (4) | | |

The former three specimens fit in the size group of "*M*". *filholi*, "*M*". *sansaniensis*, "*M*". *woodwardi* and "*M*". *leporinum*. The only species from the medium-sized group that is comparable to the larger specimens from Hammerschmiede is "*M*". *munki*, which is differentiated by its significantly narrower P4 (Table 2). The species "*M*". *woodwardi* and "*M*". *leporinum* are known only from lower teeth, so a direct comparison is impossible. However, a considerable stratigraphic difference must be taken into consideration as "*M*". *woodwardi* has been reported only from Pikermi [54, 66] and "*M*". *leporinum* only from Taraklia [60]. Both localities have been characterized as typical Turolian (MN 12) faunas [38]. Significant faunal turnovers have taken place during the Aragonian-Vallesian and Vallesian-Turolian transitions [67–71] making the unaltered survival of a marten species (known for being rather speciose) seem highly

improbable. Therefore, the comparison is mainly focused on the species "*M*". *munki*, "*M*". *filholi* and "*M*". *sansaniensis.*

The species "*M*". *munki* is rather similar to "*M*". *sansaniensis*, but it exhibits considerably narrower upper carnassials (Table 2). This difference can be seen both metrically and in relation to M1L. Additionally, [52] mentions the presence of a protoconule and a metaconule in the M1, which are not present in the herein described specimens. The two remaining forms are similar in several characteristics of their dentition. However, the size of M1, both metrically and in relation to P4L, is far larger in "*M*". *filholi* (Table 2). Additionally, this tooth in "*M*". *filholi* is relatively long, having a M1L/M1W ratio of 68%, whereas in "*M*". *sansaniensis* this ratio ranges between 42% and 52%. The specimens published by [65] from Rudabánya as "*M*". cf. *filholi* represent a smaller form with a M1L/M1W ratio closer to that of "*M*". *sansaniensis* (55%). The Hammerschmiede molars are metrically smaller than the "*M*". *filholi* specimens from Vieux-Collonges and La Grive, being more similar to "*M*". *sansaniensis* (Table 2). Additionally, the M1L/M1W ratio ranges between 48% and 56% indicating closer affinities with the latter species. In terms of morphology, the two species are very similar [49, 53, 72]. [72] stated that the most considerable morphological difference between these two forms is that the M1 metaconule (when present) is not connected to the protocone in "*M*". *sansaniensis*, whereas these two cusps are always connected in "*M*". *filholi*. No sign of a metaconule is present in the described specimens, resembling more the morphology of "*M*". *sansaniensis.*

Therefore, the larger "*Martes*" specimens from Hammerschmiede are here considered to be closer to the species "*M*". *sansaniensis*. Some small differences can be traced with regard to the Sansan material, as the slightly more developed P4 parastyle and the mesiodistally shorter lingual platform of M1 in the Hammerschmiede specimens. However, we don't consider these differences to be of any taxonomic value and they can be interpreted as intraspecific variability.

The specimen GPIT/MA/10959 from Hammerschmiede is the first known skull of "*M*". *sansaniensis* and the youngest record of this form. Therefore, a comparison to that of the extant *M. martes* is here reported. [45] enumerated several dental traits in common of "*M*". *sansaniensis* from Sansan with extant *Martes*. Among these are the presence of the p4 distal accessory cuspid, and the overall similarities in the carnassials and the M1 (large P4 protocone, a relatively elongated and basined m1 talonid with a conical hypoconid linking the metaconid by an entocristid; a non-reduced M1 lingual platform, and the posession of a narrow M1 crown at about mid-width). However, they also found differences, as the presence of a diastema between p2–3, a higher m1 protoconid, and M1 with distinct proportions (more elongated buccolingually) and morphology (larger M1 parastyle, stronger development of the metacone, metaconule more developed [when present], and a protocone located more buccally) compared to *Martes*. To these notes, we have observed the following differences. The skull and the dentition of the fossil species are larger than those of the extant marten. However, there is an inconsistency to the difference percentage between the two species (Table 4). The skull length, condylobasal length and palate length differences indicate that the skull of the extant species is approximately 82% as long as the fossil. However, other measurements that concern the width of the skull such as the rostrum width at the canines, the choana width, the braincase width and the mastoid width suggest that the two species are relatively more similar (94%). Some extreme differences (e.g. the palate width at P4 and the height of the foramen magnum) can be attributed to the diagenetic deformation of the fossil specimen. Additional cranial differences between the species are: the longer paraoccipital processes, the less inflated entotympanic (also possibly related to deformation), the stronger zygomatic arch and the longer anterior palatine foramina in "*M*". *sansaniensis*. Other cranial differences (such as the lesser development of the sagittal, nuchal and postorbital crests and the width of the zygomatic

**Table 4. Comparison of the skull measurements of "*Martes*" *sansaniensis* from Hammerschmiede (GPIT/MA/10959) with that of *Martes martes* (GPIT/MA/18609).**

| Measurement | "*Martes*" *sansaniensis* (GPIT/MA/10959) | *Martes martes* (GPIT/MA/18609) | Difference % |
|---|---|---|---|
| Skull Length | 108.4 | 88.7 | 82% |
| Condylobasal Length | 101.4 | 81.9 | 81% |
| Rostrum Width at Canines | 17.6 | 18.0 | 102% |
| Palate Length | 54.4 | 45.4 | 83% |
| Palate Width at P4 | (26.4) | 28.3 | 107% |
| Choana Width | 8.5 | 8.2 | 96% |
| Preorbital Constriction | (25.3) | 21.3 | 84% |
| Postorbital Constriction | (20.8) | 16.6 | 80% |
| Maximum Braincase Width | 40.2 | 34.7 | 86% |
| Maximum Braincase Height | (26.3) | 26.2 | 100% |
| Foramen Magnum Width | 11.1 | 12.0 | 108% |
| Foramen Magnum Height | (6.6) | 9.1 | 138% |
| Mastoid Width | 43.7 | 40.3 | 92% |

arches) can be attributed to the relatively young age of the fossil individual, which is also supported by the absence of dental wear. The fossil species also exhibits some dental differences in comparison to the extant one: a more evident size difference between I2 and I3, the equal length of P2 and P3, the absence of the lingual expansion of P3, the less acute P4 protocone and paracone, the more developed cingulum in P4 and M1, the restriction of P4 parastyle to a faint elevation of the mesial cingulum, the higher M1 cusps, the slightly more homogenous connection between the M1 protocone and paracone, the more semi-circular buccal part of the tooth and the less developed lingual platform. The new skull of "*M*". *sansaniensis* from Hammerschmiede provides significant new evidence on the dentognathic morphology of this species, supporting the hypothesis of [45] of a generic split from the extant martens. A future detailed review of the heterogeneous group of "*Martes*" from the Miocene is expected to solidify this separation.

"*Martes*" cf. *munki* Roger, 1900 [48]

**Holotype:** NMA 80–39 a.S., right hemimandible with p3–m1.

**Type Locality:** Häder (Germany).

**Referred Specimens:** HAM 4: GPIT/MA/16924, right hemimandible with p4–m2. HAM 5: GPIT/MA/10666, right M1; GPIT/MA/18606, right M1; GPIT/MA/10636, left m1.

**Description**: The upper molar (GPIT/MA/10666; Fig 3B) exhibits a plesiomorphic mustelid morphology. There are no signs of wear and a moderately developed cingulum surrounds the tooth. The paracone is slightly larger than the metacone and there is not a high crista connecting them. The buccal side of the tooth is more enhanced over the paracone than over the metacone. A relatively high paraconule is present followed by both a preparaconular crista and a preprotocrista.

The hemimandible (GPIT/MA/16924; Fig 3A) is broken just mesially to p4 and at the centre of the masseteric fossa. It is relatively slender and no mental foramina are preserved. The masseteric fossa is deep and reaches the plane of the mesial part of m2. The fourth premolar is relatively high, with a developed cingulid and a large distal accessory cuspid. The lower carnassial is low, with developed wear facets and a smooth cingulid. The protoconid is the largest cuspid, connected with the paraconid through an obtuse angle. The metaconid is large and high, well separated from the protoconid and lingually inclined. The talonid is strongly worn, so no cuspids are exhibited. The talonid valley is moderately wide and shallow with a sagittal cristid

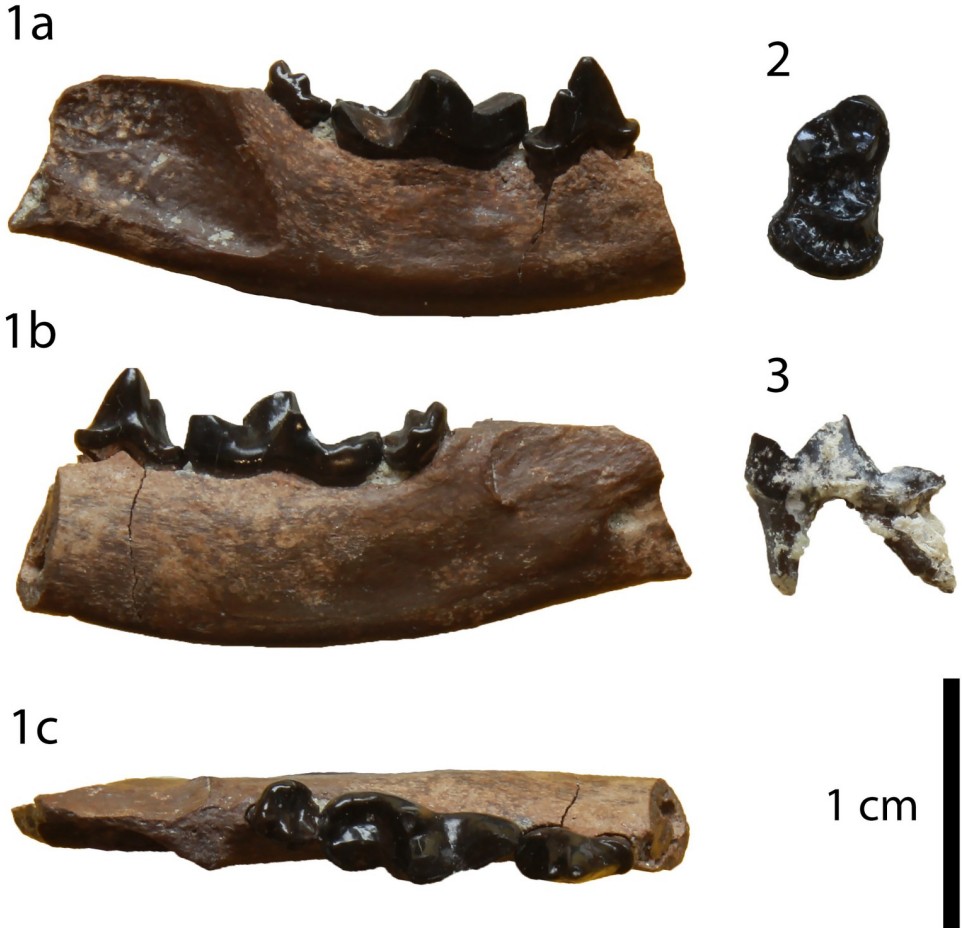

**Fig 3.** Material of "*Martes*" cf. *munki* from Hammerschmiede: (A) hemimandible (GPIT/MA/16924) in (1) buccal, (2) lingual and (3) occlusal views; (B) M1 (GPIT/MA/10666) in occlusal view; (C) m1 (GPIT/MA/10636) in buccal view.

reaching the distal cingulid. The second molar is unworn, one-rooted and with a faint cingulid. Its buccal side hosts three similar-sized cuspids, the hypoconid, the protoconid and the paraconid, while its lingual side hosts only the metaconid. All the cuspids are situated in a perimeter ridge, and there are no other cristids connecting them.

**Comparison**: As mentioned before, these specimens' size can be better corresponded to that of the smallest members of the medium-sized group (Tables 2 and 3). In particular, this material can be attributed to a form that is smaller than most specimens of "*M*". *munki*, "*M*". *basilii*, "*M*". *sainjoni* and "*M*". *melibulla* and larger than "*M*". *cadeoti*, "*M*". *jaegeri*, "*M*". *lefkonensis*, "*M*". *laevidens* and "*M*". *delphinensis*. Therefore, only the species "*M*". *anderssoni*, "*M*". *collongensis* and "*M*". *burdigaliensis* are fitting this size-group.

The two latter species are known from very scarce material from the locality of Vieux-Collonges [50, 51]. The species "*M*". *burdigaliensis* (despite its relatively small size) is a considerably robust form, with wide trigonid and talonid and thick enamel and high hypoconid, being significantly different from the herein described specimens. The species "*M*". *collongensis* has an enlarged talonid, a relatively high metaconid (similar to that of "*M*". *cadeoti*) and developed hypoconid, hypoconulid and cuspid-like distal crest of the protoconid. Therefore, it is also considered to be significantly different from the present material.

The species "*M*". *anderssoni* was created by [57] based on dental and postcranial material from the latest Miocene/Pliocene of China. [36] published some dental remains (isolated p4, a broken m1 and a complete M1) from Can Ponsic (Late Miocene, MN 9) as "*M*". aff. *anderssoni*. The herein presented specimens are morphologically very close to the material from Can Ponsic. [36] described the specimens from this locality stating that they were resembling both "*M*". *munki* and "*M*". *anderssoni*. However, the author did not specify why the material was suggested to be closer to "*M*". *anderssoni*. Both of these forms exhibit some taxonomic problems concerning their holotype material. The mandibles of the original material of "*M*". *anderssoni* are lost (B. KEAR, pers. comm.). Therefore, the only comparable material with the herein described specimens is the M1 that [57] tentatively attributed to this species. This tooth differs from GPIT/MA/10666 in the following traits: the narrower outline, the larger metacone, the much shorter postprotocrista that ends at the plane of the paracone and the less expanded lingual platform. These differences are also evident between the material from Can Ponsic and the material from China. On the other hand, the holotype of "*M*". *munki* is a fragmentary and deformed mandible from Häder published by [48]. Therefore, this species is also not well-defined. However, this name has been used broadly (possibly as a wastebasket nomen) to include medium- to small-sized martens from all over Europe during the Middle Miocene [27, 38, 48, 52, 74–76]. Therefore, we are inclined to tentatively attribute the Hammerschmiede material to this species as "*Martes*" cf. *munki*, pointing out that a thorough revision of its taxonomic status is needed. A depiction of the dimensions of some of the discussed forms can be seen in Fig 4).

"*Martes*" sp.

**Referred Specimens:** HAM 1: SNSB-BSPG-1973-XIX-34, left P4.

**Description**: This specimen is identical to the upper carnassials described above, but it is considerably smaller (Fig 5).

**Comparison**: [4] figured two tiny carnassials of a small mustelid from Hammerschmiede as *Martes pusillus* (SNSB-BSPG-1973-XIX-34 left P4; and SNSB-BSPG-1973-XIX-32 right m1). However, the P4 does not have the typical P4 morphology of *Proputorius*, which is the currently accepted genus for this species (see below). [52] figured a P4 of *P. pusillus* from Vieux-Collonges (MN5, France). It has a mesially placed protocone, not separated from the parastyle area and with a reduced neck, a morphology typical for *Proputorius* spp. On the contrary, SNSB-BSPG-1973-XIX-34 has a very individualized protocone mesiolingually projected and buccodistally slender. In this sense, we confidently doubt the classification of this P4 as belonging to this taxon. Its morphology resembles other Early–Middle Miocene marten-like forms such as *Circamustela*? *laevidens* (Dehm, 1950) [46] from Wintershoft-West, *Circamustela hartmanni* n. sp. from Hammerschmiede (this paper) or *Aragonictis araid* from Escobosa [45]. Tough, it has a noticeable smaller size than the previously three mentioned taxa, even smaller than "*M*". *lefkonensis* and "*M*". *cadeoti* (Table 2). The only known species that have a smaller size than these two forms are "*M*". *jaegeri* from Salmendingen (MN 10) [55] and "*M*". *delphinensis* from several Middle Miocene sites of Europe (Table 3). However, only the lower dentition of these two taxa is known, so more complete and associate material is needed to test if they are present in Hammerschmiede. Therefore, we prefer to refer to this form as "*Martes*" sp.

Genus *Circamustela* Petter, 1967 [36]

**Type species:** *Circamustela dechaseauxi* Petter, 1967 [36]

**Other included species:** *Circamustela peignei* Valenciano et al., 2020 [62]; *Circamustela hartmanni* sp. n.

**Remarks:** The genus *Circamustela* is a rare hypercarnivorous member of the subfamily Guloninae (sensu [79], including martens, wolverines and their relatives). It was firstly

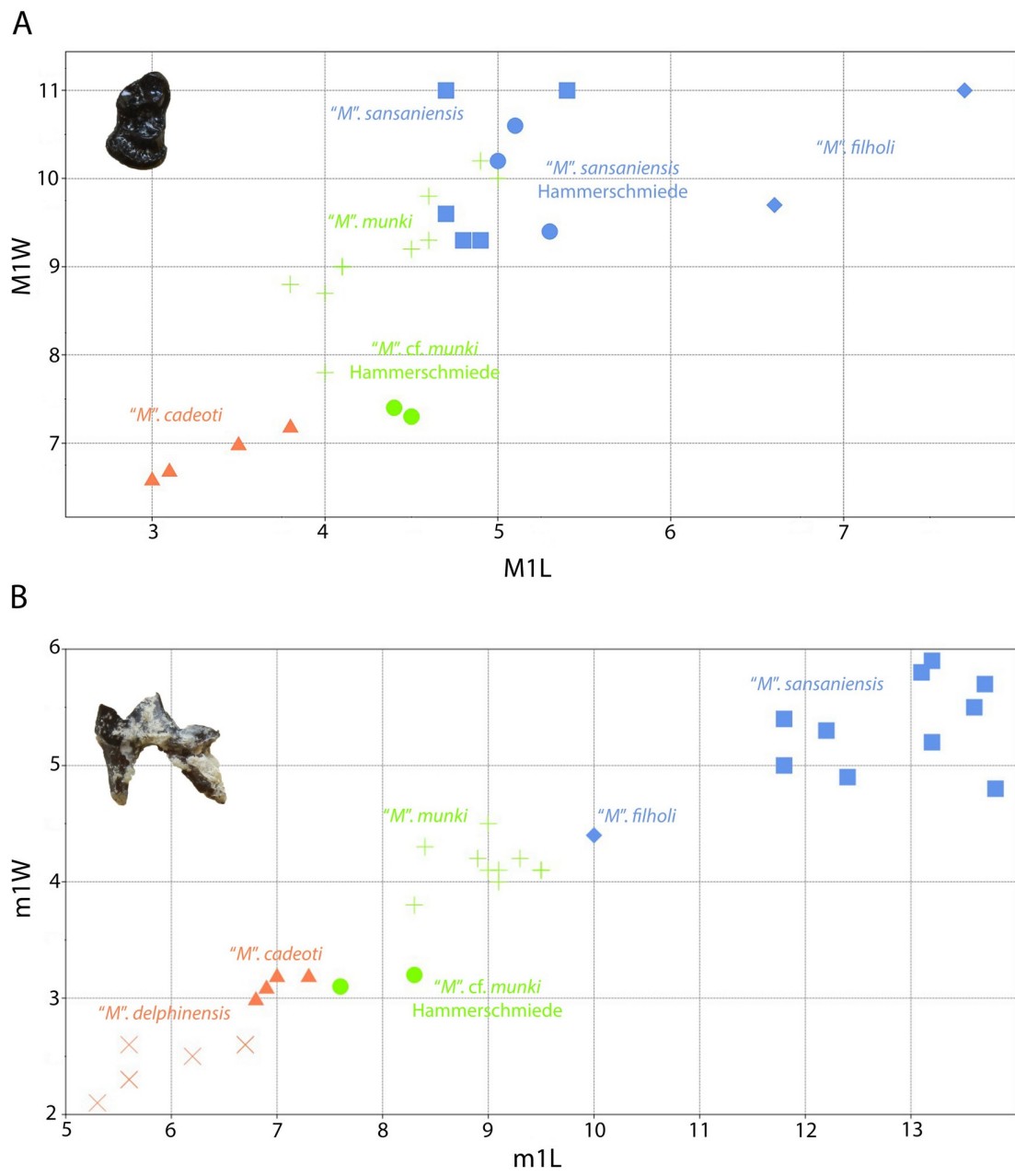

**Fig 4.** Comparison of the M1 (A) and m1 (B) of *"Martes"* from Hammerschmiede to that of other species of the genus. Data sources similar to that of Tables 2 and 3.

described by [36] based on a mandibular fragment with m1 and a damaged p4 from the early Late Miocene (MN 9) Spanish locality of Can Llobateres. Some years later, the same author published an upper molar from the same locality [78]. Both specimens were attributed to the type species *Circamustela dechaseauxi*. Some years later, [80] published a fragmentary M1 (lingual platform) from the early Late Miocene (MN 9) locality of Los Valles de Fuentidueña in Spain that they also attributed tentatively to *C. dechaseauxi*. Another report of the genus was made by [81], who described an M1 and a p4 from the Turolian locality Dorn-Dürkheim (Germany) as? *Circamustela* sp. Recently, [62] published a second species of the genus,

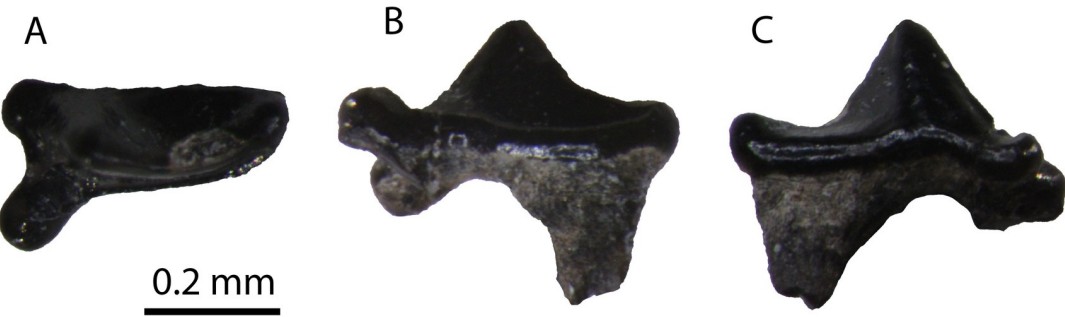

**Fig 5.** The P4 of *"Martes"* sp. from Hammerschmiede (SNSB-BSPG-1973-XIX-34) in (A) occlusal, (B) ventrolingual and (C) buccal view.

*Circamustela peignei*, from the sites Batallones 3 and 5 (Late Miocene, MN 10, Spain) that differs from *C. dechaseauxi* mainly in the morphology of M1, but also in the more developed m1 metaconid and more conical m1 hypoconid. The herein presented material represents a new species for the genus, named as *Circamustela hartmanni* n. sp.

*Circamustela hartmanni* Kargopoulos et al. sp. nov. urn:lsid:zoobank.org:act:670F9140-2E0F-4F51-96A7-C4935B8A4FAA.

**Holotype:** GPIT/MA/17238, right hemimandible with p3–m1 and alveoli of p1, p2 and m2.

**Hypodigm:** HAM 1: SNSB-BSPG-1973-XIX-23, right P4. HAM 4: GPIT/MA/17033, left hemimandible with fragmentary m1 and complete m2. HAM 5: GPIT/MA/10388, right P4.

**Etymology:** The name *hartmanni* was chosen to acknowledge the help of Antonie Hartmann that has granted the permission for excavations in her land in Hammerschmiede for all these years.

**Type locality:** HAM 4 (Germany).

**Other Localities:** HAM 5 and HAM 1 (Germany).

**Stratigraphy:** Base of the Tortonian (11.44 Ma for HAM 4 and 11.62 for HAM 5).

**Diagnosis:** Species of the genus *Circamustela* with approximately 80% the size of *C. dechaseauxi* and *C. peignei* (P4L≈6.5 mm; m1L≈8.0 mm); P4 protocone slender with a long neck; p3 and p4 with developed distal accessory cuspids; lower carnassial with short crown and metaconid of intermediate development between *C. peignei* and *C. dechaseauxi*.

**Differential Diagnosis:** Differs from *C. dechaseauxi* in the smaller size (m1L 7.8 mm in contrast to the 9.7 mm for the Can Llobateres species) and the more developed m1 metaconid. Differs from *C. peignei* in the smaller size (P4L = 6.4 mm in contrast to 7.9–8.6 mm for the Batallones species), the presence of p3 distal accessory cusp, a much more developed p4 accessory cuspid (virtually absent in *C. peignei*), and a slenderer m1 talonid.

**Description:** The specimens GPIT/MA/10388 and SNSB-BSPG-1973-XIX-23 (Fig 6) are two complete P4, with signs of wear in their carnassial blades. They are three-rooted, with the roots under the mesial border and the protocone being in close proximity. The paracone is the largest cusp and it is connected with the metastyle via a crest without a notch. There is no parastyle. The protocone is low, has a long neck and it is mesially situated. Their only differences consist of the rougher enamel surface and the more developed lingual cingulum in SNSB-BSPG-1973-XIX-23.

The hemimandible GPIT/MA/17238 (Fig 7A) retains p3–m1 and the alveoli for p1, p2 and m2, whereas the specimen GPIT/MA/17033 (Fig 7B) retains the talonid of m1 and a complete m2. The mandibular body is slender and dorsoventrally short. Two mental foramina are present: one below the mesial half of p2 and one below the distal half of p3. The masseteric fossa ends at the plane of the distal end of m2. No developed cingula are present in any of the teeth.

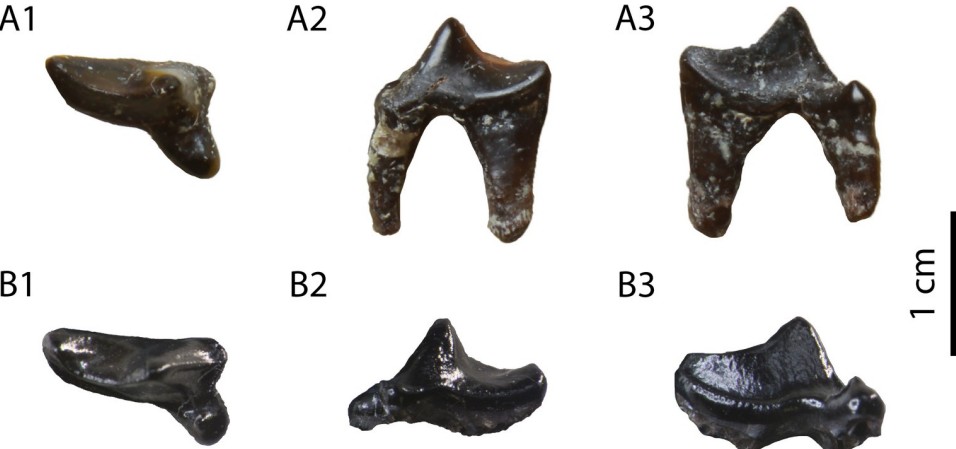

**Fig 6.** The two P4 (GPIT/MA/10388) of *Circamustela hartmanni* sp. n. from Hammerschmiede: (A) GPIT/MA/10388 and (B) SNSB-BSPG-1973-XIX-23 in occlusal, buccal and lingual views.

The second premolar is two-rooted and of similar size as the p3. The third premolar is relatively acute with a small distal accessory cuspid. The fourth premolar is damaged. However, it also seems to have been relatively high and it possesses a large and high distal accessory cuspid. The lower carnassial is long, with a low talonid and a high trigonid. The trigonid has clear wear facets in the carnassial blade and it overlaps with the distal part of the p4. The protoconid is the largest cuspid, clearly higher than the paraconid, divided from the latter by a deep notch. The metaconid is very small and blunt. The talonid covers approximately 25% of the tooth length. It is buccolingually reduced. It has a centrally placed hypoconid, which is blunt, conical and rounded by the cingulid. The second molar is reduced and almost circular in outline. It exhibits only one small protoconid and its trigonid covers approximately 1/3 of the tooth length.

**Comparison:** The plethora of marten-like mustelid taxa during the Middle and Late Miocene creates a relatively obscure taxonomic spectrum for the group. The genera *Martes* and *Mustela* have been used as wastebaskets for several fragmentarily known species, so the affinities of every taxon are not always clear (e.g., [45, 62]). However, the genus *Circamustela* can be easily distinguished from most of the other mustelids by considering its size and carnassial

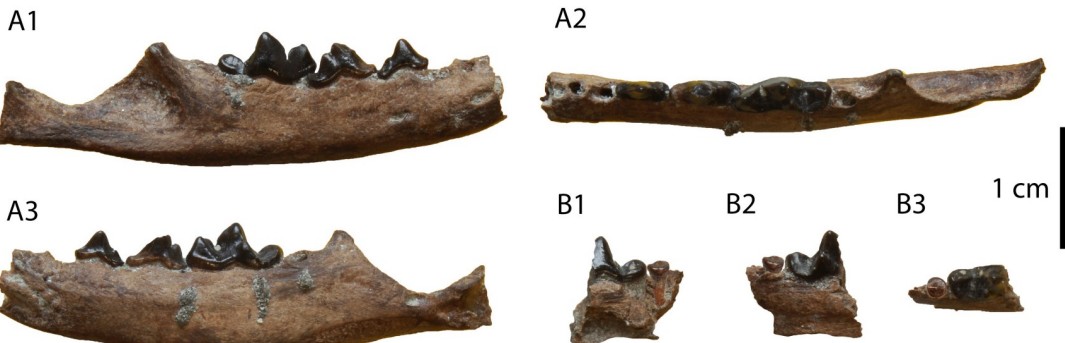

**Fig 7.** Lower dentition of *Circamustela hartmanni* sp. n. from Hammerschmiede: (A) right hemimandible (GPIT/MA/17238; holotype) in (1) buccal, (2) occlusal and (3) lingual views and (B) m1 and m2 (GPIT/MA/17033) in (1) buccal, (2) lingual and (3) occlusal views.

**Table 5. Metrical comparison of the dimensions of P4 and M1 of *Circamustela hartmanni* sp. n., *Circamustela* sp., *C. dechaseauxi* and *C. peignei* indicating the source of data.** Localities: HAM 5 (Hammerschmiede 5), LVF (Los Valles de Fuentidueña), DD (Dorn-Dürkheim), CLL (Can Llobateres) and BAT3 (Batallones 3).

| Species | Code | Locality | P4L | P4W | M1L | M1W |
|---|---|---|---|---|---|---|
| *Circamustela hartmanni* n. sp. | SNSB-BSPG-1973-XIX-23 | HAM 1 | 7.2 | 4.3 | | |
| | GPIT/MA/10388 | HAM 5 | 6.4 | 4.3 | | |
| *Circamustela* sp. | LVF-55-y [80] | LVF | | | 3.4 | |
| | DD-14 [81] | DD | | | 3.4 | 6.8 |
| *Circamustela dechaseauxi* | IPS 28086 [3] | CLL | | | 3.9 | 8.2 |
| *Circamustela peignei* | BAT-3'10.1570 [62] | BAT3 | 7.9 | 4.9 | 3.8 | 8.5 |
| | BAT-3'11.1041 [62] | BAT3 | 8.1 | 4.9 | | |
| | BAT-3'10.1246l [62] | BAT3 | (8.6) | | | |
| | BAT-3'10.1246r [62] | BAT3 | 8.3 | 4.3 | 3.9 | (8.8) |

morphology: the small dimensions, the low P4 metastyle, the high m1 protoconid, the low m1 metaconid and the mesiodistally enlarged p4 are herein considered sufficient enough for the attribution of the present specimens to the hypercarnivorous genus *Circamustela*. However, as noted in the past [45, 62, 81] the fossil record of the genus *Martes* is very problematic and a revision is needed in order to enable more secure attributions in the future.

It is evident from the metrical comparison with the upper (Table 5) and lower (Table 6) dentition, that the material described here is considerably smaller than that of *C. dechaseauxi* and *C. peignei*. In particular, most specimens seem to be approximately 20% smaller than the respective material of the other two species. Additionally, some morphological differences between the material from Hammerschmiede and that from Spain have been noted: the protocone of the upper carnassial is thinner and it has a longer neck than in *C. peignei*. The third lower premolar has a reduced distal accessory cuspid, which is absent in *C. peignei*. The distal accessory cuspid of the fourth lower premolar is much more developed than in *C. peignei*, which is absent in the majority of the specimens. The crown of the lower carnassial is in general shorter than in the other species of *Circamustela*, and the m1 metaconid is moderately reduced (intermediate between *C. peignei* and *C. dechaseauxi*). However, the slenderness of the m1 talonid of *C. hartmanni* is similar to that of the type species. The combination of metrical and morphological differences between the material from Hammerschmiede and that of Can Llobateres and Batallones are herein considered sufficient enough for its attribution to a

**Table 6. Metrical comparison of the lower teeth of *Circamustela hartmanni* n. sp. from Hammerschmiede and other material of the genus *Circamustela* indicating the source of data.** Localities: DD (Dorn-Dürkheim), and BAT3/5 (Batallones 3/5). Measurements on the holotype of *C. dechaseauxi* have been taken by the present authors.

| Species | Code | p3L | p3W | p4L | p4W | m1L | m1W | m2L | m2W |
|---|---|---|---|---|---|---|---|---|---|
| *Circamustela hartmanni* n. sp. | GPIT/MA/17238 | 4.0 | 1.9 | 5.4 | 2.1 | 7.8 | 3.0 | | |
| | GPIT/MA/17033 | | | | | | 2.8 | 1.9 | 1.8 |
| ?*Circamustela* sp. | DD-15 [81] | | | 5.0 | 2.4 | | | | |
| *Circamustela peignei* | BAT-3'10.1246l [62] | 5.1 | 2.5 | 6.1 | 2.9 | | | | |
| | BAT-3'10.1246r [62] | 5.0 | 2.5 | 6.2 | 2.8 | 10.0 | 4.1 | (2.4) | (1.7) |
| | BAT-3'13.1056 [62] | 4.8 | 2.5 | 6.0 | 2.9 | 9.2 | 3.8 | 3.5 | 2.8 |
| | BAT-3'13.1048 [62] | 5.3 | 2.4 | 5.8 | 2.9 | 9.4 | 3.9 | 3.4 | 2.7 |
| | BAT-3'10.1570A [62] | - | - | 6.5 | 3.0 | 9.4 | 3.6 | | |
| | BAT-3'10.1570B [62] | 5.2 | 2.6 | 6.1 | 2.8 | 9.1 | 3.7 | | |
| | BAT-5'10.G14.129 [62] | | | | | 9.0 | 4.1 | | |
| *Circamustela dechaseauxi* | IPS 2016 | | | | | 9.7 | 3.9 | | |

new species. The interpretation of this size difference as sexual dimorphism is presumed to be unfounded for the time being, since recently [62] studied material of several individuals of *C. peignei* without finding any sign of sexual dimorphism.

[80] published a fragmentary M1 from Los Valles de Fuentidueña, which they attributed to the type species *C. dechaseauxi*. However, the size of the tooth is considerably smaller (approximately 20%) than that published in [36] and [62]. Additionally, [81] published one M1 and one p4 from Dorn-Dürkheim as? *Circamustela* sp., based on their small size in comparison to *C. dechaseauxi*. The size of the material from Los Valles de Fuentidueña and Dorn-Dürkheim is similar to that of *C. hartmanni*. This could be interpreted as an indication for a common taxonomic attribution for all these forms, but, since the limited material does not enable a secure identification, it is herein preferred to retain the uncertain status of these specimens.

One further similar small marten-like mustelid is *Aragonictis araid* from the Middle Miocene of Spain (12.65–11.33 Ma) [45]. Both forms are similar at the first glance, but *C. hartmanni* clearly differs from *A. araid* in the presence of marked cingulids in the lower dentition, a distal accessory cuspid in p3 and p4, a relatively enlarged p4, m1 with more conical and rounded hypoconid and a higher entocristid, a reduced m2 with a single cuspid, as well as a relatively shorther P4 with a less individualized protocone. Lastly, *C. hartmanni* resembles the morphology "*M*". *jaegeri* from Salmendingen (MN 10) in some traits. However, it is much smaller (m1L = 5.5 mm; [55]) and the lower carnassial has a more marked buccal cingulid and a high bulbous hypoconid that makes it a more robust tooth. A metrical comparison of the species of Circamustela to Aragonictis can be seen in Fig 8.

Genus *Laphyctis* Viret, 1933 [82]

**Type Species:** *Laphyctis mustelinus* Viret, 1933 [82]

**Other included species:** *Laphyctis*? *comitans* Dehm, 1950 [46]

**Remarks:** *Laphyctis* Viret, 1933 [82] is very closely related to the Middle Miocene genus *Ischyrictis* Helbing, 1930 [83]. The latter has been used as a wastebasket of medium- to large-sized gulonines that are now distributed also to the genera *Dehmictis* Ginsburg & Morales, 1992 [84] and *Hoplictis* Ginsburg, 1961 [85]. This lineage also includes the genus *Iberictis* Ginsburg & Morales, 1992 [84]. The systematic position of the basal gulonine *Laphyctis* is unclear. It has been classified either as a subgenus of *Ischyrictis* (e.g. [76, 85]), as a valid genus [84] or as a junior synonym of *Ischyrictis* (e.g. [72]). Recently it has been re-validated [86, 87] and herein we follow this hypothesis.

*Laphyctis mustelinus* Viret, 1933 [82]

**Holotype:** UCBL-213784, right fragmentary maxilla with P3-4 and M1.

**Type Locality:** La Grive-Saint-Alban (France).

**Referred Specimens:** HAM 4: SNSB-BSPG-2020 XCIV-3395, right M1.

**Description**: The upper molar (SNSB-BSPG-2020 XCIV-3395; Fig 9) exhibits a relatively plesiomorphic gulonine morphology. The paracone is the largest cusp being significantly longer and higher than the metacone. The buccal part of the paracone is far more developed than that of the metacone. The border of the tooth is slightly constricted just lingually to the buccal cusps. A very small paraconule is present lingually to the paracone, followed by a moderately developed protocone. The protocone is mesially placed. The distal crest of the protocone ends at the plane of the paracone. No signs of any other cusps are present. The enamel surface of the tooth is wrinkled and the lingual platform is moderately developed, being circular in its lingual area.

**Comparison**: This relatively plesiomorphic upper molar differs from the specimens of "*Martes*" by its very large size (Table 7), the absence of any cusps in its distolingual part, the considerable difference in the buccal expansion of the paracone in comparison to that of the metacone, the restricted distal crest of the protocone and the outline constriction lingually to

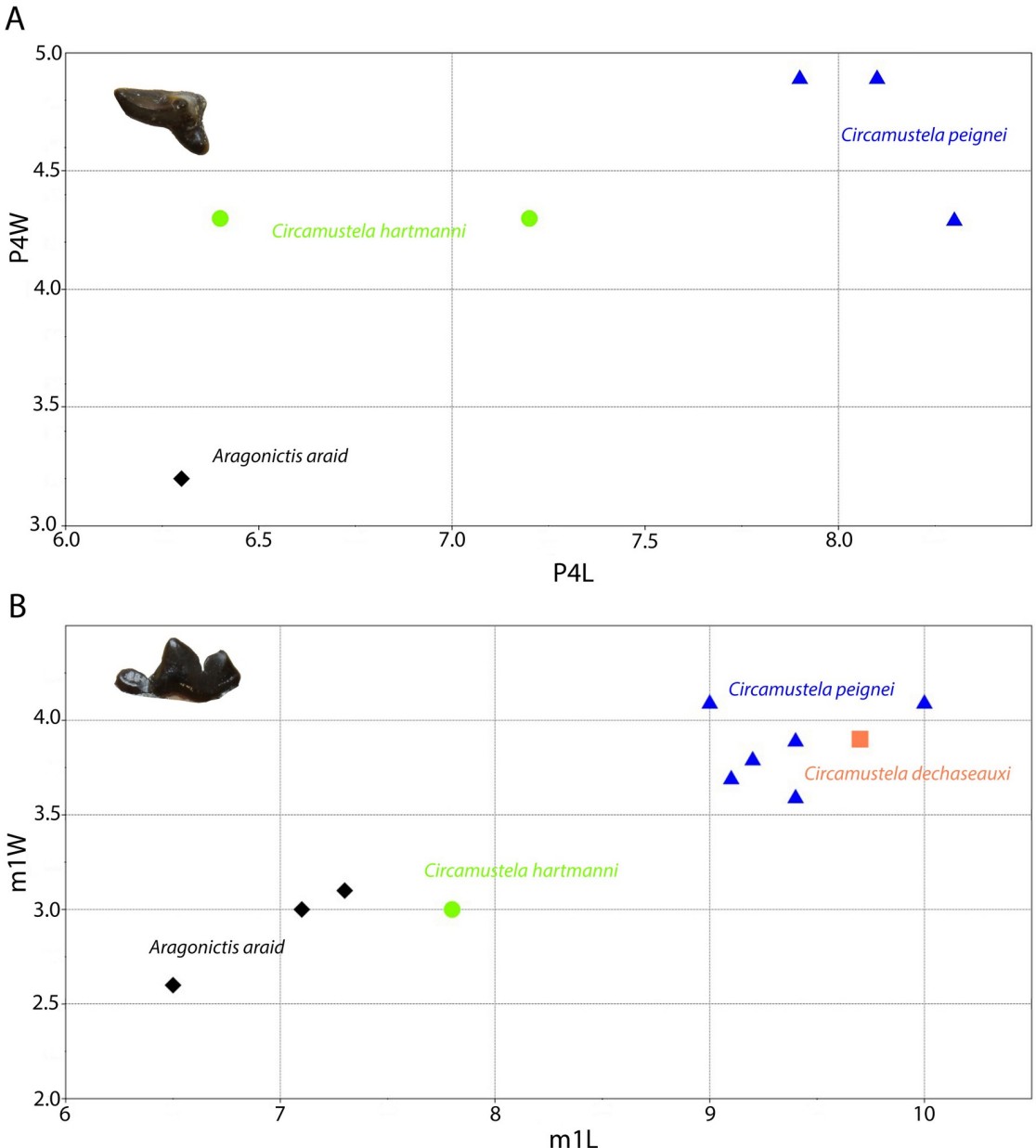

**Fig 8.** Comparison of the P4 (A) and m1 (B) of *Circamustela hartmanni* from Hammerschmiede to that of *Circamustela peignei*, *Circamustela dechaseauxi* and *Aragonictis araid*. Data sources: *C. peignei* [62], *C. dechaseauxi* (personal data) and *A. araid* [45].

the buccal cusps. It differs from the genus *Iberictis* in the absence of a metaconule, a more reduced metacone and a more circular lingual platform, missing the incisure in its central area [88]. It differs from the genus *Hoplictis*, in the more developed lingual platform and the larger metacone [87]. It differs from *Dehmictis* by the more reduced protocone and metacone [84]. The morphology of the cusps and outline of the tooth seem to fit better with that of the genus *Ischyrictis* and *Laphyctis*. The genus *Ischyrictis* traditionally included three Middle Miocene species: the scarcely known *Ischyrictis bezianensis* Ginsburg & Bulot, 1982 [75], the better-known and larger form *Ischyrictis zibethoides* [17, 89], and *I. mustelinus* (herein considered as *Laphyctis mustelinus*). The herein described specimen differs from the two species of *Ischyrictis*

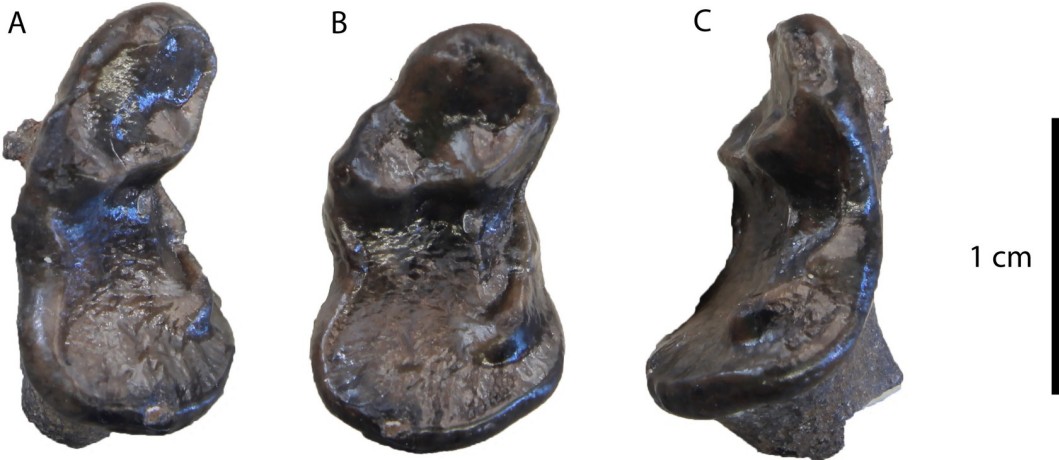

**Fig 9.** The right upper molar attributed to *Laphyctis mustelinus* (SNSB-BSPG-2020 XCIV-3395) in disto-occlusal (A), occlusal (B) and mesio-occlusal view (C).

on the basis of the evident constriction lingual to the buccal cusps, the higher width of the tooth, the less developed paracone and metacone, the more developed buccal part of the paracone (in comparison to the metacone) and the larger lingual platform of *L. mustelinus* (e.g., [52, 75, 84]). Based on the previous described characters, the Hammerschmiede molar fits to the holotype of *L. mustelinus* from La Grive MN 7/8 [82]. Both forms share the same morphology of the parastylar area and the lingual platform. SNSB-BSPG-2020 XCIV-3395 only differs in a slightly lesser incisure below the paracone-metacone and in a more mesial position of the protocone. However, these differences can be explained by intraspecific variability. Additionally, this species has been found in Central Europe in the localities of Steinheim (MN 7; [90]), and Erkertshofen 2 (MN 4; [76]), as well as in Western Europe in Can Mata 1 (MN 7/8; [30, 58, 91]).

Guloninae indet.

**Referred Specimens:** HAM 5: GPIT/MA/10297, a left hemimandible with p2–m1 and the alveoli of p1 and m2.

**Description:** The hemimandible preserves p2–m1, as well as the alveoli of p1 and m2 (Fig 10). There are two mental foramina: one smaller below the distal part of p3 and one larger

**Table 7. Comparison of the dimensions of SNSB-BSPG-2020-XCIV-3395 with that of other upper molars of Miocene gulonines indicating the data source.**

| Species | Code/Locality | M1L | M1W |
|---|---|---|---|
| *L. mustelinus* | SNSB-BSPG-2020 XCIV-3395 | 8.6 | 14.4 |
| | Vieux-Collonges [52] | 6.2–7.3<br>6.7 (4) | 12.7–14.0<br>13.7 (4) |
| | La Grive [27] | 6.5–8.0<br>7.3 (2) | 12.2–15.4<br>13.8 (2) |
| | Hostalets de Pierola [30] | 8.0 | 15.0 |
| *I. zibethoides* | Sansan [72] | 6.9–10.8<br>9.2 (5) | 13.8–18.0<br>16.0 (4) |
| | Sandelzhausen [73] | 7.6–9.1<br>8.6 (3) | 14.4–16.3<br>15.4 (3) |
| | Artenay [52] | 7.3 | 17.6 |
| *I. bezianensis* | Bézian [75] | 8.1 | > 13.0 |

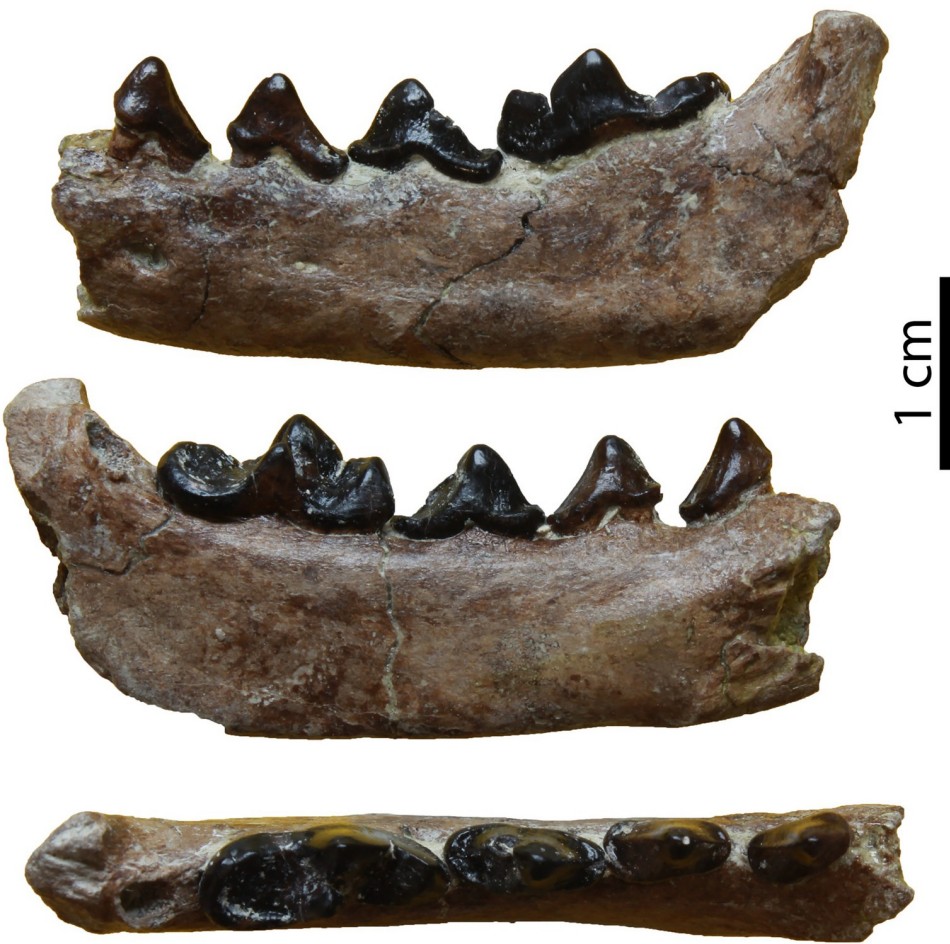

**Fig 10. The hemimandible attributed to Guloninae indet.** (GPIT/MA/10297) from Hammerschmiede.

below p2. The alveolus of p1 is small and circular, being in close contact with the alveolus of c and p2. All three remaining premolars have a smooth cingulid, which becomes stronger towards p4 and they also become more elongated and less high towards p4. The main cuspids are mesially located in the premolars. Both p2 and p3 are unicuspid. The p4 has a small damaged distal accessory cuspid. The lower carnassial is relatively long and low, with the talonid covering approximately 30% of the tooth length. There are signs of wear both in the carnassial blade and in the talonid valley. The trigonid is long with faint grooves at its buccal surface. The protoconid is the largest cuspid, slightly distally bent and separated from the slightly mesially bent paraconid by a shallow notch. The metaconid is clearly lower than the other two cuspids, being blunt and lingually bent. It stems from the base of the tooth and not from the middle height of the protoconid, of which it is slightly distally located. A small beveled hypoconid is present at the buccal border, as well as a very small hypoconulid. No other cuspids occur in the talonid. The talonid valley is shallow and wide with a marked distal border. Dental measurements of the specimen are given in Table 8.

**Table 8. Dental dimensions of the hemimandible identified as Guloninae indet.** (GPIT/MA/10297).

| Code | p2L | p2W | p3L | p3W | p4L | p4W | m1L | m1W | m2L | m2W |
|---|---|---|---|---|---|---|---|---|---|---|
| GPIT/MA/10297 | 5.5 | 2.8 | 6.2 | 3.1 | 7.5 | 3.5 | 11.8 | 4.8 | [3.7] | [2.2] |

**Comparison:** The identification of this specimen is retained as Guloninae indet., because it exhibits significant differences with all the mustelid Miocene forms as far as we know. In general, it has a relatively plesiomorphic morphology, differentiating it from the more complex forms of mephitids (developed m1 entoconid, shortened rostrum), melines (more developed m1 talonid, small cuspules in the m1 cingulum) and lutrines (developed cingulum, sharp cuspids). It is also significantly smaller than *Plesiogulo* and *Eomellivora* and far larger than the small mustelines-gulonines like *Circamustela*. It also retains the p1, differentiating it from *Trochictis* and the mephitids. Usually, such characteristics fit better with the plesiomorphic gulonines such as "*Martes*" spp. However, it differs from "*Martes*" in the low height of m1 and the premolars, the elongated p4 and the small and ventrally situated distal accessory cuspid of p4. It differs from *Ischyrictis* and *Laphyctis* in the lower m1 trigonid, the longer m1 talonid and the slender premolars. The size, the low cuspids of the cheek teeth and the general morphology of the m1 cusps resemble *Martes melibulla* [58, 62]. However, the present mandible differs from that species by the shorter p2, the shape of the accessory cuspid in p4, the larger m1 talonid and the mesially inclined m2. The size, the low m1, the long p4, the accessory cuspid of p4 and the p2–p3 diastema are similar to that of *Sinictis pentelici* [92]. However, the Hammerschmiede specimen has a lower p4, not so distinct p2–p3 diastema, a lower m1 hypoconid and the p4 is only slightly overlapping with m1. Therefore, we suggest that this form is identified as Guloninae indet. until further findings clarify its affinities.

Subfamily Mellivorinae Gray, 1865 [93]

Genus *Eomellivora* Zdansky, 1924 [77]

**Type species:** *Eomellivora wimani* Zdansky, 1924 [77]

**Other included species:** *Eomellivora fricki* (Pia, 1939) [94]; *Eomellivora hungarica* Kretzoi, 1942 [95]; *Eomellivora ursogulo* (Orlov, 1948) [96]; *Eomellivora piveteaui* Ozansoy, 1965 [97]; *Eomellivora moralesi* Alba et al., 2022 [98].

**Remarks:** This genus includes Late Miocene giant mustelids from Asia, Europe and North America [98–100]. The Hammerschmiede material represents the earliest record of the genus (HAM 5; 11.62 Ma; [1]), given the fact that the type locality of *E. moralesi* from Abocador de Can Mata (ACM/PTA-A2) has been dated to 11.21 Ma [98]. The species *E. moralesi*, found in the MN 7/8 of Spain, has been considered as the most basal form of the genus, followed by the Vallesian *E. fricki* and *E. piveteaui* in the MN 9–10 that exhibit evolutionary trends towards the more derived *E. ursogulo* (MN 11), *E. wimani* (MN 12–13) and *E. hungarica* (MN 13) [98, 99]. It must be mentioned that [101] proposed an alternative, more simplistic approach, considering that only *E. wimani* is a valid species.

*Eomellivora moralesi* Alba et al., 2022 [98]

**Holotype:** ICP-IPS122262, palate with incisor, canine and left P1 alveoli, as well as left P2–M1 and right P1–M1 crowns.

**Type Locality:** ACM/PTA-A2, Abocador de Can Mata (Spain).

**Referred Specimens:** HAM 5: GPIT/MA/09877, left I3; GPIT/MA/12347, left p3; GPIT/MA/09875, right hemimandible with p4–m1; GPIT/MA/10302, left m1; GPIT/MA/09632, left m2.

**Description:** The specimen GPIT/MA/09877 (Fig 11A) is a left I3 without a root. It is robust, buccolingually compressed and it exhibits a strong lingual fold. Two crests start from the tip of the crown: one shorter, that meets mesially the lingual fold, and one longer, that ends to the distal border of a smooth cingulum. The buccal surface of the tooth is marked with very shallow grooves.

The only available p3 is complete, missing only its mesial root (Fig 11D). It is relatively worn. It is slightly asymmetrical as its distal part is slightly longer and wider than its mesial part. The distal base of the tooth is wide and slightly elevated. A faint cingulid is present in its

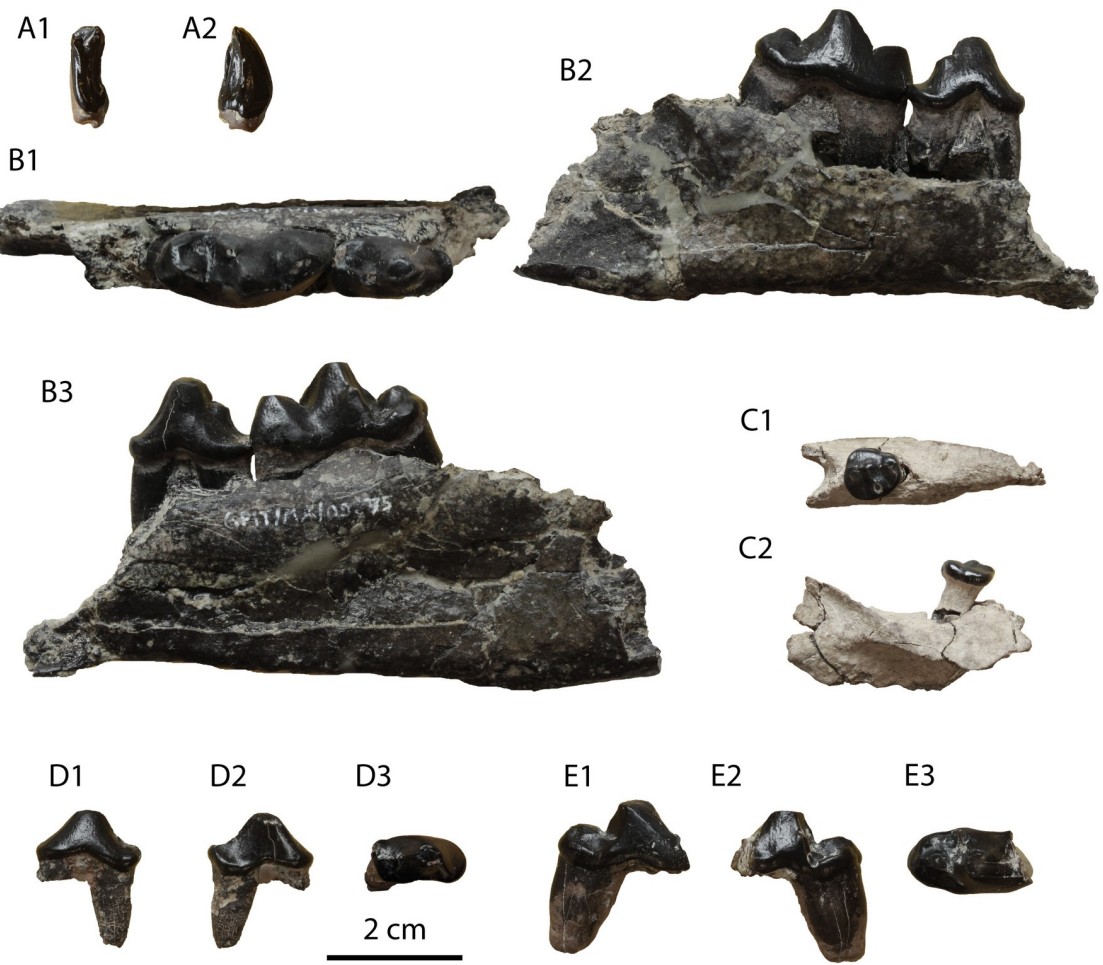

**Fig 11.** *Eomellivora moralesi* from Hammerschmiede: (A) GPIT/MA/09877 left I3, (B) GPIT/MA/09875 right p4–m1, (C) GPIT/MA/09632 left m2, (D) GPIT/MA/12347 left p3 and (E) GPIT/MA/10302 left m1.

lingual side. There are no signs of accessory cuspids. The enamel is relatively thick with indistinct grooves.

The specimen GPIT/MA/09875 consists of a right partial hemimandible with p4 and m1 (Fig 11B). The description for m1 also applies to GPIT/MA/10302 (Fig 11E). The mandibular corpus is relatively high with a moderately deep masseteric fossa. The mesial end of the masseteric fossa is at the plane of the distal border of m1. The fourth premolar is worn and asymmetrical with its distal side being longer and wider than its mesial side. There is no mesial accessory cuspid. A strong distal accessory cuspid is present close to the main cuspid. The tooth is not inclined distally. The height of the main cuspid is higher than that of the paraconid of m1 and comparable to the one of m1 protoconid. The enamel is strong and faintly wrinkled in its surface. The lower carnassial is long and robust. It is moderately worn in its trigonid and talonid cuspids. A faint cingulid surrounds the tooth, being slightly more distinct in the talonid. The protoconid is the largest cuspid covering almost half of the tooth length. The paraconid is robust and vertical. A prominent carnassial notch is formed between these two cuspids. A small metaconid is present in the lingual side of the protoconid. The talonid is relatively short, covering approximately 25% of the tooth's length. Its valley is very short as it is restricted to the part distally to the metaconid. There is only one talonid cuspid present, the hypoconid,

which is relatively long and high, being buccaly situated, close to the protoconid. However, the presence of a small hypoconulid cannot be excluded in the distal part of the talonid, which is affected by dental wear.

The m2 (Fig 11C) is still in its alveolus and it has a developed cingulid. It is slightly longer than broad (m2W/m2L = 92%). The trigonid is wider than the talonid, but they have approximately the same length. The mesiolingual part of the trigonid is slightly oblique ventrally. Four cuspids are present: a mesial paraconid, a central protoconid, a lingual metaconid and a small hypoconid at the distal end. All of them have approximately the same height and length. The protoconid is connected with all of the other cusps, resulting to a T-shaped structure of faint crests.

**Comparison:** The size of these specimens indicates that they must belong to a considerably large-sized mustelid. Given the already discussed presence of *Laphyctis mustelinus* in Hammerschmiede, a comparison with this species and the similar *I. zibethoides* is here attempted.

Unfortunately, the only reported I3 of *L. mustelinus* has been reported by [27] without providing description, measurements or figures. Therefore, a direct comparison based on this tooth is for the moment not possible. However, *L. mustelinus* has a comparable (and slightly smaller) size than *I. zibethoides* (Tables 7 and 9). Based on the dimensions given [72] for the Sansan material and by [73] for the Sandelzhausen material, the specimens from Hammerschmiede are significantly larger than those of *I. zibethoides* (Table 9). Therefore, it can also be deduced that the I3 of *L. mustelinus* would also be smaller than the Hammerschmiede incisor. [72] provided measurements for three mandibles of *L. mustelinus*, originating from La Grive-Saint-Alban, Steinheim and Erkertshofen. Additionally, [27] reported measurements for an m2. Comparing these measurements with the values for the present material, it is again clear that all the Hammerschmiede specimens are far larger than *L. mustelinus* (Table 9). Additional differences based on the descriptions of [27] and [90] include: the smaller p4 distal accessory cuspid, the larger m1 metaconid, the wider m1 talonid basin, the smaller m1 hypoconid and the more elongated m2.

As mentioned before, the specimens from Hammerschmiede are significantly larger than those of *I. zibethoides* (Table 9). Additionally, there are considerable morphological differences between this form and the Hammerschmiede material: the third upper incisor of *I. zibethoides* is relatively small and plesiomorphic, without any ridges and grooves [72]. As in *L. mustelinus*,

**Table 9. Comparison of dental dimensions between the material of *Eomellivora moralesi* from Hammerschmiede and other relatable forms indicating the data source.**

| Species | Code | I3L | I3W | p3L | p3W | p4L | p4W | m1L | m1W | m2L | m2W |
|---|---|---|---|---|---|---|---|---|---|---|---|
| *E. moralesi* | GPIT/MA/09877 | 7.4 | 5.4 | | | | | | | | |
| | GPIT/MA/12347 | | | 14.6 | 7.1 | | | | | | |
| | GPIT/MA/09875 | | | | | 17.3 | 8.7 | 24.7 | 10.7 | | |
| | GPIT/MA/10302 | | | | | | | | 10.3 | | |
| | GPIT/MA/09632 | | | | | | | | | 7.6 | 7.0 |
| *E. moralesi* [98] | | [6.0] | [7.4] | [12.4] | 6.4 | | | | | 7.6 | 6.4 |
| *E. piveteaui* [100] | | 7.2 | 5.5 | 10.7–11.8 11.4 (3) | 7.3–8.4 8.0 (3) | 13.7–16.1 15.2 (5) | 7.3–8.8 8.1 (5) | 19.7–24.5 22.5 (7) | 8.1–9.4 8.7 (6) | 6.9–7.3 7.1 (2) | 4.9–5.8 5.4 (2) |
| *E. fricki* [99] | | | | | | 17.9 | 8.7 | 26.5 | 10.5 | [9.0] | [5.6] |
| *L. mustelinus* [27, 72] | | | | 8.1–8.3 8.2 (2) | 4.3–4.6 4.5 (2) | 9.7–11.1 10.2 (3) | 5.1–5.2 5.1 (3) | 14.5–15.5 15.1 (3) | 6.3–6.6 6.5 (3) | 6.5 | 5.5 |
| *I. zibethoides* [72, 73] | | 5.0 | 3.6 | 9.2–10.3 9.8 (15) | 4.4–5.2 4.8 (15) | 11.0–12.9 11.8 (19) | 5.0–6.3 5.7 (21) | 15.7–18.9 16.9 (19) | 6.5–8.7 7.2 (19) | 6.0–6.3 6.2 (2) | 5.6–6.0 5.8 (2) |
| *I. bezianensis* [102] | | | | | | 9.1 | 4.3 | 14.1 | 6.0 | | |

*I. zibethoides* differs from the present specimens in the lower p4 accessory cuspid, the m1 protoconid that is much higher than the m1 paraconid, the larger m1 metaconid, the lower m1 hypoconid and the wider and flatter m1 talonid basin [72]. Finally, the cuspids of the m2 of *I. zibethoides* are situated closer to the borders of the tooth, creating a central valley in the middle of the molar [72].

*Eomellivora moralesi* is known based on a complete maxilla with P2–M1, and fragmentary hemimandibles including complete p2 and m2, and a broken p3 from the locality of Can Mata, dated to 11.21 Ma [98]. Consequently, only the alveolus of the I3, part of the p3, and the complete m2 are directly comparable.

The dimensions of the described specimens fit only to that of the two large-sized mustelid genera typical from the Middle-Late Miocene: *Eomellivora* and *Plesiogulo*. Morphological differences between the Hammerschmiede material and the genus *Plesiogulo* include: the presence of grooves and ridges in I3, the longer p3, the presence of a distal accessory cuspid in p4, the higher trigonid cuspids of m1 and the more developed cuspids of m2 [77, 103–105].

On the contrary, the overall morphology of the presented material fits perfectly with the genus *Eomellivora*. The identification of the incisor is based on the remarkable similarity of the described specimen with the I3 of *Eomellivora piveteaui* from the Late Miocene of Batallones 3 (BAT-3'09.688; [99]). This specimen has the same morphology as the Hammerschmiede incisor (lingual fold, the pair of ridges and buccal grooves). The measurements of the I3 alveolus of *E. moralesi* [98] are to some extent different from those of GPIT/MA/09877 (Table 9), but as seen in [98] the alveolus is clearly mediolaterally compressed. Therefore, the measurements given in [98] do not fit with the depicted morphology. This fact is herein interpreted as metrical bias and the measurements are considered only as indicators of the approximate size of the tooth, which is relatively similar to that of GPIT/MA/09877.

The dimensions of the p3 indicate that it is larger and relatively slenderer than that of *E. piveteaui* (Table 9). Additionally, this species is characterized by the presence of a distal accessory cuspid in p3 [100], which is absent in the present specimen. The size is also slightly larger and slenderer than that of the paratype of *E. moralesi*, but the dimensions of the latter were taken in the alveolus, so it can be expected that the real length of the tooth must have been slightly larger.

The p4 is of intermediate size between *E. piveteaui* and *E. fricki* (Table 9). The p4 of *E. piveteaui* is much more robust, exhibiting a mesial accessory cuspid and also having a stronger distal accessory cuspid [98]. The p4L in relation to m1L is slightly longer in comparison to *E. fricki*. Additionally, this species is characterized by a relatively smaller p4 distal accessory cuspid [99].

The lower carnassial is also intermediate in size between *E. piveteaui* and *E. fricki* (Table 9), being a similar proportion to that of *Eomellivora* sp. from Gritsev [99]. It differs from that of *E. piveteaui* in the presence of a metaconid, the non-buccolingually compressed hypoconid and the possible absence of the hypoconulid. Though, [100] stated that the presence of the hypoconulid is evident only in one unworn specimen, so it is possible that this trait might have been present in *E. moralesi* too. It differs from the m1 of *E. fricki* in the shorter talonid, the larger metaconid and the more buccally situated hypoconid. However, the preserved m1 of *E. fricki* (NHMW-2016/0065/0001) is extremely worn [99], not enabling a solid comparison.

The m2 of *E. piveteaui* has a relatively long trigonid (75% of the total length) and it does not have a metaconid [100]. In contrast, the second lower molar of *E. moralesi* has a more symmetrical ratio between trigonid and talonid and it possesses a metaconid [98], fitting to the present specimens. However, there is a difference in the width/length ratio between the two specimens, as the lectotype of *E. moralesi* has a ratio of 84%, whereas the molar from Hammerschmiede has a ratio of 92%. Finally, the m2 of *E. fricki* (judging from its alveolus) is far more elongated

and larger than GPIT/MA/09632 [99]. Therefore, due to the same temporal range and similar morphology of the Hammerschmiede *Eomellivora* with the late Aragonian *Eomellivora moralesi*, we assign it to this taxon. These findings confirm the validity of this species, and undoubtedly differentiate it from the Vallesian *E. fricki* and *E. piveteaui*. It represents the earliest record of the genus (HAM 5; 11.62 Ma; [1]) and the first report of its p4 and m1.

Subfamily Lutrinae Bonaparte, 1838 [106]

Genus *Vishnuonyx* Pilgrim, 1932 [107]

**Type species:** *Vishnuonyx chinjiensis* Pilgrim, 1932 [107]

**Other included species:** *Vishnuonyx? angololensis* Werdelin, 2003 [108]; *Vishnuonyx maemohensis* Grohé et al., 2020 [109]; *Vishnuonyx neptuni* Kargopoulos et al., 2021 [22].

**Remarks:** Four different species have been attributed to the genus *Vishnuonyx*. The oldest species is *V. maemohensis*, which has been described from the middle-late Middle Miocene locality of Mae Moh in Thailand [109]. The type species *V. chinjiensis* has been reported from the late Middle / early Late Miocene of India [105, 110–112] and Kenya [113, 114]. [108] described the species *V.? angololensis* based on an upper carnassial from the late Late Miocene of Lothagam, but the attribution of this form to the genus *Vishnuonyx* has been doubted [115, 116]. [115] published a mandible from the Early Pliocene of Haradaso (Ethiopia) as *Vishnuonyx* sp., which is the youngest known occurrence of the genus in the fossil record. Finally, the only report of this genus from Europe is that of [22] that erected the species *V. neptuni* based on material from Hammerschmiede. The present study includes additional material that was found in the recent excavations at the same locality.

*Vishnuonyx neptuni* Kargopoulos et al., 2021 [22]

**Holotype:** SNSB-BSPG-2020-XCIV-0301, a right hemimandible with p1 alveolus and complete p2–m1.

**Type Locality:** HAM 4 (Germany)

**Referred New Specimens:** HAM 4: SNSB-BSPG-2020 XCIV-5702, right M1; SNSB-BSPG-2020 XCIV-5700, right p4; SNSB-BSPG-2020 XCIV-4029, right m1; SNSB-BSPG-2020 XCIV-5701, left m1.

**Description:** The M1 (Fig 12A) is complete with slight signs of wear in the buccal cusps. Its outline is relatively rectangular with the mesial and distal border being parallel to each other. The paracone is longer and slightly higher than the metacone. A strong protocone followed mesiobuccally by a paraconule is evident in the mesial part of the tooth. A developed metaconule is present just lingually to the metacone. The cingulum is relatively strong, especially in its lingual part, and marked with small notches. The enamel surface is wrinkled.

The p4 (Fig 12B) is asymmetrical, being considerably wider distally. Its cingulid is strong, especially in its buccal side. The main cuspid is high and robust, followed distobuccally by a significantly developed distal accessory cuspid. The enamel surface of the tooth (especially on its buccal side) is wrinkled.

Both lower carnassials (Fig 12C and 12D) are fragmentary. A strong cingulid is present, being stronger mesiobuccally. The paraconid is not preserved in both specimens. The protoconid is the higher preserved cuspid (considerably higher than the metaconid) and it is followed distally by a marked hypoconid.

**Comparison:** The genus *Vishnuonyx* has been considered as member of the group of bunodont otters [117]. However, it exhibits some unique characteristics that differentiate it from more typical genera of this group (such as *Enhydriodon* Falconer, 1868 [118] or *Sivaonyx* Pilgrim, 1931 [54]) [22]. The upper molar is mesiodistally short and its lingual platform is reduced; the accessory cuspid of p4 is relatively fused mesially with the main cuspid and the talonid is shorter than the trigonid, surrounded by a crenulated rim [22]. All these characteristics fit perfectly with the herein presented specimens.

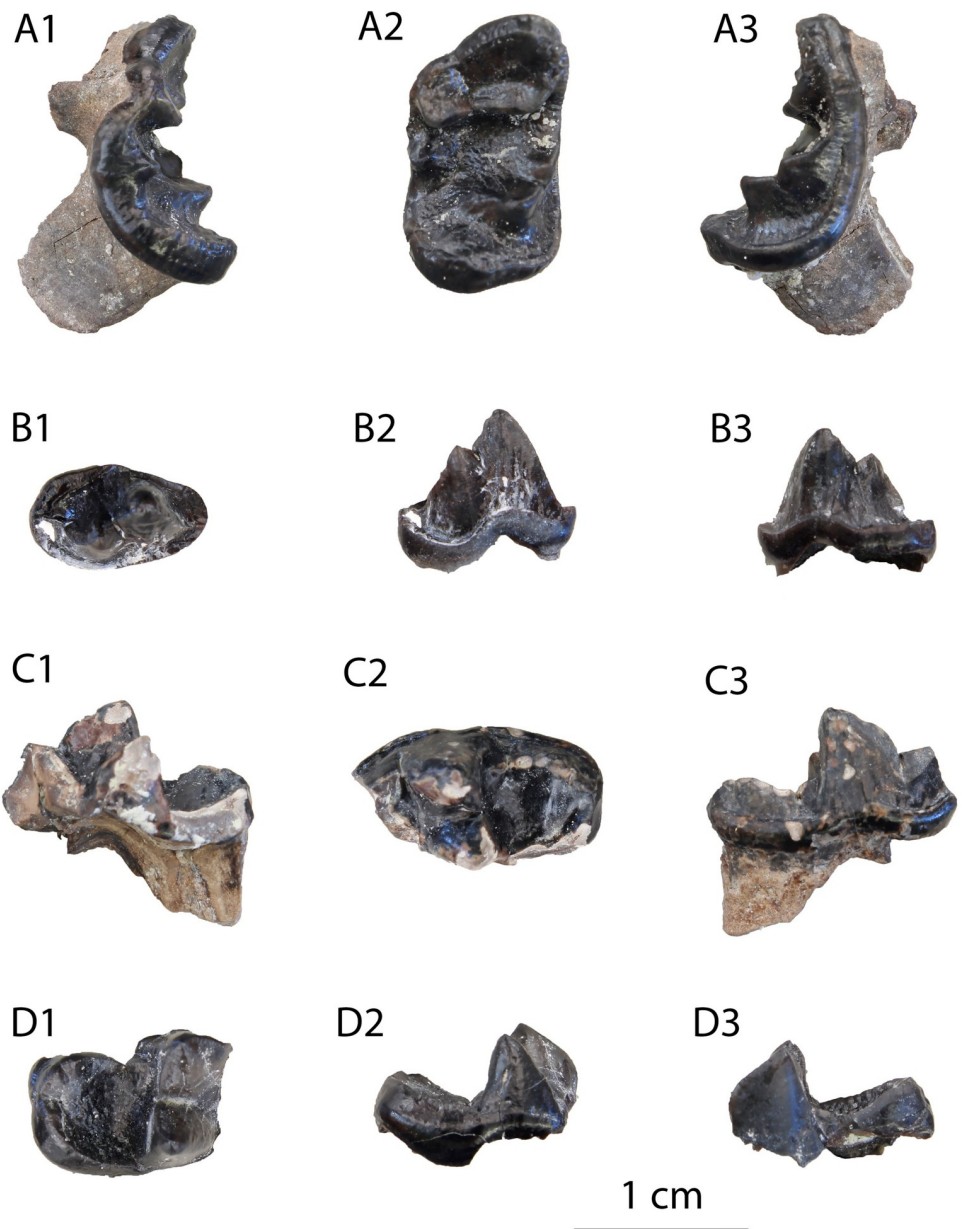

**Fig 12.** Material of *Vishnuonyx neptuni* from HAM 4: (A) SNSB-BSPG-2020 XCIV-5702 M1 in distal (A1), occlusal (A2) and mesial view (A3); (B) SNSB-BSPG-2020 XCIV-5700 p4 in occlusal (B1), buccal (B2) and lingual view (B3); (C) SNSB-BSPG-2020 XCIV-4029 right m1 in lingual (C1), occlusal (C2) and buccal view (C3); (D) SNSB-BSPG-2020 XCIV-5701 right m1 in occlusal (D1), buccal (D2) and lingual view (D3).

The genus *Vishnuonyx* includes *V. neptuni* in Europe, *V. chinjiensis* in Asia and Africa, *V. maemohensis* in Asia and possibly *V?. angololensis* in Africa. These four species are differentiated based on their size (Table 10; [22]), as *V. chinjiensis* and *V. maemohensis* are relatively small, *V. neptuni* is of intermediate size and *V?. angololensis* is relatively large. Other morphological characteristics of *V. neptuni* that differentiate it from the other species of the genus include the small M1 paraconule, the large protoconule and metaconule, the more expanded buccal outline of the paracone, the shorter lower premolars and the m1 trigonid being slightly wider than the m1 talonid. All of these characteristics are clearly seen in the herein described

**Table 10. Metrical comparison of the dimensions of M1, p4 and m1 from the herein presented specimens of *Vishnuonyx neptuni*, with previously published material from this species and other species of the genus indicating the source of data.**

| Species | Code | M1L | M1W | p4L | p4W | m1W |
|---|---|---|---|---|---|---|
| *V. neptuni* | SNSB-BSPG-2020 XCIV-5702 | 8.1 | 14.2 | | | |
| | SNSB-BSPG-2020 XCIV-5700 | | | 9.2 | 5.5 | |
| | SNSB-BSPG-2020 XCIV-4029 | | | | | (7.2) |
| | SNSB-BSPG-2020 XCIV-5701 | | | | | (7.7) |
| | SNSB-BSPG-2020 XCIV-1552 [22] | 7.6 | 14.0 | | | |
| | GPIT/MA/16733 [22] | | | 8.9 | 6.2 | 7.7 |
| | SNSB-BSPG-2020 XCIV-0301 [22] | | | 9.0 | 5.7 | (7.3) |
| | SNSB-BSPG-2020 XCIV-1301 [22] | | | 9.9 | 6.5 | |
| *V. chinjiensis* [109] | | | | 7.2–7.3 7.3 (2) | 4.2–4.3 4.3 (2) | 5.9 |
| *V. maemohensis* [109] | | 5.0–5.9 5.4 (3) | 11.1–11.5 11.3 (2) | 6.7–8.3 7.6 (6) | 3.5–4.6 3.9 (4) | 4.8–6.4 5.7 (7) |

specimens, so they are attributed to the species *V. neptuni*, which is already known from HAM 4 [22].

Genus *Paralutra* Roman & Viret, 1934 [119]

**Type species:** *Paralutra jaegeri* (Fraas, 1862) [120]

**Other included species:** *Paralutra transdanubica* Kretzoi, 1951 [121].

**Remarks:** This genus is a typical member of the Middle Miocene assemblages of Europe, mainly represented by the type species *Paralutra jaegeri*, which was initially reported from Steinheim [90, 120]. However, the stratigraphic range of this species covers from the Early Miocene (Pellecahus; MN 4; [119]) to the early Late Miocene, as it has been found in Rudabánya (MN 9; [65]). The species "*Paralutra garganensis*" Willemsen, 1983 [122], was recently suggested to belong to a different genus [123].

*Paralutra jaegeri* (Fraas, 1862) [120]

**Holotype:** SMNS-4082, a left maxilla with P2–P4.

**Type Locality:** Steinheim (Germany).

**Referred Specimens:** HAM 5: GPIT/MA/10393, left P4; GPIT/MA/12322, left M1. HAM 4: SNSB-BSPG-2020 XCIV-5704, left M1; SNSB-BSPG-2020 XCIV-5703, left M1.

**Description:** The specimen GPIT/MA/10393 (Fig 13A) is a complete left P4, missing only part of its roots. It exhibits small facets of wear. The tooth is relatively short with no developed cingulum. The paracone is the highest cusp, but it is considerably low. The metastyle is extremely small and worn, forming a very low crest with the paracone. The parastyle is restricted to a tiny worn cusp at the preparacrista. The protocone is wide, high and acute, with a developed medial shelf that hosts a small hypocone. The mesial border of the protocone is in the same plane as the total distal tooth end. Two additional ridges are present: one connecting the protocone with the parastyle and one connecting the paracone with the hypocone. Two small basins are formed, medially and distally to the crest that connects the paracone with the hypocone. Another oblique basin is visible mesially to the crest that connects the protocone with the parastyle.

The upper molars (Fig 13B–13D) are surrounded by a developed cingulum, which is stronger in its lingual side. The paracone and the metacone are relatively low and a preparacrista is present mesially to the paracone. The protocone is not developed as a distinct cusp, but a crista is developed near the mesial end of the preparacrista and expanded until the lingual cingulum at the plane of the metacone. The lingual side is relatively wide creating a valley lingually to the buccal cusps.

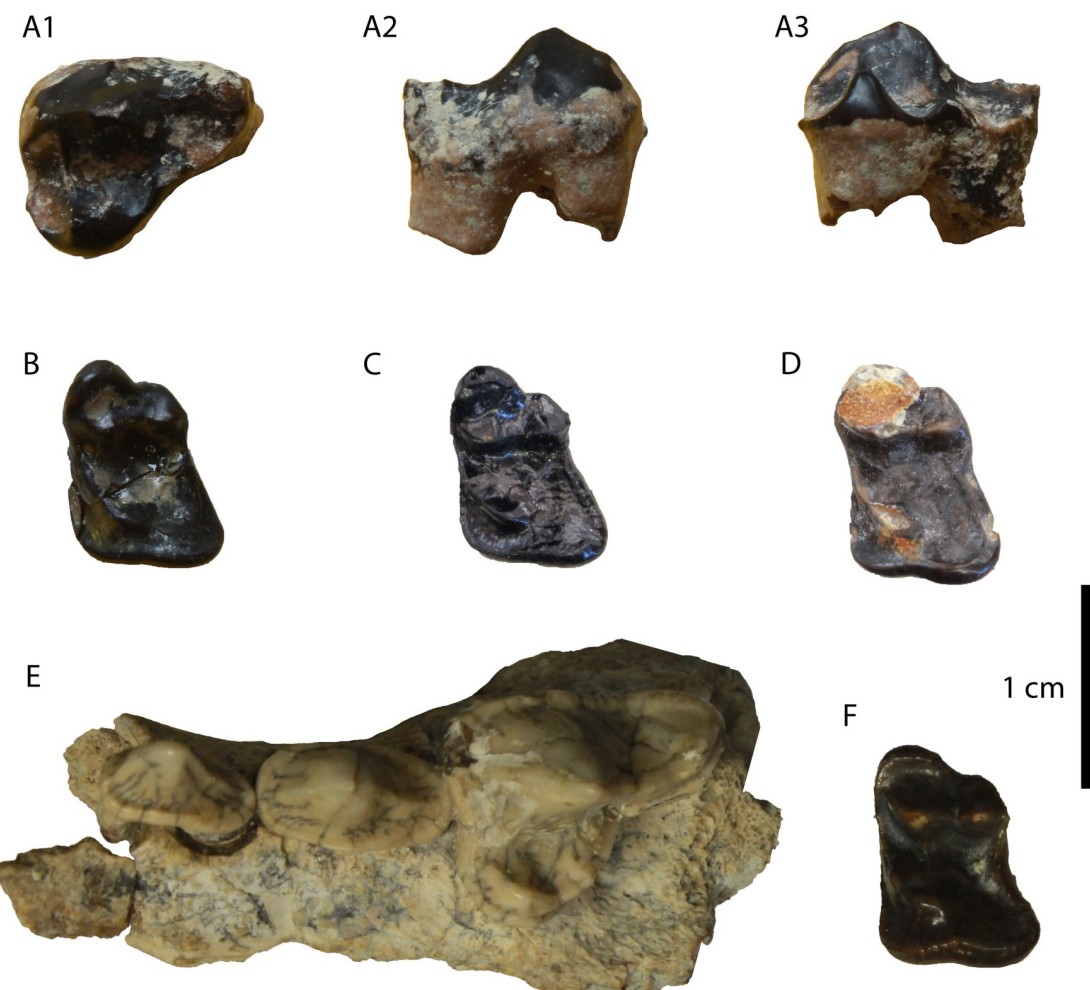

**Fig 13.** Material of *Paralutra jaegeri* from Hammerschmiede (A–D) and Steinheim (E–F): (A) GPIT/MA/10393 P4 in occlusal (A1), buccal (A2) and lingual view (A3); (B) GPIT/MA/12322 M1 in occlusal view; (C) SNSB-BSPG-2020 XCIV-5704 M1 in occlusal view; (D) SNSB-BSPG-2020 XCIV-5703 M1 in occlusal view; (E) SMNS-4082 left maxilla with P2–P4 in occlusal view, Holotype; (F) SMNS-16816 M1 in occlusal view.

**Comparison:** The P4 does not belong to the subfamily Leptarctinae, because the medial shelf is not bent mesially and because of its larger size [124]. It cannot be attributed to the lineage of *Potamotherium*, since it has a relatively weak lingual cingulum [125]. Additionally, the presence of a hypocone is another character excluding Melinae [126]. On the contrary, general morphology of the specimen fits to that of the Miocene lutrines.

The subfamily Lutrinae is represented by the genera *Sivaonyx*, *Limnonyx* Crusafont Pairó, 1950 [127], *Lartetictis* Ginsburg and Morales, 1996 [128] and *Paralutra* at the Middle/Late Miocene of Europe. The genus *Sivaonyx* is represented by the species *Sivaonyx hessicus* (Lydekker, 1890) [129] in Europe, and no upper carnassial of it has ever been published. However, other members of the genus exhibit a far more developed medial shelf that results in a square outline in the carnassial [130]. It is reasonable to suggest that the morphology of *S. hessicus* would not have been significantly different from that of the other members of its genus. Concerning the genus *Limnonyx*, [34] considered it as a member of the tribe Aonyxini, which is characterized by an enlarged medial shelf, similar to extant *Aonyx* Lesson, 1827 [131]. The genus *Lartetictis* exhibits a slenderer and more mesially situated protocone [132].

The upper molars from the herein presented material are characterized by a moderately developed lingual platform and a significant width difference between the paracone and the metacone, resembling a plesiomorphic musteline-like profile. This morphology fits both to *Paralutra jaegeri* and *Marcetia santigae* Petter, 1967 [36]. The latter species has been described by [36] from the locality of Can Llobateres 1 by a left M1, a right M1, a right P3 and a left P4 (the right M1 was erroneously misspelled as a right m1 in [36], but it is clear from the descriptions and the figures that it is an upper molar). The author noted that the upper molars were extremely similar to that of *Paralutra jaegeri*, whereas the upper carnassial had a musteline-like morphology. Additionally, [133] stated that this form was referred to as "*Paralutra* sp." in publications concerning the fauna of Can Llobateres, before the publication of [36]. However, the new name, *Marcetia santigae*, was given to this material based on the fact that these four teeth were found together. Though, it must be noted that none of the four teeth published by [36] have been found in actual association with each other as no traits of connective bone is present [36]. Therefore, since exact taphonomic data from these excavations do not exist, there is no direct evidence indicating that these teeth actually belong to the same individual or to the same species.

The species *Paralutra jaegeri* has been found to have significantly high intraspecific variability as seen in the material from Steinheim [90]. Some specimens exhibit a relatively restricted lingual platform, whereas others a considerably developed one [90]. The three specimens from Hammerschmiede also exhibit a notable variability: GPIT/MA/12322 (as well as the upper molars of *Marcetia*) resemble more the morphology of SMNS-16814 [90], because of the non-evident constriction lingually to the buccal cusps, whereas the two other Hammerschmiede molars fit better with SMNS-16816 (Fig 13C; [90]). Unfortunately, the two molars described by [36] as "*Marcetia santigae*" are either lost or destroyed (Robles, pers. comm.), so a more detailed comparison is not possible. Therefore, we suggest that based on the similar variability of the M1s of Hammerschmiede and Steinheim the Hammerschmiede specimens belong to the species *P. jaegeri* (Table 11). This leads us to hypothesize based on the M1 that *Marcetia* could be a junior synonymy of *Paralutra*. Only new findings from Can Llobateres, especially complete and associated P4 and M1, would clarify the taxonomic status of *Marcetia*.

Genus *Lartetictis* Ginsburg & Morales, 1996 [128]

**Type species:** *Lartetictis dubia* (de Blainville, 1842) [89]

**Other included species:** *Lartetictis pasalarensis* Valenciano et al., 2020 [132].

**Remarks:** The genus *Lartetictis* is typical for the Middle Miocene faunas of Europe, as it has been reported in several localities ([132] and references therein). Its subfamily status is still

**Table 11. Comparison of P4 and M1 dimensions between the Hammerschmiede specimens of *Paralutra jaegeri* and material from other localities indicating the data source.**

| Code/Locality | P4L | P4W | M1L | M1W |
|---|---|---|---|---|
| GPIT/MA/10393 | 11.1 | 9.4 | | |
| GPIT/MA/12322 | | | 6.7 | 11.0 |
| SNSB-BSPG-2020-XCIV-5704 | | | 6.5 | 9.7 |
| SNSB-BSPG-2020-XCIV-5703 | | | 6.8 | 9.8 |
| Steinheim | 10.9–12.7 11.8 (2) | 8.0–10.8 9.4 (2) | 7.6–8.1 7.9 (4) | 10.2–10.4 10.3 (4) |
| Rudabánya [65] | 10.3–12.0 11.2 (2) | 9.7–10.3 10.0 (2) | 9.3–9.6 9.5 (2) | 12.1–12.4 12.3 (2) |
| Can Llobateres "*Marcetia santigae*" [36] | 8.5 | | 6.8 | 9.9 |

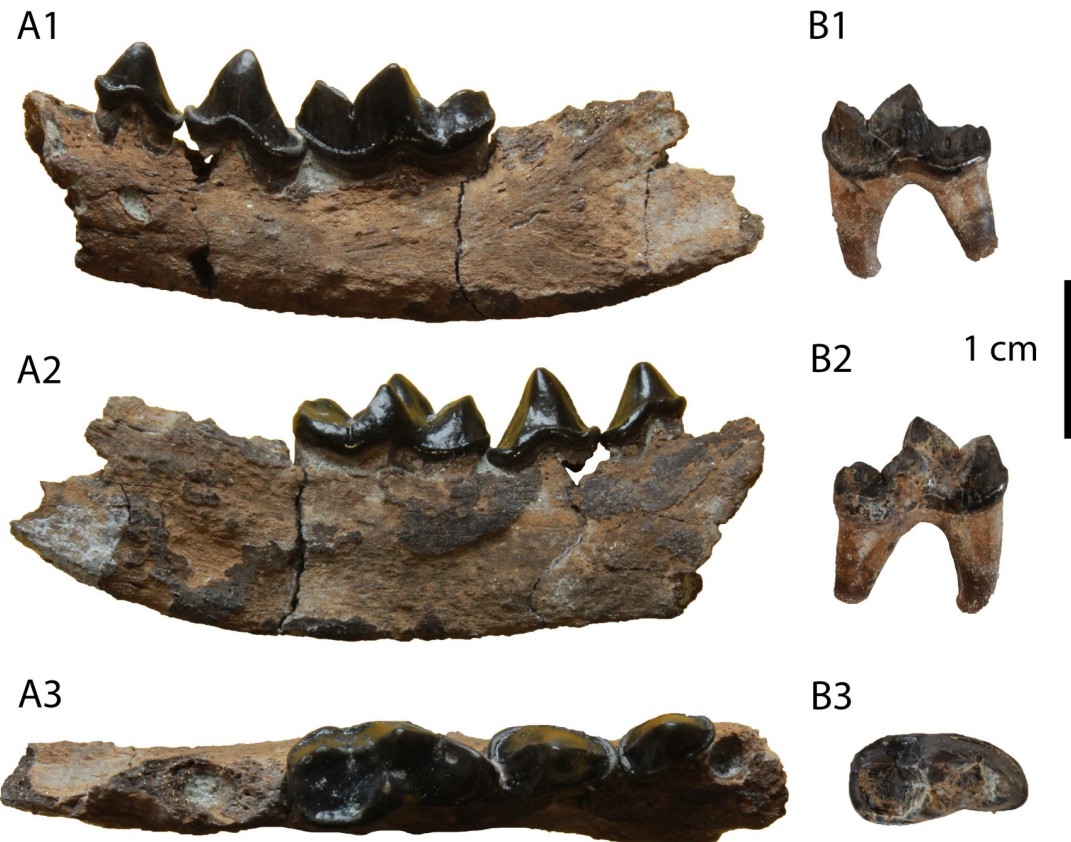

**Fig 14.** Material of cf. *Lartetictis* cf. *dubia* from Hammerschmiede: (A) GPIT/MA/17790 hemimandible and (B) GPIT/MA/13749 m1.

debatable as some authors consider it as a lutrine [132], while others as a musteline [74, 134]. However, its semi-aquatic adaptations have been widely accepted [134].

*Lartetictis* cf. *dubia* (de Blainville 1842) [89]

**Holotype:** MNHN Sa 801, right hemimandible with p2–m2.

**Type Locality:** Sansan (France).

**Referred Specimens:** HAM 4: GPIT/MA/17790, left hemimandible with p3–m1 and the alveoli of p2 and m2; SNSB-BSPG-2020-XCIV-2683, left m1.

**Description:** The specimen GPIT/MA/17790 (Fig 14A) is a left mandibular corpus with p3–m1 and the alveoli of p2 and m2. A mental foramen is present below p3 and the masseteric fossa ends at the level of m2. Both p3 and p4 are asymmetrical, they have a smooth cingulid and they don't have accessory cuspids. The fourth premolar is significantly higher and longer than the third one. The lower carnassial is long and relatively low, with the talonid covering approximately 30% of the tooth length. A relatively developed cingulid encircles the tooth, being more developed in its buccal part. There are faint signs of wear in the carnassial blade. The protoconid is the highest cuspid, separated from the paraconid by a shallow notch. The metaconid is high (approximately at the same height as the paraconid), acute and not lingually bent. The talonid valley is shallow and the only talonid cuspid is the large and conical hypoconid. The same description applies for SNSB-BSPG-2020-XCIV-2683, which is an isolated left m1 (Fig 14B).

**Comparison:** Two members of the genus *Lartetictis* have been described: the type species *L. dubia* (MN 5–MN 8 of central Europe) and *L. pasalarensis* (MN 5 of Turkey) (132)

**Table 12. Metrical comparison of the lower teeth of the different forms of *Lartetictis*.** Values in parentheses indicate measurements of the alveolus indicating the data source.

| Code/Species | p3L | p3W | p4L | p4W | m1L | m1trL | m1W | m2L | m2W |
|---|---|---|---|---|---|---|---|---|---|
| GPIT/MA/17790 | 5.9 | 3.3 | 7.7 | 4.5 | 12.2 | 8.7 | 6.2 | (4.2) | (3.5) |
| SNSB-BSPG-2020-XCIV-2683 | | | | | 10.4 | 8.1 | 5.4 | | |
| *Lartetictis dubia* [72, 132, 134] | 6.8–8.1 7.4 (6) | 3.8–4.5 4.2 (4) | 9.0–10.8 9.9 (6) | 4.7–5.6 5.1 (5) | 12.2–17.2 15.6 (13) | | 6.5–8.3 7.5 (12) | 6.9–7.2 7.1 (2) | 6.3–7.0 6.7 (2) |
| *Lartetictis pasalarensis* [132] | 7.8 | 4.4 | 9.7–9.9 9.8 (2) | 5.0–5.1 5.1 (2) | 15.2–17.0 16.0 (6) | | 7.2–8.4 7.7 (7) | 6.4–6.6 6.5 (2) | 5.9–6.0 6.0 (2) |

(Table 12). Therefore, the specimens from Hammerschmiede (together with those from Mörgen; [135]) consist of the youngest record of the genus. The two species have been differentiated on the basis of the morphology of P4, M1, m1 and m2. Consequently, only the lower carnassial can be used herein for the identification. The most distinct difference between the two species is the morphology of the m1 hypoconid, which is relatively higher and narrower in *L. pasalarensis*, whereas it is shorter and wider in *L. dubia*. The Hammerschmiede material is closer to the type species in this character, as well as in the absence of a hypoconulid and the lingual cuspulets of the m1 talonid. However, the talonid basin is relatively shallow and the trigonid cuspids are relatively high, so we prefer to refer to this specimen as *Lartetictis* cf. *dubia*.

Subfamily Leptarctinae Gazin, 1936 [136]

Genus *Trocharion* Forsyth Major, 1903 [137]

**Type species:** *Trocharion albanense* Forsyth Major, 1903 [137]

**Remarks:** The subfamily Leptarctinae includes five genera of small-sized mustelids, mainly from North America and Asia. The only genus that has been found in Europe is *Trocharion albanense*, which is a typical member of the late Early to Late Miocene faunas of central Europe, as it has been found in La Grive-Saint-Alban [137], Vieux-Collonges [52], Steinheim [90] and in several sites in the Vallès-Penedès Basin [124]. Regarding *Gaillardina*, recorded from La Grive (MN 7 and MN 8; [138]), it was classified into the Leptarctinae by [38], based on the possession of a double temporal crest. However, as noted by [139], the M1 of this taxon does not display the typically bunodont leptarctine morphology, but rather a derived mustelid condition, as shown by the expanded lingual cingulum around the protocone (a morphology more typical of the Guloninae and Melinae). On this basis, [139] considered it more likely that the double temporal crest of this taxon is an independent acquisition, so that it must be excluded from the Leptarctinae. As such, *Trocharion* remains as the only representative of the Leptarctinae in Europe [124].

*Trocharion albanense* Forsyth Major, 1903 [137]

**Holotype:** NHMUK 5307, a right hemimandible with p4–m2.

**Type Locality:** La Grive-Saint-Alban (France).

**Referred Specimens:** HAM 4: GPIT/MA/16579; partial skull; GPIT/MA/12553, right M1; SNSB-BSPG-2020-XCIV-2690, left M1. HAM 5: GPIT/MA/13462, right M1; GPIT/MA/13712, right M1; GPIT/MA/18601, right M1; GPIT/MA/18607, right M1.

**Description:** The subfamily is represented in the present material only by one partial skull and six upper molars. The partial skull (GPIT/MA/16579; Fig 15A) is part of the distal region of the braincase. It is convex dorsally and concave ventrally, indicating tha the braincase of the species was relatively globular. It is marked by two converging well-developed temporal crests that merge at their most distal point. Their medial profile is more marked than the labial one.

The molars are three-rooted, sub-trapezoidal with the lingual side being shorter than the mesial one. They have a faint cingulum and exhibit developed facets of tooth wear (especially

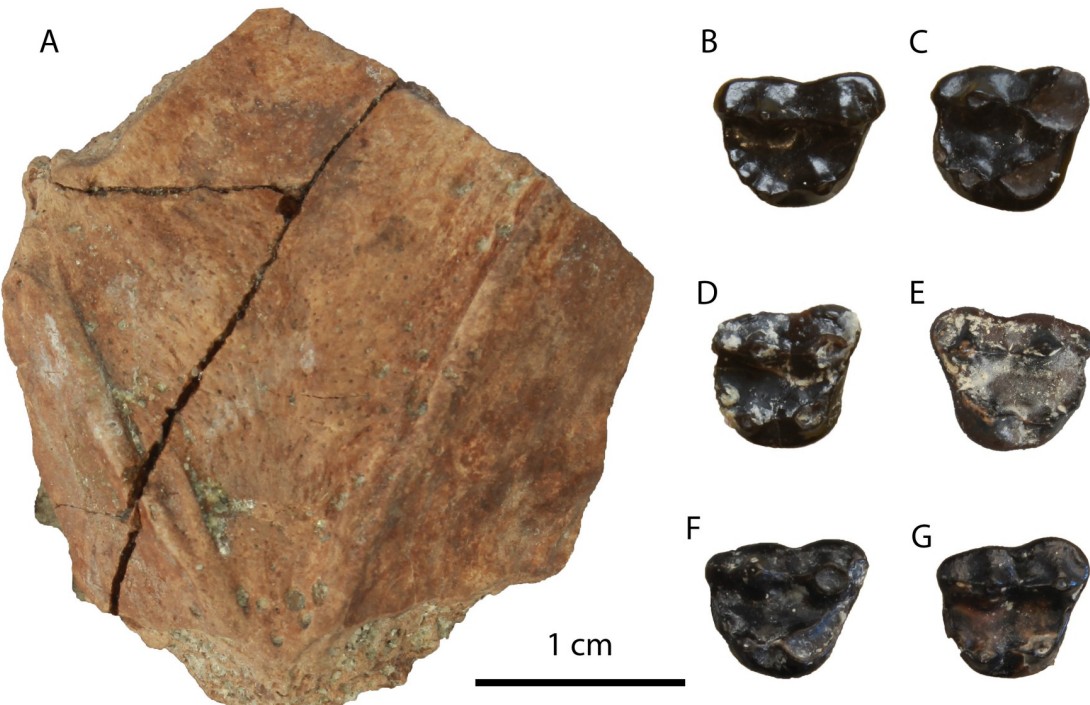

**Fig 15.** The material of *Trocharion albanense* from Hammerschmiede: (A) the partial skull GPIT/MA/16579 in dorsal view; (B–G) the M1 in occlusal view: (B) GPIT/MA/12553; (C) GPIT/MA/13462; (D) GPIT/MA/13712; (E) SNSB-BSPG-2020-XCIV-2690; (F) GPIT/MA/18601; (G) GPIT/MA/18607.

in the mesial side) that restrict the detail of the description. The paracone is the longest cusp of the tooth, followed by the shorter and higher metacone, with which it is connected with a low crista. A postmetacrista connects the metacone with the distal cingulum that hosts a small accessory cusp (metastyle). The lingual part of the tooth hosts a large protocone mesially and two consecutive tiny cusps distally. The more mesially located one probably corresponds to the hypocone, while the second can be interpreted as a hypoconule. The most unworn specimen (GPIT/MA/12553; Fig 15B) exhibits another small cuspule, distobuccally to the other two.

**Comparison:** The described partial braincase is considerably fragmentary, which makes its identification problematic. However, the two well-defined temporal crests that converge distally are an apomorphy of the leptarctines [124]. Temporal crests are present in other groups of mammals through the Miocene, such as artiodactyles and primates, but in these cases they are now combined to a globular braincase or the absence of a sagittal crest.

The only subfamily that matches the rectangular outline, the cusp structure and the small size of the discovered M1 is Leptarctinae. Most leptarctine genera (except *Leptarctus* and *Trocharion*) are characterized by upper molars that are far wider than long [124, 139, 140]. Hence, the only comparable genus to *Trocharion* is *Leptarctus*, which has been found in Asia and North America. As aforementioned, the only European representative of the subfamily in Europe is *Trocharion albanense* (e.g. [124, 141]). Additionally, despite their overall similarity, the M1 of *Leptarctus* has a relatively similar lingual and buccal mesiodistal length [141–143], while that of *Trocharion* is clearly narrower lingually [124] (Table 13). Thus, based on the temporospatial range and morphological features, the material is assigned to *Trocharion albanense*.

**Table 13. Metrical comparison of the Hammerschmiede material of *Trocharion albanense* to that from Vallès-Penedès and *Leptarctus* spp indicating the data source.**

| Species | Code | L | W | W/L |
|---|---|---|---|---|
| *Trocharion albanense* | SNSB-BSPG-2020-XCIV-2690 | 8.1 | 6.8 | 0.84 |
| | GPIT/MA/12553 | 7.8 | 6.4 | 0.82 |
| | GPIT/MA/13712 | 6.9 | 6.5 | 0.94 |
| | GPIT/MA/13462 | 7.4 | 6.7 | 0.91 |
| | GPIT/MA/18601 | 7.4 | 6.4 | 0.86 |
| | GPIT/MA/18607 | 7.0 | 6.3 | 0.86 |
| | Vallès-Penedès [124] | 5.8–7.4 6.5 (12) | 6.0–8.2 7.1 (14) | 0.82–1.00 0.92 (11) |
| *Leptarctus neimenguensis* [142] | | 8.5 | 7.5 | 0.88 |
| *Leptarctus primus* [142] | | 8.2 | 7.0 | 0.85 |

Family Mephitidae Bonaparte, 1845 [32]

Subfamily Mephitinae Bonaparte, 1845 [32]

Genus *Palaeomeles* Villalta Comella & Crusafont Pairó, 1943 [30]

**Type species:** *Palaeomeles pachecoi* Villalta Comella & Crusafont Pairó, 1943 [30]

**Remarks:** *Palaeomeles pachecoi* is the only species of the genus *Palaeomeles*. It was originally described based on two associated fragmentary maxillas from Hostalets de Pierola in Spain [30]. One year later, the same authors published a mandibular fragment with a lower carnassial and the alveolus of the m2 from another site in the same region [144]. Several years later, [145] published additional dental material and some postcranial remains of this species from the locality of Castell de Barberá (early Vallesian, MN 9). In this study, the authors provided more details about the sites where the previous material was found. They stated that the material published by [30] was found "near Can Mata de La Garriga", while the material published by [144] comes from "the vicinity of Can Vila, near the local road of Can Mata". Therefore, it is registered in sediments of the late Aragonian to early Vallesian (MN 7/8 and MN 9) of Spain from Vallès-Penedès basin and potentially from Escobosa de Calatañazor [146].

*Palaeomeles pachecoi* Villalta Comella & Crusafont Pairó, 1943 [30]

**Holotype:** ICP-IPS697, a fragmentary maxilla including the right C alveolus, right P2 alveolus, right P3 and right P4, in addition to the left P4 and left M1.

**Type Locality:** Hostalets de Pierola (Spain).

**Referred Specimens:** HAM 5: GPIT/MA/12650, right P4; GPIT/MA/09884, right M1; GPIT/MA/09926, right M1; GPIT/MA/13711, right hemimandible with p3–m2; SNSB-BSPG-2020-XCV-0032, left hemimandible with p3–m1; GPIT/MA/13749, left m1. The hemimandibles were found in close proximity and they most probably belong to the same individual.

**Description:** The upper carnassial (GPIT/MA/12650; Fig 16A) is complete with faint signs of wear in the carnassial blade. A moderately developed cingulum surrounds the whole tooth. The paracone is the largest cusp and it is connected to the metastyle through a ridge. Mesially, there is a very small parastyle. The protocone region is long and distally situated. The protocone is relatively small and a slightly larger hypocone is also present.

The upper molars do not exhibit extensive signs of wear. However, the specimen GPIT/MA/09884 (Fig 16C) is partly damaged, so the description is based mainly on GPIT/MA/09926 (Fig 16B). The molars have an oval-trapezoidal shape and a strong cingulum in their perimeter, which is especially developed in the buccal and distal parts of the tooth. All the cusps are considerably low. The largest cusp is the paracone, which is slightly higher, longer and wider than the metacone. These two cusps are connected with a low and relatively

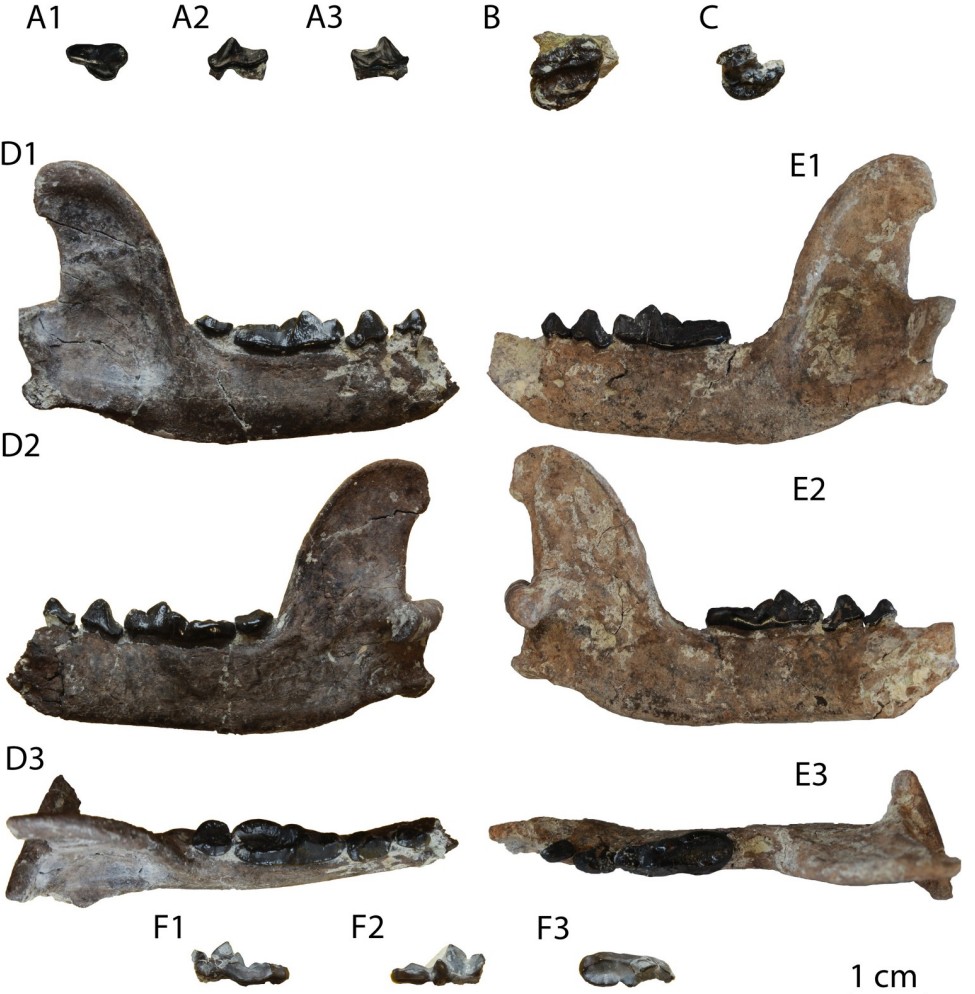

**Fig 16.** Material of *Palaeomeles pachecoi* from Hammerschmiede: (A) GPIT/MA/12650 P4; (B) GPIT/MA/09926 M1; (C) GPIT/MA/09884 M1; (D) GPIT/MA/13711 right hemimandible; (E) SNSB-BSPG-2020-XCV-0032 left hemimandible; (F) GPIT/MA/13749 m1.

continuous crista. A small mesostyle is present in the buccal cingulum. A relatively large proto-conule is also present, followed by a strong preprotocrista that ends at the plane of the meta-cone. This part is slightly worn, not enabling a detailed description. The distolingual part of the tooth is moderately enhanced, having a more robust cingulum, which is distally placed, in opposition to the mesial protocone. The surface of the tooth is marked by faint grooves.

Both hemimandibles are broken just mesially to the alveolus of p3 (Fig 16D, 16E). The mandibular body is thick and it hosts two mental foramina: one smaller and long under the distal root of p4 and one larger and circular under the distal root of p3. The mandibular corpus slowly becomes lower mesially. The masseteric fossa is deep and it reaches the level of m2's distal border. Both angular processes are broken. The mandibular condyle is considerably wide and it is situated quite dorsally to the angular process. The coronoid process is high and sigmoid-shaped with its tip placed distally, resembling that of the extant red panda (*Ailurus fulgens*). No diastemata are present.

The third and the fourth premolars have a similar simple morphology, without accessory cuspids, with a faint ridge extending disto-lingually of the main cuspid and a smooth cingulid.

They are both relatively robust, blunt and asymmetrical. However, the p4 is considerably larger than the p3. The lower carnassial has a stronger cingulid and its talonid is very long (especially in its buccal side), covering approximately 45–50% of the tooth length. It is very slightly worn in both the talonid cingulid and the carnassial blade. The buccal surface of the carnassial is rough. The trigonid cuspids are low. The protoconid is the largest one, while the metaconid and the paraconid have a relatively similar size, with the paraconid being longer and narrower. The protoconid is separated from the paraconid with a shallow notch. The metaconid is blunt and distolingually oriented. The talonid valley is long and shallow. All talonid cuspids are very small, especially the hypoconulid. Signs of wear are present between the hypoconid and the protoconid. Faint folds of the cingulid surface create tiny cuspules at its lingual part. The second molar is almost circular and the occlusal surface is not mesially bent. There is a strong cingulid that congregates several small cuspulids in its perimeter. The largest is located in its distobuccal part, and most probably corresponds to a hypoconid, whereas the protoconid is situated in its mesiobuccal region.

**Comparison:** This species is characterized by the derived features of its dentition towards a badger-like ecomorphotype. The upper carnassial has a developed, distally placed protocone region that hosts a developed hypocone. The upper molar is oval-like, much different than the rectangular molar of the intermediate fossil "badgers", such as *Promeles* (e.g. [66, 104]), *Taxodon* (e.g. [72]), *Trochictis depereti* (e.g. [147]) and *Ferinestrix* (e.g. [148]). All these taxa also have a relatively shorter m1 talonid and higher cusps in both M1 and m1 compared to *Palaeomeles*. The only genera with a comparable relative length of m1 talonid are the extinct *Melodon* and *Palaeomeles*. The genus *Melodon* exhibits some intrageneric variability, with the three species *Melodon incertum* Zdansky, 1924 [77], *Melodon major* Zdansky, 1924 [77] and *Melodon sotnikovae* (Tedford & Harington, 2003) [149] having some important differences. In comparison to *Palaeomeles*, *M. incertum* has a much more primitive M1 of rectangular shape, while *M. major* has a much more derived one, very similar to that of *Meles* (see [77]). On the other side *M. sotnikovae* has a M1 similar to that of *M. major* and an m1 with a relatively shorter talonid [149, 150]. All these species exhibit high cuspids in their m1 talonid. The only species that fits perfectly with the present specimens with an extremely derived molar morphology (enhanced lingual part of M1, very long m1 talonid and low m1 cuspids) and its moderate size is *Palaeomeles pachecoi*. The specimens of *P. pachecoi* from Hammerschmiede are the largest of this species in the fossil record (Tables 14 and 15).

Genus *Proputorius* Filhol, 1890 [26]

**Type species:** *Proputorius sansaniensis* Filhol, 1890 [26]

**Other included species:** *Proputorius pusillus* (Viret, 1951) [27]

*Proputorius sansaniensis* Filhol, 1890 [26]

**Holotype:** MNHN-Sa 776, left hemimandible with c and p3–m1.

**Type Locality:** Sansan (France).

**Referred Specimens:** HAM 1: SNSB-BSPG-1973-XIX-24, right p4; SNSB-BSPG-1973-XIX-25, right m1.

**Table 14. Metrical comparison of the two M1 of *Palaeomeles pachecoi* from HAM 5 with that of *P. pachecoi* from Spain indicating the data source.**

| Code/Locality | P4L | P4W | M1L | M1W |
|---|---|---|---|---|
| GPIT/MA/12650 | 7.4 | 4.8 | | |
| GPIT/MA/09926 | | | 9.6 | 7.7 |
| GPIT/MA/09884 | | | | (7.4) |
| Castell de Barberá & Hostalets [30, 145] | 7.5 | 5.0 | 8.0–8.6 8.3 (3) | 6.9–7.7 7.2 (3) |

**Table 15. Metrical comparison of the lower dentition of *Palaeomeles pachecoi* from HAM 5 with that of *P. pachecoi* from Spain indicating the data source.**

| Code/Locality | p3L | p3W | p4L | p4W | m1L | m1trL | m1W | m2L | m2W |
|---|---|---|---|---|---|---|---|---|---|
| GPIT/MA/13711 | 4.2 | 2.6 | 5.4 | 3.8 | 14.2 | 8.4 | 5.7 | 5.0 | 4.7 |
| BSPG 2020 XCV-0032 | 4.1 | 2.5 | 5.4 | 3.6 | 14.4 | 8.7 | 5.7 | | |
| GPIT/MA/13749 | | | | | 12.7 | 6.9 | 4.6 | | |
| Castell de Barberá & Hostalets [30] | 4.1 | 2.3 | 5.2 | 3.0 | 10.8–11.0 11.0 (3) | | 4.0–4.5 4.3 (3) | 3.8–4.5 4.3 (3) | 2.5–3.9 3.4 (3) |

**Description:** The specimen SNSB-BSPG-1973-XIX-24 is a right p4 with no signs of wear (Fig 17A). It is asymmetrical with the distal part being longer and wider than the mesial part. It is unicuspid with a faintly rough enamel surface. The mesial and distal ridges of the main cuspid host a cristid. It is two-rooted and with a faint cingulid.

The lower carnassial (SNSB-BSPG-1973-XIX-25; Fig 17B) exhibits distinct signs of wear in the carnassial blade. The protoconid is the highest and longest cuspid. It is separated from the paraconid by a deep notch. The metaconid is high (almost as high as the paraconid) and well-individualized from the protoconid. A small valley is formed between the metaconid and the paraconid. The talonid is long with a deep valley. The hypoconid is enlarged (height, width and length) and the rest of the talonid border hosts small cuspulids. The talonid is relatively developed in the mesiobuccal part of the tooth, but more reduced in the rest of the dental border.

**Comparison:** The known dentition sample of this species from Sansan is quite complete. [4] described some teeth of *P. sansaniensis* from Hammerschmiede including the I3, P4, p3, m1 and m2. The isolated incisor does not exhibit any diagnostic features, so it is not attributed to any group herein. The P4 clearly differs to the P4 of *P. sansaniensis* in being more elongated and having an enlarged and mesiolingually projected protocone to that of the specimen from Sansan Sa 15668 [72], which is more reduced and closer to the paracone. As we mentioned

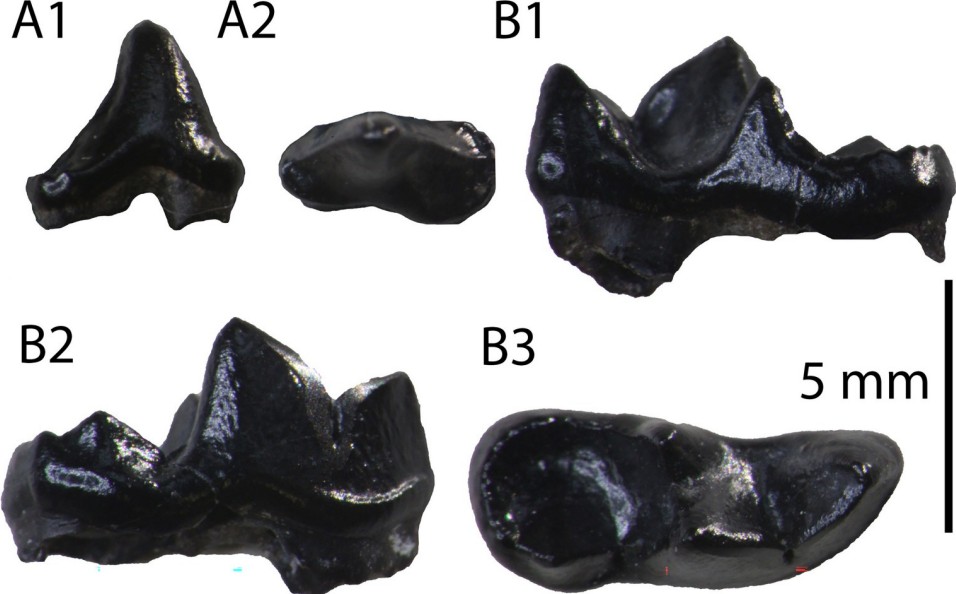

**Fig 17.** Material of *Proputorius sansaniensis* from Hammerschmiede: (A) SNSB-BSPG-1973-XIX-24: right p4 in buccal (A1) and occlusal (A2) views; (B) SNSB-BSPG-1973-XIX-25: right m1 in lingual (B1), buccal (B2) and occlusal (B3) views.

**Table 16. Metrical comparison of the upper and lower dentition of *Proputorius sansaniensis* and *Proputorius pusillus* from HAM 1 (the abbreviation SNSB-BSPG-was removed to save space) with other material of the genus indicating the data source.**

| Species | Code/Locality | p3L | p3W | p4L | p4W | m1L | m1W | m2L | m2W |
|---|---|---|---|---|---|---|---|---|---|
| *P. sansaniensis* | 1973-XIX-24 | | | 4.3 | 2.2 | | | | |
| | 1973-XIX-25 | | | | | 8.3 | 3.6 | | |
| *P. pusillus* | 1973-XIX-30 | 1.9 | 0.9 | | | | | | |
| | 1973-XIX-31 | 2.1 | 1.0 | | | | | | |
| | 1973-XIX-32 | | | | | 4.2 | 1.6 | | |
| | 1973-XIX-33 | | | | | 4.5 | 1.7 | | |
| *P. sansaniensis* | Sansan [72] | 3.7–4.3 3.9 (10) | 2.0–2.3 2.1 (9) | 4.1–5.3 4.9 (14) | 2.2–2.9 2.6 (14) | 8.0–10.4 9.4 (18) | 3.6–4.7 4.1 (18) | 3.4–4.0 3.7 (3) | 2.7–3.2 3.0 (3) |
| *P. pusillus* | Vieux-Collonges [52] | 2.3–2.3 2.3 (3) | 1.2–1.4 1.3 (3) | 2.7–2.9 2.8 (6) | 1.4–1.5 1.5 (6) | 4.4–5.4 4.9 (15) | 1.9–2.4 2.1 (15) | 1.9–2.3 2.1 (6) | 1.5–1.7 1.6 (6) |

above, this tooth is herein re-classified as *Circamustela hartmanni* n. sp. The m2 of *Proputorius* spp. includes only one cuspid (protoconid) at its central-buccal side [52, 72]. On the contrary, the m2 described by [4] has two larger and one smaller cuspid in its border. As discussed below, this tooth is here attributed to a viverrid. The rest of the teeth are similar to that of *P. sansaniensis* from Sansan (Table 16).

*Proputorius pusillus* (Viret, 1951) [27]

**Lectotype:** [27] did not define a holotype. However, the first specimen mentioned by [27] is MHNL-Lg 1256, which is a right hemimandible with p3–m1. This is also the most complete specimen published by [27]. Therefore, it is proposed as the lectotype of the species by present designation under the provisions of ICZN Art. 74.

**Type Locality:** La Grive (France).

**Referred Specimens:** HAM 1: SNSB-BSPG-1973-XIX-30, right p3; SNSB-BSPG-1973-XIX-31, right p3; SNSB-BSPG-1973-XIX-32, right m1; SNSB-BSPG-1973-XIX-33, right m1.

**Description:** The p3 are of typical mephitid morphology. They are triangular in buccal and occlusal, unicuspid and distally enlarged with a complete cingulid rounded the tooth view (Fig 18A). The lower carnassials (Fig 18B) have an open m1 trigonid with very tall protoconid, and a metaconid almost reaches the height of the paraconid. The talonid shows some variability, being slenderer with a lower entocristid in SNSB-BSPG-1973-XIX-32. Both talonids are rhomboidal in occlusal view. The hypoconid is buccally located, relatively tall and crested-like, with a bevelled wall in its lingual part.

**Comparison:** This material was reported in [4], but it was never described. Both lower premolars are similar to the ones of *P. pusillus* in terms of morphology and dimensions [52] (Table 16). Among the two m1s, one is comparable to *P. pusillus* (SNSB-BSPG-1973-XIX-33) based on the specimens from La Grive and Vieux-Collonges, and the other one (SNSB-BSPG-1973-XIX-32) is close to "*Martes*" *jaegeri* in having similar metaconid and talonid morphology but slightly smaller and with a less marked cingulid. The sample of *P. putorius* from the type locality shows also some variability in the width of the m1 talonid, with some forms being more robust than others [27]. Consequently, until more material is available from Hammerschmiede we classify the smallest musteloid taxon as *P. pusillus*, with the exception of the P4 SNSB-BSPG-1973-XIX-34. This material fits to the smallest specimens of the species from La Grive [27], Vieux-Collonges [52] and Sandelzhausen [73].

Family Ailuridae Gray, 1843 [33]

Subfamily Simocyoninae Dawkins, 1868 [151]

Genus *Alopecocyon* Camp & Vanderhoof, 1940 [152]

**Type species:** *Alopecocyon goeriachensis* (Toula, 1884) [153]

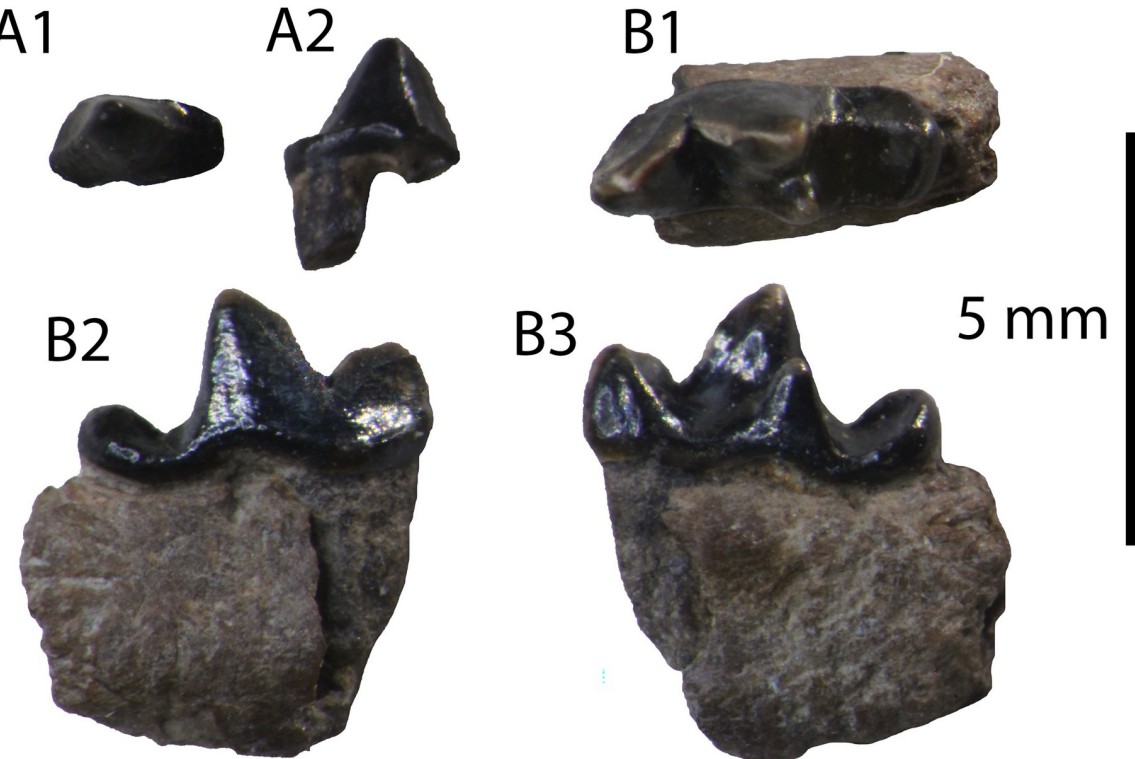

**Fig 18.** Material of *Proputorius pusillus* from Hammerschmiede: (A) SNSB-BSPG-1973-XIX-31, left p3 in (A1) lingual and (A2) occlusal views; (B) SNSB-BSPG-1973-XIX-32, right m1 in (B1) occlusal (B2) buccal and (B3) lingual views.

**Questionable other included species:** *Alopecocyon getti* Mein, 1958 [52]; *Alopecocyon leardi* Stock, 1947 [154].

**Remarks:** The only well-known species of this genus is the type species, *Alopecocyon goeriachensis* (Toula, 1884) [153]. This form has been described from several localities in the Middle Miocene of Europe, such as Göriach [153], Sansan [72, 85], La Grive-Saint-Alban [27], but also from the locality Duolebulejin in China [155]. Some material from Vieux-Collonges has been considered to be a different form, *Alopecocyon getti* Mein, 1958 [52], but the taxonomic position of this species has been questioned by some scholars [156]. The species "*Alopecocyon*" *leardi* (Stock, 1947) [154] was originally described as a species of *Actiocyon* Stock, 1947 [154], but it was later transferred into *Alopecocyon* (Webb, 1969) [157]. However, once again, this attribution is not considered undoubtful [156]. The youngest record of *Alopecocyon* is from Rudabánya [65] as "*Viretius* sp.", but this form is far larger than *A. goeriachensis* (Table 17; Fig 21). Thus, the herein described molar consists of the youngest report of *A. goeriachensis* in the fossil record.

*Alopecocyon goeriachensis* (Toula, 1884) [153]

**Holotype:** NHMW 470/1963, an assemblage of specimens (possibly from the same individual; [27]) that includes right C, P4, M1 and M2, fragmentary left p4 and m1 and fragmentary right p4 and m1.

**Type Locality:** Göriach (Austria).

**Referred Specimens:** HAM 5: SNSB-BSPG-2020 XCV-382, right M1.

**Description:** The upper molar (Fig 19) is complete with no signs of wear. Its outline is triangular and it has a relatively strong cingulum on its buccal and mesial side. The lingual

**Table 17. Comparison of M1 and m2 between the specimens from Hammerschmiede and other Miocene ailurids from Europe indicating the data source.**

| Species | M1L | M1W | M1W/ M1L | m2L | m2W | m2W/ m2L |
|---|---|---|---|---|---|---|
| *Alopecocyon goeriachensis* SNSB-BSPG-2020 XCV-382 | 8.1 | 10.1 | 124% | | | |
| Simocyoninae indet. (SNSB-BSPG-2020 XCIV-5705) | | | | 10.3 | 5.7 | 55% |
| *Alopecocyon goeriachensis* Göriach [168], Oppeln [168], Schlieren-Uetikon [168], La Grive-Saint-Alban [82], Sansan [72], Leoben [72] | 8.2–9.3 8.8 (5) | 9.5–10.5 9.9 (4) | 107–118% 111% (4) | 6.3–8.0 7.0 (3) | 4.0–5.0 4.1 (3) | 57%–63% 59% (3) |
| *Protursus simpsoni* Can Llobateres [159], Rudabánya [65] | 11.8–11.9 11.9 (2) | 12.9–13.1 13.0 (2) | 109%–110% 110% (2) | 11.8–12.4 12.1 (2) | 6.8–6.9 6.9 (2) | 55–58% 57% (2) |
| *Magerictis imperialensis* Madrid [167] | | | | 12.1 | 5.8 | 48% |
| *Simocyon diaphorus* Rudabánya [65], Eppelsheim [169] | 14.8 | 18.8 | 127% | 14.2 | 7.9 | 56% |
| *Simocyon batalleri* Batallones-1 [170] | 15.7–17.0 16.4 (4) | 18.6–20.0 19.3 (4) | 115%–120% 117% (4) | 15.9–16.0 16.0 (3) | 7.6–8.4 8.1 (3) | 48%–53% 51% (3) |

border of the tooth also forms a robust wall, which is bent distally. The cusps are relatively low. The paracone is the highest cusp, being slightly longer than the metacone. These two cusps are connected at their bases through their crests. A crest starts from the paracone and it ends just before the buccal crest of the protocone. A minute metaconule is present at the distal part of the tooth. No sign of a paraconule or any other accessory cusp is present. However, the lingual border of the tooth is marked by faint notches, creating very small cuspules in its occlusal surface.

**Comparison:** The fossil record of the ailurids includes several different forms during the Miocene of Europe. [156] made a comprehensive review of the fossil representatives of Ailuridae. The smallest genus (and the most basal after *Amphictis* Pomel, 1853 [158]) is *Alopecocyon*, which was erected for the species *Alopecocyon goeriachensis* from Göriach by [153]. This taxon is attributed to the subfamily Simocyoninae, which also includes the genera *Protursus* Crusafont Pairó & Kurtén, 1976 [159] (including *Protursus simpsoni* Crusafont Pairó & Kurtén, 1976 [159]) and *Simocyon* Wagner, 1858 [160] [including the smaller-sized *Simocyon diaphorus* (Kaup, 1832) [161] and *Simocyon batalleri* Viret, 1929 [162], as well as the larger-sized *Simocyon primigenius* (Roth & Wagner, 1854) [163] and *Simocyon hungaricus* (Kadic & Kretzoi, 1927) [164]]. This subfamily includes forms with relatively hypercarnivorous adaptations (reaching their climax in *S. primigenius*). The subfamily Ailurinae Gray, 1843 [33], characterized by hypocarnivorous adaptations, includes the extant genus *Ailurus* Cuvier, 1825 [165], as well as the fossil genera *Parailurus* Schlosser, 1899 [166] (morphologically close to *Ailurus*; Pliocene), *Pristinailurus* Wallace & Wang, 2004 [150] (latest Miocene/Pliocene of the USA) and *Magerictis* Ginsburg et al., 1997 [167] (including *Magerictis imperialensis* Ginsburg et al., 1997 [167] from the Middle Miocene of Spain).

The herein described specimen is relatively small in size and has a high M1W/M1L ratio (Table 17). These dimensions only fit to *A. goeriachensis* from La Grive-Saint-Alban. The species *Magerictis imperialensis* is known exclusively from an m2. However, based on its dimensions, the size of its M1 is expected to be similar or larger to the M1 described as "Simocyoninae indet." below (Figs 20 and 21). Therefore, it is also considerably larger than SNSB-BSPG-2020 XCV-382.

[65] published two upper molars from Rudabánya as "*Viretius* sp." and a partial hemimandible as *Ursavus brevirhinus*. However, these three specimens are herein considered to be of different taxonomic status. The upper molars are considerably larger than those of *A. goeriachensis* (Table 17; Fig 20) and they also exhibit some morphological differences, such as

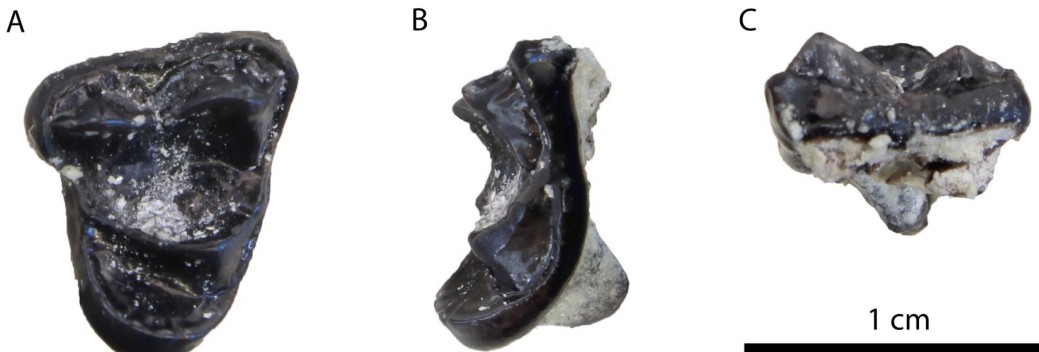

**Fig 19.** The upper first molar of *Alopecocyon goeriachensis* from Hammerschmiede (SNSB-BSPG-2020 XCV-382) in (A) occlusal, mesial (B) and buccal view (C).

the less evident constriction of the tooth at the lingual base of the buccal cusps, the deeper valley between the buccal and the lingual cusps, the more developed metaconule, the more restricted region between the protocone and the lingual cingulum, as well as the more developed cingulum around the tooth. The hemimandible from Rudabánya shares several similarities with the ailurids, such as the absence of m3, the high and separated paraconid and metaconid in m2, the simple talonid in m2 and the high m2L in relation to m2W that differentiate it from the ursids. In particular, the m2W/m2L ratio in the specimen is 55%, whereas the known range for the ursid genus is 59–67% [171]. Judging from the size (Table 17; Figs 20 & 22) and morphology, the mandible and upper molars from Rudabánya can be attributed to the species *Protursus simpsoni*. The preserved m2 (RUD/1989/142) is almost identical to the one from Can Llobateres, with the exception of stronger cingulum crests being present in the specimen from Rudabánya. Additionally, the age of the two localities is similar. Therefore, it is

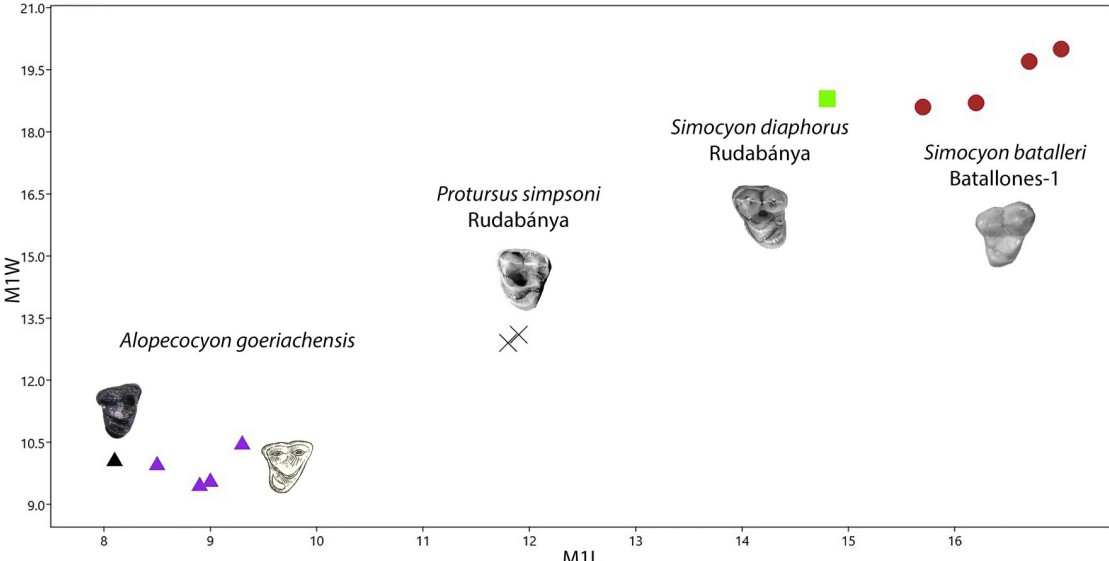

**Fig 20. Scatter plot comparing the M1 dimensions of Middle and Late Miocene ailurids of Europe: black triangle =** *Alopecocyon goeriachensis* **from Hammerschmiede; violet triangle =** *Alopecocyon goeriachensis* **from other localities [82, 168], black X =** *Protursus simpsoni* **from Rudabánya [65], green square =** *Simocyon diaphorus* **from Rudabánya [65] and brown dots =** *Simocyon batalleri* **from Batallones-1 [170].**

herein suggested that this specimen belongs to *P. simpsoni*. Based on size, distinct morphology, correlated ecomorphological trends and taxonomic parsimony, we suggest that the two upper molars from Rudabánya also belong to this form.

In this new scheme, the Hammerschmiede molar differs from the Rudabánya in all discussed traits. The morphological characteristics typical for the genus *Alopecocyon* are evident in the present specimen, such as the relatively narrow and bent lingual part of the tooth, its short length in relation to its width, the absence of a cavity between the buccal and lingual cusps, the absence of accessory cuspules, the relatively small buccal cusps, the small metaconule and the moderately developed cingulum.

Simocyoninae indet.

**Referred Specimens:** HAM 4: SNSB-BSPG-2020 XCIV-5705, right m2.

**Description:** The specimen SNSB-BSPG-2020 XCIV-5705 is a complete right m2 with no sings of wear (Fig 21). It is two-rooted, with the distal root being more elongated and thinner than the mesial one. The talonid is very large, covering approximately 60% of the tooth's length. The overall shape of the tooth outline is bean-like with the buccal side being concave and the lingual one convex, while the tooth is most wide at the level of the protoconid-metaconid. A faint rounded cingulid can be seen in the distal part of the tooth. The trigonid hosts two large cuspids: the protoconid and the metaconid. Both of them are blunt, stemming from the buccal and lingual border of the tooth respectively and they are placed 1–2 mm from the corner of the tooth. They are connected with a transverse cristid that forms a small notch in its middle point. The protoconid is slightly higher and vertical, whereas the metaconid is lower and lingually inclined. The tooth border mesially to these cuspids is marked by four notches that create small cuspulids in the faint cingulid. The largest of them, placed at the buccal side of the tooth, can be tentatively interpreted as a small paraconid. The talonid valley is relatively shallow and long. Distally to the metaconid there is a cuspid just buccally to the lingual border of the tooth that corresponds to a small hypoconid. Again, four notches mark the distal part of the talonid creating cuspulids. The largest of them, at the distolingual part of the tooth, can be interpreted as a hypoconulid.

**Comparison:** The m2 of *A. goeriachensis* is considerably smaller than the specimen found in Hammerschmiede (Table 17; Figs 20 and 22). [85] noted that the second lower molar of *A. goeriachensis* has a developed paraconid, hypoconid, hypoconulid and entoconid. These structures are only faintly seen in SNSB-BSPG-2020 XCIV-5705 as intermediate regions between the cingulid's notches. Additionally, the distal part of that tooth in *A. goeriachensis* is more bent lingually than in the Hammerschmiede molar.

The lower second molar of *Magerictis* is slightly larger than the herein presented respective tooth and has a lower m2W/m2L ratio (Table 17; Fig 22). Additionally, this tooth differs significantly from SNSB-BSPG-2020 XCIV-5705 in several morphological traits: the outline of the tooth is not oblique, there are no connecting cristae between the metaconid and the

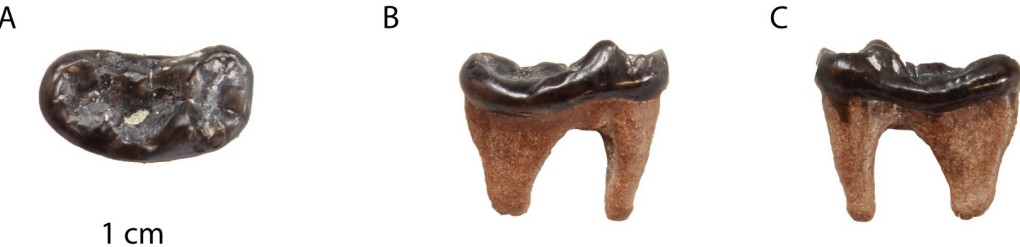

**Fig 21.** The right m2 of Simocyoninae indet. from Hammerschmiede (SNSB-BSPG-2020 XCIV-5705) in: (A) occlusal, (B) buccal and (C) lingual view.

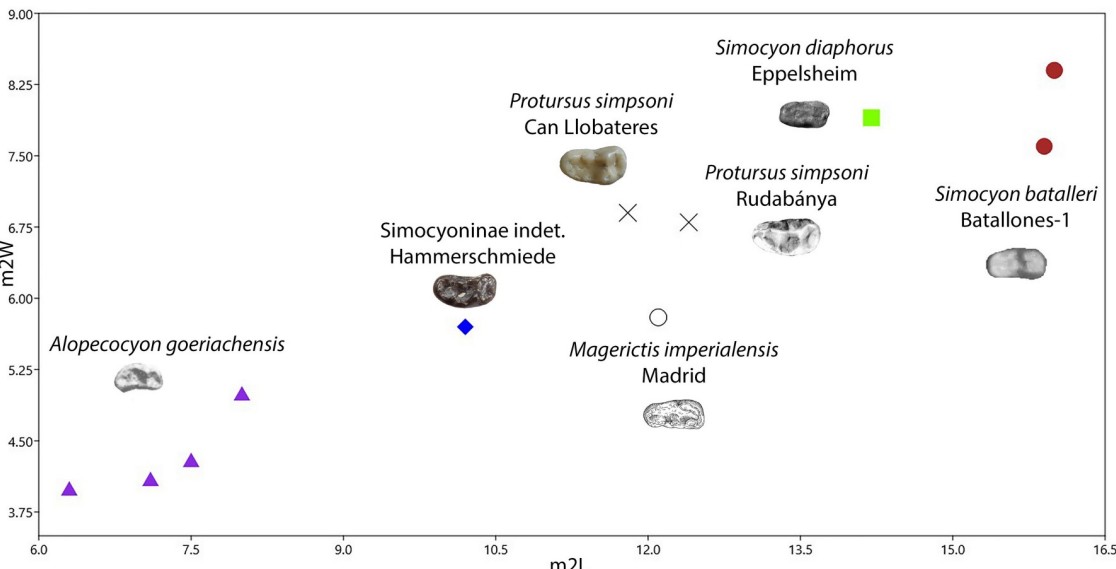

**Fig 22. Scatter plot comparing the m2 dimensions of Middle and Late Miocene ailurids of Europe:** violet triangles = *Alopecocyon goeriachensis* [27, 72, 168], blue rhombus = Simocyoninae indet. from Hammerschmiede (SNSB-BSPG-2020 XCIV-5705), black X = *Protursus simpsoni* from Can Llobateres [159] and Rudabánya [65], black circle = *Magerictis imperialensis* from Madrid [167], green square = *Simocyon diaphorus* from Eppelsheim [169] and brown dots = *Simocyon batalleri* from Batallones-1 [170].

protoconid, the metaconid is more mesially positioned, the hypoconid is larger, the paraconid is considerably developed and several additional cuspulids can be seen in the distal part of the talonid [167].

The dimensions of the specimen from Hammerschmiede are significantly lower than that of *S. diaphorus* and *S. batalleri* (Table 17; Fig 22). *Simocyon diaphorus* exhibits a similar m2W/m2L ratio, whereas the values for *S. batalleri* are slightly lower (Table 17). The valley mesially to the metaconid-protoconid plane in the m2 is shallower and less marked in *S. batalleri* [170], whereas in *S. diaphorus* it is almost absent [169]. In both species no additional cuspids or cristids are seen creating a simpler overall morphology of the teeth than that of SNSB-BSPG-2020 XCIV-5705 [169, 170].

The genus *Protursus* is known from one m2 from the locality of Can Llobateres [159] and, as discussed above, from another mandibular specimen in Rudabánya. Both specimens are larger than SNSB-BSPG-2020 XCIV-5705 (Table 17; Fig 22). Furthermore, there are some morphological differences between these two specimens and the molar from Hammerschmiede, such as the less bent talonid, the more restricted valleys (mesially to the protoconid-metaconid and the talonid valley), the smaller cuspids (especially the protoconid) and the more bulbous (and less cristrid-like) border of the talonid.

Consequently, the taxonomic status of this form remains unclear. Its size is intermediate between *Alopecocyon* and *Protursus*, whereas its morphology is distinct from both genera. Given the aforementioned presence of a typical M1 of *A. goeriachensis* in Hammerschmiede (even though in a different layer), it is possible that this m2 is an abnormal specimen that follows the size increase tendency of the species through the Aragonian [168]. On the other hand, the morphological differences are important, in order to be interpreted through intraspecific variability. It is possible that this specimen corresponds to a primitive form of the Vallesian clade of *Protursus*. However, until further material comes to light, it is preferred to refer to it as Simocyoninae indet.

Family indet.

Subfamily Potamotheriinae Willemsen, 1992 [34]

Genus *Potamotherium* Geoffroy Saint-Hilaire, 1833 [172]

**Type species:** *Potamotherium valletoni* Geoffroy Saint-Hilaire, 1833 [172]

**Other included species:** *Potamotherium miocenicum* Peters, 1868 [173].

**Remarks:** This genus is known from the Late Oligocene to the Middle Miocene of Europe [125]. During the past decades it has been considered as member of the *Semantor* lineage that links the morphology of otters and pinnipeds (e.g. [38]). Different approaches consider this group as a distinct subfamily of Mustelidae (as Potamotheriinae e.g. in [38]), as part of the subfamily Lutrinae (e.g. [125, 174]), as part of the subfamily Oligobuninae [175], as a distinct subfamily of Phocidae [176] or as a sister group of Phocidae [177]. Unfortunately, very few specimens of this group have been described yet, limiting our capability to reconstruct solidly its phylogenetic affinities. Recent studies support the distinction of this lineage from the family Mustelidae, suggesting a basal connection to the pinnipeds [178, 179]. The present specimen is tentatively considered as the youngest report of the genus up to date.

*Potamotherium* sp.

**Referred Specimens:** HAM 4: SNSB-BSPG-2020-XCIV-3551, right P3. HAM 5: GPIT/MA/ 10505, left M1.

**Description:** The right P3 (SNSB-BSPG-2020-XCIV-3551; Fig 23A) is elongated, but distally broad. A well-developed distal accessory cuspid is present, as well as one small cusp in the mesial and distal borders of the tooth. There are signs of wear in both the main and distal accessory cusps. The available M1 (GPIT/MA/10505; Fig 23B) is broken in its mesiobuccal part, showing no signs of wear and a moderately developed cingulum. The specimen was erroneously added in the first table of [22]. It is relatively narrow and the lingual part is bent occlusally. The buccal border of the tooth is considerably more extended at the level of the paracone (parastylar area) than at the level of the metacone. The remaining buccal part of the tooth includes the narrow and moderately low paracone and metacone. There are no signs of a metastylid. The valley between the buccal cusps and the lingual cingulum does not host any cusps or crests. Only an additional cusp is present at the mesio-lingual part of the cingulum. A small pit is present lingually to the metacone.

**Comparison:** The P3 is attributed to the genus *Potamotherium*, based on the characteristic distal enlargement of the dental base. As demonstrated in Table 18, this specimen is slightly larger than the known P3 of both *Potamotherium* species. Additionally, the accessory cusps are slightly more developed in the Hammerschmiede specimen.

The absence of any M1 lingual cusps, other than the one present at the lingual cingulum is a characteristic of the Potamotheriinae and the genus *Potamotherium* in particular, which is the dominant genus of the subfamily in Europe during the Miocene. The subfamily is also represented by the genus *Semantor* Orlov, 1931 [184], which is however, far larger, and exclusively known by postcranial remains [184–186].

The genus *Potamotherium* is an aquatic or semi-aquatic carnivoran that has been described by two species: the type species *P. valletoni*, which has been found in the Late Oligocene and Early Miocene of Europe, and *P. miocenicum* from the Early and Middle Miocene of Europe [38, 134]. Additionally, it has been cited in North America as *Potamotherium* sp. in sediments from the beginning of the Hemingfordian (Early Miocene, circa 18–19 Ma) [187, 188]. [182] differentiated *P. miocenicum* from *P. valletoni*, stating (among other characteristics) that the former was larger, more robust, and it has a wider M1 with a more conical (and less crest-like) protocone. Fig 23 illustrates that there is considerable morphological variability between the members of this genus, even inside the same species. The metrical comparison at Table 18 demonstrates that indeed *P. miocenicum* is slightly larger than *P. valletoni*, but no difference

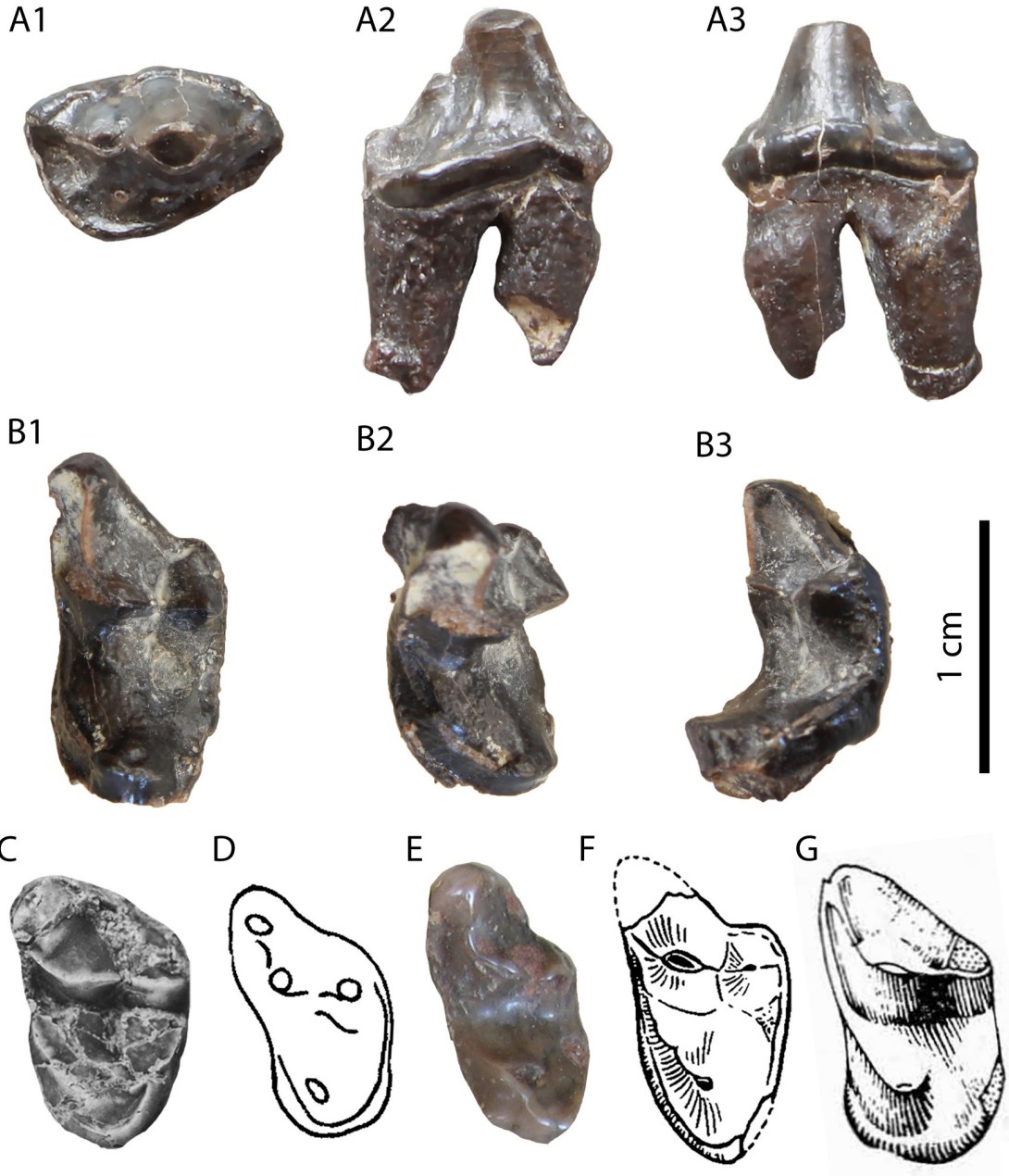

**Fig 23.** The material of *Potamotherium* sp. from Hammerschmiede (A, B), in comparison to specimens of Potamotherium valletoni (C–E) and Potamotherium miocenicum (F–G) from other localities: (A) right P3 from Hammerschmiede (SNSB-BSPG-2020-XCIV-3551); (B) left M1 from Hammerschmiede (GPIT/MA/10505); (C)left M1 from Enspel (Naturhistorisches Museum Mainz—5009/1a; [180]); (D) left M1 from Allier (British Museum of Natural History, London—M 7651; [125]); (E) right M1 of a skull from Allier (flipped; SNSB-BSPG-1885-I-13, [181]); (F) right M1 from Vordersdorf (flipped; Naturhistorisches Museum Wien– 126; [182]); (G) left M1 from Baigneaux-en-Beauce (Bale S.O. 5991; [183]).

can be seen in the relative width. The specimen GPIT/MA/10505 is even larger than the known molars of *P. miocenicum* and it exhibits a higher W/L ratio. Additionally, the protocone is not present in the Hammerschmiede specimen. Even if this lingual cuspule is considered as a protocone, then it is even less developed than in *P. valletoni*. Therefore, the studied molar differs from that of both *P. valletoni* and *P. miocenicum* in its larger size, higher W/L ratio and

**Table 18. Metrical comparison of the *Potamotherium* P3 and M1 from Hammerschmiede to other published specimens for the genus indicating the data source.**

| Species | Locality | P3L | P3W | M1L | M1W | M1W/L |
|---|---|---|---|---|---|---|
| *Potamotherium* sp. (SNSB-BSPG-2020-XCIV-3551) | Hammerschmiede | 9.6 | 6.7 | | | |
| *Potamotherium* sp. (GPIT/MA/10505) | | | | (7.8) | 12.8 | (61%) |
| *P. miocenicum* | Voitsberg [182] | 7.4 | 4.6 | 6.6 | >12 | <55% |
| | Baigneaux-en-Beauce [183] | | | 6.4 | 12.2 | 52% |
| *P. valletoni* | Allier [125] | 7.9 | 4.3 | 5.6 | 10.2 | 55% |
| | Wiesbaden-Amöneburg [190] | 7.3 | 5.2 | 5.8 | 11.1 | 52% |

the more reduced lingual cusps. These morphological differences, together with the considerable age difference between the last appearances of *P. miocenicum* (Devinska Nova Ves Sandberg; approximately 13 Ma; MN 6; [189]) and Hammerschmiede, can be interpreted potentially as indicators of a distinct form being present in central Europe during the early Late Miocene. However, due to the scarcity of the material, additional fossils are needed to test this hypothesis about the status of this enigmatic form.

Suborder Feliformia Kretzoi, 1945 [191]

Family Viverridae Gray, 1821 [35]

Genus *Semigenetta* Helbing, 1927 [192]

**Type species:** *Semigenetta sansaniensis* (Lartet, 1851) [28]

**Other included species:** *Semigenetta elegans* Dehm, 1950 [46]; *Semigenetta cadeoti* Roman & Viret, 1934 [119]; *Semigenetta laugnacensis* (de Bonis, 1973) [193]; *Semigenetta grandis* Crusafont Pairó & Golpe Posse, 1981 [29].

**Remarks:** This genus was recently reviewed by [21]. It exhibits a relatively uniform morphology throughout the Miocene (MN 1/2 to MN 10) and the species differentiations are based mainly on size.

*Semigenetta sansaniensis* (Lartet, 1851) [28]

**Lectotype:** MNHN Sa 808, left hemimandible with p3–m1.

**Type Locality:** Sansan (France).

**Referred New Specimens:** HAM 1: SNSB-BSPG-1973-XIX-26, right m2. HAM 4: SNSB-BSPG-2020 XCIV-4024, right hemimandible with p3–m1 and the alveoli of p1, p2 and m2; GPIT/MA/13751, right p4; SNSB-BSPG-2020 XCIV-3614, right m1; SNSB-BSPG-2020 XCIV-5706, right m1. HAM 5: GPIT/MA/13452, right M1; GPIT/MA/12130, left p4; GPIT/MA/10298, right calcaneum.

**Description:** The right M1 (GPIT/MA/13452; Fig 24A) is complete with no signs of wear. It outline is sub-triangular and relatively narrow. The mesial part is relatively wider, especially at the part buccally to the paracone. The paracone is considerably higher than the metacone, but both cusps are pyramidal and pointy. There are no signs of other cusps. A faint cingulum surrounds the tooth and creats an elevation is ints lingual border that resembles a cusp.

The right hemimandible is broken at the alveolus of the canine and it preserves p3–m1 and the alveoli of p1, p2 and m2 (Fig 24D). The hemimandible is moderately robust, exhibiting a subangular lobe below m1 and m2 and two mental foramens: one below p1 and one below p2. The masseteric fossa is deep, reaching the plane of m2. The third premolar is high (higher than p4) and sharp, having a prominent distal accessory cuspid and two (one mesial and one distal) cuspids at its cingulid. The fourth premolar is relatively shorter and blunter and it has the same three accessory cuspids as the third one, but all of them are more developed. The lower carnassial exhibits faint signs of wear in its carnassial blade. The protoconid is the largest cuspid, separated from the paraconid by a deep carnassiform notch. The paraconid is slightly mesiolingually bent. The metaconid is relatively small and it is only slightly connected to the

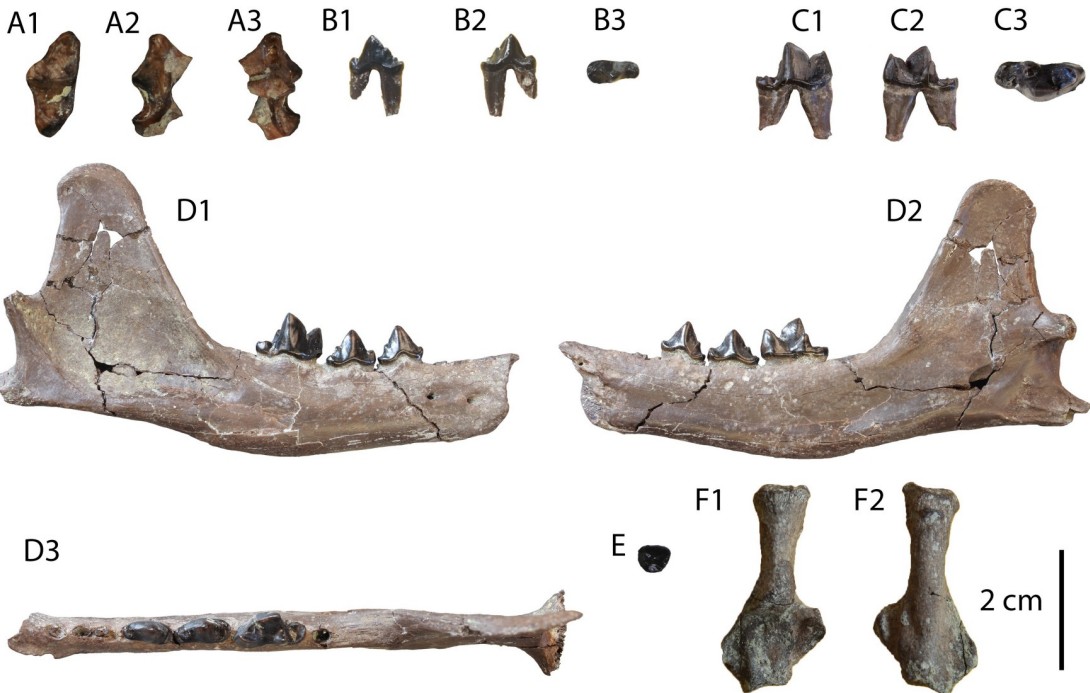

**Fig 24.** New material of *Semigenetta sansaniensis* from Hammerschmiede: (A) GPIT/MA/13452 right M1 in (A1) occlusal, (A2) mesial and (A3) distal view; (B) GPIT/MA/12130 left p4 in (B1) buccal, (B2) lingual and (B3) occlusal view; (C) SNSB-BSPG-2020 XCIV-5706 right m1 in buccal (C1), lingual (C2) and occlusal (C3) view; (D) SNSB-BSPG-2020 XCIV-4024 right hemimandible in occlusal (D1), buccal (D2) and lingual (D3) view; (E) SNSB-BSPG-1973-XIX-26, right m2 in occlusal view; (F) GPIT/MA/ 10298 right calcaneum in (F1) dorsal and (F2) ventral views. The size of the M1 (A) and m2 (E) has been doubled for the sake of visibility.

protoconid. It is placed distally to the protoconid. The talonid is short, with a shallow basin and a low rim. Only a small hypoconid is present at its buccal side. No distinct cingulids are present in all three remaining teeth.

[4] reported an m2 from HAM 1 that they attributed to *Proputorius sansaniensis* (SNSB-BSPG-1973-XIX-26; Fig 24E). As mentioned above, this specimen is now attributed to *Semigenetta sansaniensis*. It is unworn and relatively complete, missing only its root. Its outline is semi-circular, being convex in its lingual and straight in its buccal side. The protoconid is the largest cuspid, followed by the slightly smaller hypoconid. The lingual side of the tooth hosts two small cuspulids that can be identified as reduced metaconid and entoconid.

The specimen GPIT/MA/10298 is a complete right calcaneus (Fig 24F). Its general shape is long and slender. The groove for the tendo achillis is restricted to the posterior part of the bone. The posterior talar articular surface is oval-shaped and laterally folded through the shaft. The surface between the posterior talar articular surface and the distal epiphysis is more long than wide. The sustentaculum tali is wide and long, while the middle talar articular surface is oval. The groove for the tendon of flexor hallucis longus is deep and narrow. A peroneal tubercle also exists, creating space for the attachment of muscles. Finally, the articular surface for the cuboid is semicircular laterally, until the middle of the surface, followed by a shallow groove and an additional triangular surface.

**Comparison:** The morphological homogeneity of the genus *Semigenetta* in the fossil record leads to the differentiations being held mainly by metrical comparisons [21]. It is demonstrated in Tables 19 and 20 that *S. sansanienis* is a forme of intermediate size between the

**Table 19. Metrical comparison of the dental dimensions between the new specimens of *Semigenetta sansaniensis* from Hammerschmiede with the currently known measurements of the genus indicating the data source.** Data of *Semigenetta grandis* are mentioned in Table 20.

| Species | Code | p3L | p3W | p4L | p4W | m1L | m1W | M1L | M1W |
|---|---|---|---|---|---|---|---|---|---|
| *S. sansaniensis* | GPIT/MA/13452 | | | | | | | 4.6 | 8.6 |
| | SNSB-BSPG-2020-XCIV-4024 | 7.1 | 3.2 | 8.0 | 3.7 | 10.5 | 5.0 | | |
| | SNSB-BSPG-2020-XCIV-5706 | | | | | 11.5 | 5.1 | | |
| | SNSB-BSPG-2020-XCIV-3614 | | | | | | 6.0 | | |
| | GPIT/MA/17351 | | | | 3.9 | | | | |
| | GPIT/MA/12130 | | | 9.0 | 4.0 | | | | |
| *S. sansaniensis* [22, 52, 72, 74, 78, 192, 194–198] | | 6.3–8.4 7.3 (20) | 1.8–3.5 2.9 (19) | 7.0–9.0 8.0 (20) | 2.9–4.1 3.5 (19) | 8.5–11.5 9.9 (44) | 3.5–5.5 4.7 (44) | 4.6–6.4 5.7 (7) | 7.4–9.6 8.5 (7) |
| *S. elegans* [46, 119, 198–201] | | 5.7–6.4 6.0 (10) | 2.4 | 6.1–7.3 6.6 (11) | 2.9 | 7.7–8.8 8.2 (11) | 4.0 | 4.3–5.7 4.6 (6) | 6.3–7.8 6.9 (6) |
| *S. cadeoti* [119] | | | | | | 6.0 | | | |
| *S. laugnacensis* [195] | | 4.8–5.1 5.0 (2) | 1.8–1.9 1.9 (2) | 5.8–6.2 6.0 (5) | 2.0–2.5 2.3 (5) | 7.1–7.5 7.4 (5) | 3.2–3.6 3.4 (5) | 6.2 | 3.2 |

small-sized *S. elegans*, *S. cadeoti* and *S. laugnacensis* and the large-sized *S. grandis*. The herein described material fits to the values provided for *S. sansaniensis*. Though, it must be noted that the fragmentary m1 (SNSB-BSPG-2020 XCIV-3614) is slightly wider than the known specimens of this form. However, we consider this as an indicator of intraspecific variability (an aspect already discussed by [21]) and not as a character of taxonomic relevance.

The dimensions of the m2 (m2L = 2.9 mm; m2W = 2.5 mm) indicate that this tooth must belong to a small- to medium-sized feliform. The hyaenid genera *Protictitherium* Kretzoi, 1938 [202] and *Plioviverrops* Kretzoi, 1938 [202] are characterized by well-developed m2 cuspids, including a hypoconulid [103]. On the contrary, the m2 of the viverrids are more plesiomorphic. The dimensions of the known m2 of *S. sansaniensis* (2.9 x 1.9 mm from Vieux-Collonges [52]; 3.0 x 2.3 mm and 3.8 x 3.0 mm from La Grive-Saint-Alban [72]) fit very well to the size of the present specimen. This species is the only viverrid of its size during the late Middle and early Late Miocene [21], so we are inclined to tentatively attribute this molar to *S. sansaniensis*.

The described calcaneum exhibits typical feliform characteristics, such as the slenderness of the shaft (also described as mediolateral depression), the high position of the articular surface for the astragalus and the moderate development of the sustentaculum tali. However, based on the comparison of [194], the calcaneum of *Semigenetta* differs from that of the felids in the relatively lower position of the sustentaculum tali, the more developed medial tubercle and the more distodorsally oriented articular surface for the astragalus. All these characteristics seem to fit to this morphology and differentiate it from the calcaneus of *Leptofelis vallesiensis* (Salesa

**Table 20. Comparison of the m1 dimensions between the known specimens of *Semigenetta grandis* indicating the data source.**

| Code/Locality | m1L | m1W | m1W/m1L |
|---|---|---|---|
| SNSB-BSPG-2020-XCIV-4220 HAM 4 | 15.3 | 7.4 | 48% |
| GPIT/MA/12452[21] HAM 4 | 15.6 | 7.7 | 49% |
| Castell de Barberá [29] | 13.5 | 6.9 | 51% |
| Rudabánya [65] | 12.7–14.5 13.9 (3) | 5.7–6.3 6.1 (3) | 43%–45% 44% (3) |

et al., 2012) [203] described in [204]. Additionally the measurements of the Hammerschmiede calcaneum (total length = 32.0 mm; maximum width = 13.5 mm) are similar to those reported by [192] for the specimen from Steinheim (total length ≈ 35.0 mm; maximum width = 15.3 mm).

*Semigenetta grandis* Crusafont Pairó & Golpe Posse, 1981 [29]

**Holotype:** IPC-IPS 94790, left hemimandible with p2–m1 and the alveolus of m2.

**Type Locality:** Castell de Barberá (Spain).

**Referred New Specimens:** HAM 4: SNSB-BSPG-2020 XCIV-4220, right m1.

**Description:** The specimen SNSB-BSPG-2020 XCIV-4220 (Fig 25) is a right m1 with small wear facets in its carnassial blade. It is almost identical to GPIT/MA/12452 from HAM 5 reported by [21]. It is robust, with a strong cingulid, especially in its mesiobuccal and distobuccal sides. The trigonid is long, with the protoconid being the largest cuspid. It is clearly higher than the paraconid, from which it is separated by a deep notch. The metaconid is developed, pointy and slightly distally bent. The talonid consists of a shallow basin that is oriented in a dorsobuccal-ventrolingual plane. Only a cristid-like hypoconid is present in its lingual side. The only differences between SNSB-BSPG-2020 XCIV-4220 and GPIT/MA/12452 are the slightly lower trigonid cuspids (because of the wear) and the more wrinkled enamel surface of the former.

**Comparison:** The large size of this specimen is enough to differentiate it from other species of viverrids that have been found in the Miocene of Europe [21]. The species *S. grandis* has been described from Castell de Barberá (type locality; [29]) and Rudabánya [65]. Both localities are considered to comprise typical MN 9 faunas [205]. No significant intraspecific differences can be traced between the specimens from these localities and Hammerschmiede. The lower carnassials from Rudabánya are slightly narrower (Table 20), and the specimens from Hammerschmiede (that represent the oldest occurrence of the species) are slightly larger. One noteworthy difference is the relatively low height of the m1 metaconid in the specimens from Rudabánya [65]. This cuspid is moderately developed in the specimen of Castell de Barberá and relatively high in the specimens from Hammerschmiede.

*Viverrictis* de Beaumont, 1973 [206]

**Type species:** *Viverrictis modica* Gaillard, 1899 [63]

**Other included species:** *Viverrictis vetusta* de Beaumont, 1973 [206].

**Remarks:** This small-sized viverrid genus includes only two species: *Viverrictis modica* from La Grive-Saint-Alban and *Viverrictis vetusta* from Vieux-Collonges and Sansan [206]

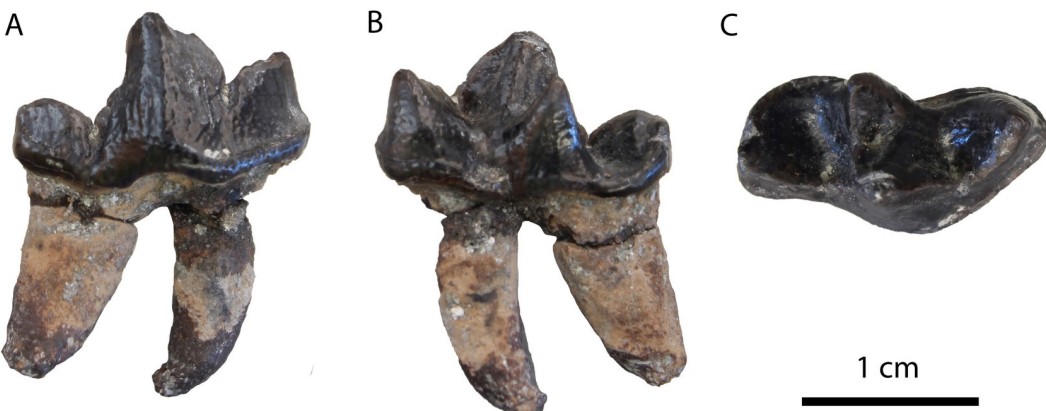

**Fig 25.** The new lower carnassial of *Semigenetta grandis* (SNSB-BSPG-2020 XCIV-4220) from HAM 4 in buccal (A), lingual (B) and occlusal (C) view.

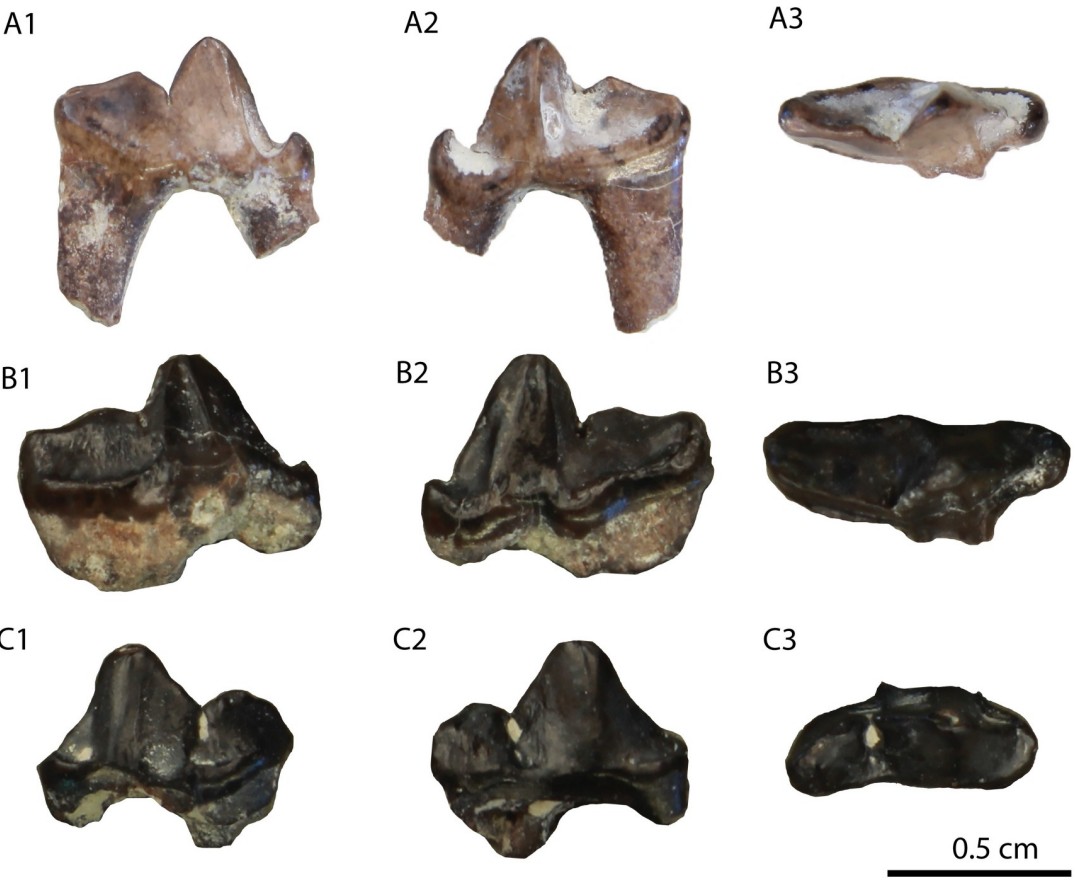

**Fig 26.** The P4 of *Viverrictis modica* from Hammerschmiede: (A) SNSB-BSPG-2020 XCIV-3995, right P4; (B) GPIT/MA/16659, right P4; (C) GPIT/MA/12649, left P4 in (1) lingual, (2) buccal and (3) occlusal view.

*Viverrictis modica* (Gaillard, 1899) [63]

**Holotype:** MHNL-LGr 13710, right hemimandible with p4–m1.

**Type Locality:** La Grive-Saint-Alban (France).

**Referred Specimens:** HAM 4: GPIT/MA/16659, right P4; SNSB-BSPG-2020 XCIV-3995, right P4; GPIT/MA/12649, left P4.

**Description:** The upper carnassials share the same morphology (Fig 26). They are almost complete, missing only their protocone, and they have no signs of wear. The teeth are in general considerably narrow, with no signs of a cingulum. The paracone is the highest cusp. It is relatively blunt, having a crest in its buccal side and it is separated from the oblique metastyle by a deep notch. The parastyle is considerably developed and slightly hook-like, being bent distally. The protocone is missing, but its base indicates that it stemmed from the paracone plane and was oriented slightly mesially.

**Comparison:** This form exhibits a typical plesiomorphic viverrid morphology, having a slender upper carnassial with a slightly mesially oriented protocone and a distinct notch between the paracone and the metastyle. However, the very small size and the considerably developed parastyle are enough to differentiate it from the genus *Semigenetta* and attribute it to the genus *Viverrictis*.

This genus includes two species: *Viverrictis vetusta* from Vieux-Collonges (MN 5) and Sansan (MN 6), and *Viverrictis modica* from La Grive-Saint-Alban (MN 7/8) [206]. These two

**Table 21. Metrical comparison of the P4L of SNSB-BSPG-2020 XCIV-3395 with the known datasets for *Viverrictis modica* and *Viverrictis vetusta* indicating the data source.**

| Species | Code | P4L |
|---------|------|-----|
| *V. modica* | SNSB-BSPG-2020-XCIV-3995 | 7.1 |
| | GPIT/MA/12649 | 6.8 |
| | GPIT/MA/16659 | 8.1 |
| | LGr 1367[63] | 7.0 |
| *V. vetusta* | Summed [52] | 6.6–7.2 7.0 (6) |

forms were differentiated by [206] based on several dental traits, despite having similar size (Table 21). One of these traits is the reduced/absent parastyle in *V. vetusta*, whereas in *V. modica* it is present [206]. Based on the considerable size of the parastyle in the Hammerschmiede specimens, it is suggested that this material belongs to *Viverrictis modica*. This report consists of the youngest record of this viverrid genus.

## Discussion

### Taxonomic diversity and biochronology

The small carnivoran fauna (sensu [37]) of Hammerschmiede includes 20 species belonging to 9 subfamilies: "*Martes*" *sansaniensis*, "*Martes*" cf. *munki*, "*Martes*" sp., *Circamustela hartmanni* n. sp., *Laphyctis mustelinus*, Guloninae indet., *Eomellivora moralesi*, *Vishnuonyx neptuni*, *Paralutra jaegeri*, *Lartetictis* cf. *dubia*, *Trocharion albanense*, *Palaeomeles pachecoi*, *Proputorius sansaniensis*, *Proputorius pusillus*, *Alopecocyon goeriachensis*, Simocyoninae indet., *Potamotherium* sp., *Semigenetta sansaniensis*, *Semigenetta grandis* and *Viverrictis modica*. The presence of so many species of small-sized carnivorans in a single locality is extraordinary. In particular, the channel HAM 5 includes 11 sympatric small carnivorans, while HAM 4 includes 13 sympatric forms (Table 22). Such a high diversity is resembling the extant tropical African diversities [207]. If the combined record of the three layers is taken into account (which is herein considered reasonable based on the restricted age difference between them) the fauna of Hammerschmiede includes comparable number of small carnivoran forms to some of the taxonomically richest Miocene localities, such as Sansan [72], La Grive-Saint-Alban fissures M+L7 and L3+L5 [27, 138], Rudabánya [65] and Can Llobateres 1 [87, 159, 208] (Table 23).

Many of the species are documented by very low numbers of individuals (Table 22) or even only by a single tooth. We therefore expect that the taxonomic diversity is not saturated and will even rise in the future excavation efforts. For example, the absence of the relatively common herpestid *Leptoplesictis* Forsyth Major, 1903 [137] (sensu lato; for an alternative point of view see [210]) could be covered in the future.

From a biochronological standpoint, the Hammerschmiede fauna shows several First Occurrence Dates (FOD) and Last Occurrence Dates (LOD) for small carnivorans (Fig 27). We report here the FODs of the genera *Circamustela*, *Eomellivora* and the species *Semigenetta grandis*. The documented LODs include the genera *Laphyctis*, *Lartetictis*, *Alopecocyon*, *Potamotherium* and *Viverrictis*. The dominance of LOD over FOD may reflect the biochronological position of Hammerschmiede at the end of the Astaracian mammal age, 0.3–0.5 million years before the onset of the Vallesian mammal age at 11.1 Ma [1].

### Ecomorphology

Some of these species have already been studied in terms of ecomorphology, including body mass, locomotor patterns and dietary habits. In particular, [209] have discussed the attribution

**Table 22. Distribution of the discussed taxa in the Hammerschmiede layers, together with their estimated body mass (BM; in kg), locomotor lifestyle (LL; GT = Generalized Terrestrial; SA = Semi-Aquatic; Sc = Scansorial; SF = Semi-Fossorial) and dietary habits (DH; I = Insectivorous; hC = Hypocarnivorous; C = Carnivorous; HC = Hypercarnivorous).** Attribution to these categories is based on [209] and this study. The numbers per layer indicate the minimum number of individuals.

| Species | HAM 1 | HAM 5 | HAM 4 | BM | LL | DH |
|---|---|---|---|---|---|---|
| "*Martes*" *sansaniensis* | | 1 | 3 | 3–10 | Sc | C |
| "*Martes*" cf. *munki* | | 2 | 1 | 1–3 | ? | C |
| "*Martes*" sp. | 1 | | | <1 | ? | C |
| *Circamustela hartmanni* n. sp. | 1 | 1 | 1 | 1–3 | Sc? | HC |
| *Laphyctis mustelinus* | | | 1 | 10–30 | GT | HC |
| Guloninae indet. | | 1 | | 3–10 | ? | hC |
| *Eomellivora moralesi* | | 1 | | 10–30 | GT | HC |
| *Vishnuonyx neptuni* | | | 3 | 10–30 | SA | C |
| *Paralutra jaegeri* | | 1 | 2 | 3–10 | SA | C |
| *Lartetictis* cf. *dubia* | | | 2 | 3–10 | SA | C |
| *Trocharion albanense* | | 4 | 1 | 1–3 | SF | C |
| *Palaeomeles pachecoi* | | 2 | | 3–10 | GT/SF | hC |
| *Proputorius sansaniensis* | 1 | | | 1–3 | Sc | C |
| *Proputorius pusillus* | 2 | | | <1 | ? | C |
| *Alopecocyon goeriachensis* | | 1 | | 3–10 | Sc | C |
| Simocyoninae indet. | | | 1 | 10–30 | ? | C |
| *Potamotherium* sp. | | 1 | 1 | 10–30 | SA | C |
| *Semigenetta sansaniensis* | 1 | 1 | 7 | 3–10 | Sc | C |
| *Semigenetta grandis* | | | 2 | 10–30 | GT | HC |
| *Viverrictis modica* | | | 1 | <1 | Sc | I |
| Number of species | 5 | 11 | 13 | - | - | - |
| Number of individuals | 6 | 16 | 26 | - | - | - |

to these categories of the species "*Martes*" *sansaniensis*, "*Martes*" cf. *munki*, *Laphyctis mustelinus* (as "*Ischyrictis mustelinus*"), *Paralutra jaegeri*, *Lartetictis dubia*, *Trocharion albanense*, *Proputorius sansaniensis*, *Proputorius pusillus*, *Alopecocyon goeriachensis* (as "*Alopecocyon leptorynchus*"), *Semigenetta sansaniensis* and *Viverrictis modica*. For these species we follow the attributions of [209]. The only exceptions concern the body size of *Laphyctis mustelinus* and *Lartetictis dubia*. The former species is relatively large (having an m1L approximately 15 mm; [90]). Therefore, based on the equation of [211] for the m1L of mustelids, this species is herein attributed to the 10–30 kg group. On the contrary, the latter species was attributed to the 10–30 kg group by [209], but given the small size of the Hammerschmiede specimens, the values (based on the equation of [211]) point towards the 3–10 kg category. The rest of the species are discussed in more detail here.

The very small "*Martes*" sp. has been attributed to the <1 kg body mass category, based on its very small dimensions. Additionally, given the preservation of the generalistic marten-like morphology in its upper carnassial, it was considered that this species is a carnivore, in agreement to the other members of the genus.

The genus *Circamustela* has been widely considered to be a hypercarnivorous form, judging from the reduction of the molar's grinding areas, the crest-like molar cusps, the high m1 protoconid and the low m1 hypoconid and entocristid [62]. Based on the equation of [211] for mustelids, when the m1L of *Circamustela hartmanni* n. sp. is considered, its estimated body mass is approximately 1 kg. No postcranial elements of *Circamustela* have been published up to now. Therefore, a secure attribution to a locomotor category is not possible. However, the

**Table 23. Distribution of the small carnivorans (sensu [37]) among the localities of Sansan [72], La Grive-Saint-Alban fissures M+L7 and L3+L5 [27, 138], Hammerschmiede, Rudabánya [65] and Can Llobateres 1 [87, 159, 208].**

| | | Sansan | La Grive (M, L7) | La Grive (L3, L5) | Hammerschmiede | Rudabánya | Can Llobateres 1 |
|---|---|---|---|---|---|---|---|
| | | MN 6 | MN 7 | MN 8 | MN 8 | MN 9 | MN 9 |
| Guloninae | "Martes" sansaniensis | x | | | x | | |
| | "Martes" filholi | | x | x | | cf. | |
| | "Martes" munki | | x | x | cf. | | x |
| | "Martes" delphinensis | | x | x | | | |
| | "Martes" melibulla | | | | | | x |
| | "Martes" sp. | | | | x | | |
| | Circamustela dechaseauxi | | | | | | x |
| | Circamustela hartmanni nov. sp. | | | | x | | |
| | Laphyctis mustelinus | | x | x | x | | |
| | Ischyrictis zibethoides | x | | | | | |
| | Plesiogulo sp. | | | | | | x |
| | Trochictis narcisoi | | | | | | x |
| | Trochictis sp. | | | | | cf. | |
| | Guloninae indet. | | | | x | | |
| Mellivorinae | Eomellivora moralesi | | | | x | | |
| | Eomellivora fricki | | | | | | x |
| Lutrinae | Paralutra jaegeri | | | | x | x | x |
| | cf. Paralutra sp. | | | | | x | |
| | Vishnuonyx neptuni | | | | x | | |
| | Lartetictis dubia | x | | | cf. | | |
| Melinae | Sabadellictis crusafonti | | | | | | x |
| | Taxodon sansaniensis | x | | | | cf. | cf. |
| | Melinae indet. | | | | | x | |
| | Trochictis depereti | | x | | | | |
| Leptarctinae | Trocharion albanense | | x | x | x | | x |
| | Trochotherium cyamoides | | x | | | | |
| | Gaillardina transitoria | | x | | | | |
| Mephitinae | Proputorius sansaniensis | x | | | x | | |
| | Proputorius pusillus | | x | | x | | |
| | Proputorius sp. | | | | | cf. | |
| | Grivamephitis pusilla | | x | | | | |
| | Grivamephitis meini | | | x | | | |
| | Mesomephitis medius | | | | | | x |
| | Promephitis pristinidens | | | | | | x |
| | Palaeomeles pachecoi | | | | x | | |
| Simocyoninae | Alopecocyon goeriachensis | x | x | x | x | | |
| | Simocyoninae indet. | | | | x | | |
| | Simocyon diaphorus | | | | | x | |
| | Protursus simpsoni | | | | | x | x |
| Potamotheriinae | Potamotherium sp. | | | | x | | |
| Viverridae | Semigenetta sansaniensis | x | x | x | x | | x |
| | Semigenetta grandis | | | | x | x | |
| | Viverrictis modica | | x | x | x | | |
| | Viverrictis vetusta | x | | | | | |

(*Continued*)

**Table 23.** (Continued)

| | | Sansan | La Grive (M, L7) | La Grive (L3, L5) | Hammerschmiede | Rudabánya | Can Llobateres 1 |
|---|---|---|---|---|---|---|---|
| | | MN 6 | MN 7 | MN 8 | MN 8 | MN 9 | MN 9 |
| Herpestidae | *Leptoplesictis atavus* | x | | | | | |
| | *Leptoplesictis filholi/aurelianensis* | | x | x | | | |
| | *Jourdanictis grivensis* | | x | | | | |
| Number of Species | | 9 | 15 | 10 | 20 | 10 | 14 |

most parsimonious approach would be to suggest that it would be scansorial, as all the relatively small-sized gulonines and mustelines.

The indetermined gulonine is suggested to have a body mass of approximately 5 kg based on its m1L and the equation of [211] for the mustelids. The general morphology of this species points towards a hypocarnivorous dietary category based on the low and blunt premolars with

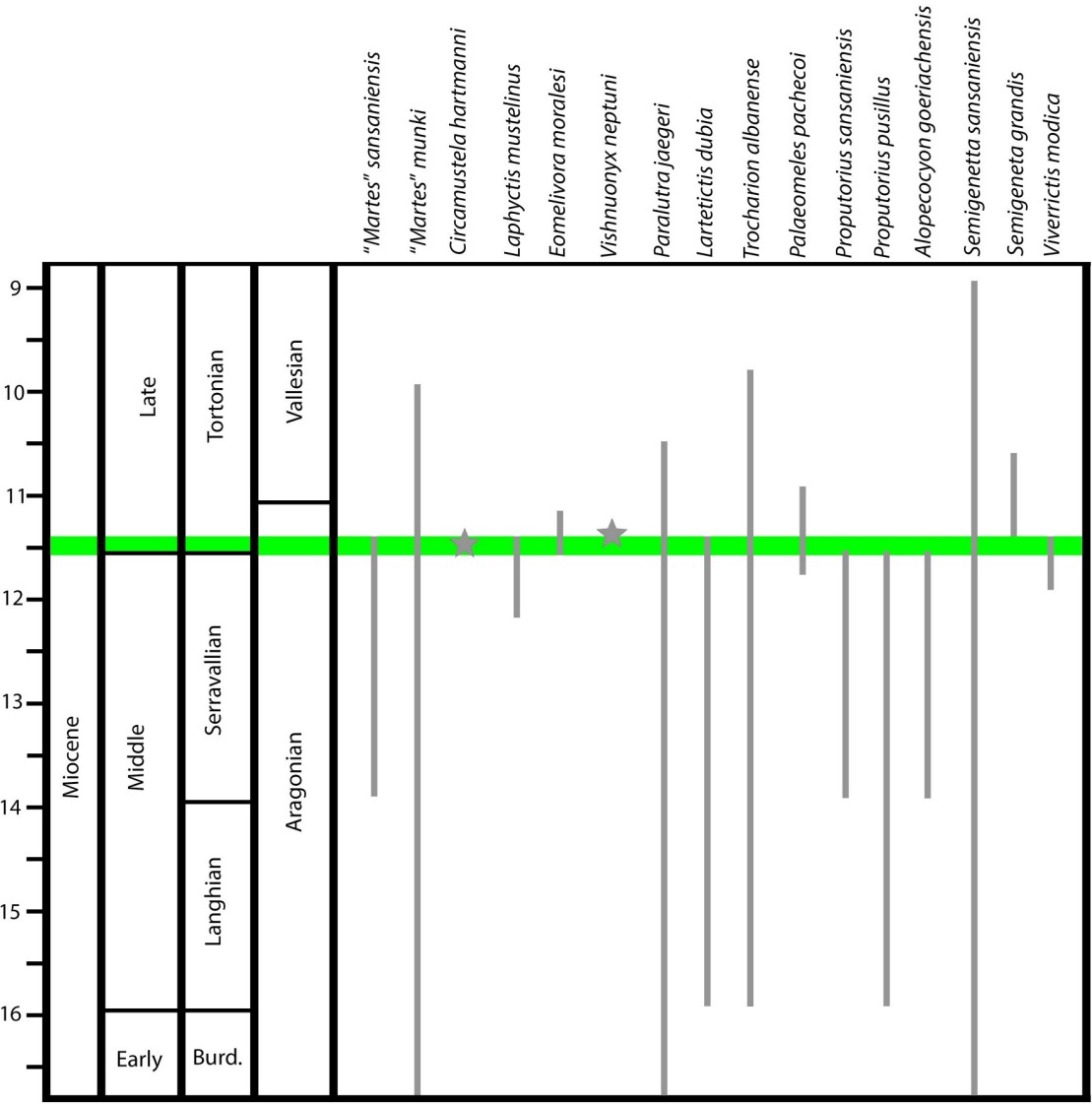

**Fig 27. Stratigraphical range of the discovered carnivorans in species level.**

no developed accessory cuspids, the moderately long m1 talonid and the low m1 trigonid cuspids [212–216]. Since no postcranial can be attributed to this mandible and the exact relationships of this form are not known, then its locomotor pattern remains unclear.

The specimen GPIT/MA/09875 is the first published complete m1 *Eomellivora moralesi*. The equation of [211] for the mustelids is based mostly on small-sized animals. Therefore, it is reasonable to suggest that its prediction accuracy can be doubted in larger body masses. The m1L of *Eomellivora moralesi* is 24.7 mm (Table 9). This value is comparable and slightly larger than that for the extant wolverine, *Gulo gulo* (Linnaeus, 1758) [44] (range 19.5–22.5, average 20.8, n = 20; [217]). Therefore, it can be deduced that *Eomellivora* from Hammerschmiede had a similar or slightly larger body mass than the extant wolverine, thus 10–30 kg [218]. [100] discussed the dietary habits of *Eomellivora*, concluding that it probably was a hypercarnivorous scavenger-opportunist. [100] and [219] also provided some preliminary comments on postcranium of *Eomellivora piveteaui*, stating that they were relatively long, possibly adapted to cursoriality. However, until a more detailed study is published, we prefer to attribute *Eomellivora moralesi* to the Generalized Teresstrial group.

[22] discussed the ecomorphological traits of *Vishnuonyx neptuni*, concluding that it was a large, semiaquatic fish-eater, comparable to *Pteronura brasiliensis* (Zimmermann, 1780) [220].

*Palaeomeles pachecoi* is characterized by several traits that indicate hypocarnivorous adaptations, such as the wide lingual platform in M1, the low cusps of M1, the low premolars without accessory cuspids, the very long m1 talonid with small cuspules, the low m1 trigonid cuspids and the relatively large m2 [212–216]. These adaptations resemble that of the extant Eurasian badger, *Meles meles* (Linnaeus, 1758) [44]. However, the size of *Palaeomeles* (m1L ≈ 14.0 mm) is slightly smaller than that of *Meles* (range 15.4–18.5, average 16.8, n = 26; [221]). Therefore, an attribution to the 3–10 kg group seems reasonable. The postcranial anatomy of *Palaeomeles* was studied by [145]. These authors concluded that *Palaeomeles* exhibits some adaptations towards fossoriality (similar to the extant badgers), but the evolutionary stage of these characteristics was not as evident as in the extant forms. Therefore, we prefer to classify this species as Generalized Teresstrial / Semi-fossorial.

The size of Simocyoninae indet. is intermediate between that of *Alopecocyon* (3–10 kg; [209]) and *Simocyon batalleri* (≈50 kg; [222]). Therefore, the attribution to the group of 10–30 kg seems possible. Since no postcranial of *Protursus* are known and its relationships with the more evolved *Simocyon* are not clear, we prefer to retain its locomotor patterns as unknown. As [156] pointed out, the subfamily Simocyoninae is characterized by more hypercarnivorous adaptations than that of Ailurinae. These traits are less developed in the early forms, but they become more evident in *Simocyon*. However, the m2 is well-developed, with long talonid and low cuspids. Therefore, a generalistic/opportunistic diet can be suggested for this form.

The semi-aquatic lifestyle of *Potamotherium* has been discussed in detail in many studies [38, 125, 178]. In general, this genus exhibits generalized, lutrine-like adaptations that are connected with piscivory, such as the high premolars and the high m1 protoconid and metaconid; but also for durophagy, such as the broad premolars (with distinct cingula), the wide P4 protocone área and the developed m1 talonid [21, 125, 212–216]. Therefore, a mixed diet of fish and bivalves is here proposed for this form. The body mass of the Hammerschmiede *Potamotherium* is not possible to be calculated from a damaged M1. However, it has been demonstrated that it is slightly larger than *Potamotherium miocenicum*. Based on the dimensions of this species [183] the body mass of the Hammerschmiede form can be deduced to be similar to that of *Lartetictis dubia*, falling into the 10–30 kg group.

Finally, the ecomorphology of *Semigenetta grandis* was discussed in [21], concluding that it is a species slightly larger than 10 kg with hypercarnivorous adaptations and possibly terrestrial locomotor habits.

A summary of these attributions as well as the distributions of the discussed taxa in the Hammerschmiede layers are provided in Table 22.

The herein described forms can be separated into three size-groups. The large-sized (10–30 kg) *Laphyctis*, *Eomellivora*, *Vishnuonyx*, Simocyoninae indet., *Potamotherium* and *Semigenetta grandis*; the medium-sized (3–10 kg) "*Martes*" *sansaniensis*, Guloninae indet., *Paralutra*, *Lartetictis*, *Palaeomeles*, *Alopecocyon* and *Semigenetta sansaniensis* and the small-sized (<3 kg) "*Martes*" cf. *munki*, "*Martes*" sp., *Circamustela*, *Proputorius*, *Trocharion* and *Viverrictis*.

The fluvial nature of the HAM 4 and HAM 5 (and most possibly HAM 1) layers has a significant impact on the carnivoran guild of the locality as several of the taxa are characterized by semi-aquatic lifestyle. In particular, the genera *Lartetictis*, *Vishnuonyx*, *Paralutra* and *Potamotherium* have been considered as semi-aquatic carnivorans feeding mainly on fish and bivalves [22, 34, 134, 209]. All the other forms are considered to have terrestrial or, possibly, semi-arboreal lifestyle. The mutual exclusion of *Lartetictis* and *Paralutra*, pointed out by [132, 134] is not evident in Hammerschmiede, as both genera have been found in the HAM 4 layer. Both studies suggested that *Paralutra* gradually replaced *Lartetictis*. However, their coexistence in Hammerschmiede (together with their significantly overlapping stratigraphical ranges; [132]) indicate that these species were able to live together in the same ecosystem, if the available resources were sufficient. The rest of the species are mainly scansorial/terrestrial. The high frequency of possibly scansorial and arboreal species can be associated to a more closed environment.

The larger terrestrial forms (*Eomellivora*, *Laphyctis* and *Semigenetta grandis*) are relatively rare. However, a partitioning can be noted, as *Eomellivora* is found only in HAM 5, whereas *Laphyctis* is found only in HAM 4. Though, this partitioning might have been caused by sample bias. *Laphyctis* is known of being able to coexist with other similar-sized carnivorans, e. g. it is known to coexist with *Ischyrictis zibethoides* in Vieux Collonges [52] or with *E. moralesi* in Can Mata [30, 58, 91, 98]. On the other hand, *S. grandis* is found only in HAM 4.

Several forms of the small- or medium-sized groups correspond to the niche of the extant martens and weasels: "*Martes*" *sansaniensis*, "*Martes*" cf. *munki*, *Circamustela*, *Semigenetta sansaniensis* and *Viverrictis*. It can be noted that the three smaller forms ("*Martes*" cf. *munki*, *Circamustela* and *Viverrictis*) are characterized by slenderer and pointier cheek teeth. This trait has been considered as an indication of hypercarnivorous diet for *Circamustela* [62]. Therefore, it is possible that all three forms were adapted to a more flesh-based diet, whereas the larger "*Martes*" *sansaniensis* and *Semigenetta sansaniensis* had a more mixed and opportunistic diet. Simocyoninae indet. also exhibits relatively narrow m2, in agreement with the general trend of simocyonines towards an opportunistic-hypercarnivorous ecomorphology ([156]. However, given the absence of data for this form, it is not easy to deduce its dietary or locomotor habits. *Trocharion*, *Palaeomeles*, Guloninae indet., *Proputorius* and *Alopecocyon* exhibit relatively developed grinding areas (including wide upper molars and long m1 talonid) and simple, blunt premolars that point towards a more hypocarnivorous diet, based more on invertebrates and plant material.

Overall, the small-carnivoran datum of Hammerschmiede points towards a relatively closed environment that was dominated by its fluvial influence. The presence of so many carnivorans of relatively similar ecological roles in a singly locality indicates that Hammerschmiede was considerably rich in terms of niche opportunities and and prey frequencies.

## Conclusions

The small carnivoran fauna of Hammerschmiede includes 20 distinct species belonging to nine different subfamilies and three size-groups, representing one of the highest taxonomic

diversities reported for the Miocene of Europe. A new species of *Circamustela* is described, representing the FOD of that genus. Furthermore, the late Astaracian Hammerschmiede fauna provides the FOD for *Eomellivora* and *Semigenetta grandis*, as well as the LOD of *Laphyctis*, *Lartetictis*, *Alopecocyon*, *Potamotherium* and *Viverrictis*. Ecomorphological comparison between the discovered forms reveals a well-established niche partitioning for all forms. The coexistence of possible competitors (*Paralutra-Lartetictis*; *Viverrictis-Circamustela*; *Semigenetta sansaniensis*-"*Martes*" *sansaniensis*) can be explained by the existence of sufficient resources in the Hammerschmiede ecosystem.

## Acknowledgments

The authors want to thank Dr I. Werneburg (GPIT), Dr E. Amson (SMNS) and Dr G. Rössner (SNSB-BSPG) for providing access to the material under their curation. Additionally, we would like to thank Dr J. Robles (ICP), Dr M. Rummel (NMA) and Dr B. Kear (Museum of Evolution, Uppsala) for providing the inventory numbers and photographs of material under their curation. We acknowledge the support of the Centre of Visualisation, Digitisation and Replication at the Eberhard Karls Universität in Tübingen for instrument use, scientific and technical assistance and Dr. G. Ferreira (University of Tübingen) for µCT-scanning of selected specimens, for the detailed observation of selected specimens. The authors would like to thank Dr. M. Morlo and one anonymous reviewer, as well as the editorial board of PLoS ONE for their fruitful comments. We furthermore are grateful to numerous volunteers and participants for their help during the excavations at Hammerschmiede.

## Author Contributions

**Conceptualization:** Nikolaos Kargopoulos, Alberto Valenciano, Madelaine Böhme.

**Investigation:** Nikolaos Kargopoulos, Alberto Valenciano, Juan Abella, Panagiotis Kampouridis, Thomas Lechner, Madelaine Böhme.

**Methodology:** Nikolaos Kargopoulos, Alberto Valenciano, Juan Abella, Panagiotis Kampouridis, Madelaine Böhme.

**Resources:** Madelaine Böhme.

**Supervision:** Nikolaos Kargopoulos, Madelaine Böhme.

**Validation:** Alberto Valenciano.

**Visualization:** Nikolaos Kargopoulos.

**Writing – original draft:** Nikolaos Kargopoulos, Alberto Valenciano, Juan Abella, Panagiotis Kampouridis, Thomas Lechner, Madelaine Böhme.

**Writing – review & editing:** Nikolaos Kargopoulos, Alberto Valenciano, Juan Abella, Panagiotis Kampouridis, Thomas Lechner, Madelaine Böhme.

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
