## [Decision Letter · Decision Letter 0]

25 Mar 2022

PONE-D-22-05374The exceptionally high diversity of small carnivorans from the Late Miocene hominid locality of Hammerschmiede (Bavaria, Germany)PLOS ONE

Dear Dr. Kargopoulos,

Thank you for submitting your manuscript to PLOS ONE. After careful consideration, we feel that it has merit but does not fully meet PLOS ONE’s publication criteria as it currently stands. Therefore, we invite you to submit a revised version of the manuscript that addresses the points raised during the review process.

We look forward to receiving your revised manuscript.

Kind regards,

Giorgio Carnevale, Ph.D

Academic Editor

PLOS ONE

Journal Requirements:

3. In your manuscript, please provide additional information regarding the specimens used in your study. Ensure that you have reported specimen numbers and complete repository information, including museum name and geographic location. 

For more information on PLOS ONE's requirements for paleontology and archaeology research, see https://journals.plos.org/plosone/s/submission-guidelines#loc-paleontology-and-archaeology-research.

"There is no conflict of interest."

6. Please take this opportunity to be sure you have met all of our guidelines for new species. For proper registration of a new zoological taxon, we require two specific statements to be included in your manuscript.

a.
In the Results section, the globally unique identifier (GUID), currently in the form of a Life Science Identifier (LSID), should be listed under the new species name, for example: Anochetus boltoni Fisher sp. nov. urn:lsid:zoobank.org:act:B6C072CF-1CA6-40C7-8396-534E91EF7FBBAnother LSID for the manuscript itself should also appear within the Nomenclature statement. You will need to contact Zoobank (zoobank.org/About) to obtain a GUID (LSID). You should receive one LSID for your manuscript and a separate, unique LSID for the new species. b.
Please also insert the following text into the Methods section, in a sub-section to be called "Nomenclatural Acts": The electronic edition of this article conforms to the requirements of the amended International Code of Zoological Nomenclature, and hence the new names contained herein are available under that Code from the electronic edition of this article. This published work and the nomenclatural acts it contains have been registered in ZooBank, the online registration system for the ICZN. The ZooBank LSIDs (Life Science Identifiers) can be resolved and the associated information viewed through any standard web browser by appending the LSID to the prefix "http://zoobank.org/". The LSID for this publication is: urn:lsid:zoobank.org:pub: XXXXXXX. The electronic edition of this work was published in a journal with an ISSN, and has been archived and is available from the following digital repositories: PubMed Central, LOCKSS [author to insert any additional repositories]. All PLOS ONE articles are deposited in PubMed Central and LOCKSS. If your institute, or those of your co-authors, has its own repository, we recommend that you also deposit the published online article there and include the name in your article.Following a recent ruling by the International Commission on Zoological Nomenclature, electronic journals are now a valid format for publication of new zoological taxa. In order to ensure the valid publication of your new species, please be sure to include the updated version of Nomenclatural Acts (above). A complete explanation of our guidelines for publishing new species can be found on our website: http://www.plosone.org/static/guidelines#zoological.

Reviewers' comments:

Reviewer's Responses to Questions

**Comments to the Author**

1. Is the manuscript technically sound, and do the data support the conclusions?

Reviewer #1: Yes

Reviewer #2: Yes

2. Has the statistical analysis been performed appropriately and rigorously? 

Reviewer #1: N/A

Reviewer #2: N/A

3. Have the authors made all data underlying the findings in their manuscript fully available?

Reviewer #1: Yes

Reviewer #2: Yes

4. Is the manuscript presented in an intelligible fashion and written in standard English?

Reviewer #1: Yes

Reviewer #2: Yes

5. Review Comments to the Author

Reviewer #1: This is a highly welcomed and thoroughfully presented contribution to the evolution of carnivorans in the Miocene of Europe. I have only few comments to be found in the attached pdf.

My most important suggestion is to include a locality map and a stratigraphic chart, showing the age of the different horizons.

Reviewer #2: The paper "The exceptionally high diversity of small carnivorans from the Late Miocene hominid locality of Hammerschmiede (Bavaria, Germany)" is a really valuable contribution enriching our knowledge on the too-often-neglected portion of the Carnivora diversity, i.e., small-sized taxa, and doing so in a crucial time of differentiation and radiation of modern clades like Late Miocene. In this sense the fauna from Hammerschmiede elevates not only at a regional level, but potentially to the continental/intercontinental one for its abundant and diverse assemblage.

This said, I really enjoyed reading the manuscript, the descriptions, and the arguments supporting each instance made by the author. I find they made a valuable, thoroughly thought, and neat piece of work. I agree with the author in the erection of the new species of Circamustela, although I would like the author to consider one thing: if the author wish to refer the species to “the Hartmann family” -line 403- I think the correct latin case to be used, if I am not mistaken, should not be singular genitive but rather a plural one, as they refer to a group of people and not to the single owner who granted the permission in the first place. This means, is we latinize the surname “Hartmann” as a second declension name, to change “hartmanni” in “hartmannorum”. Food for thoughts.

I am pleased to see a great number of tables with measures of the described species and of comparison material of the different genera. What is odd is the absolute absence of morphometric analyses throughout the text (with the single exception of Fig. 18). I think this is the major flaw of the work: it seems very few statistics can be done on single specimens, yet I would have expected some analyses of the beautiful cranium of “Martes” sansaniensis (if not to other fossil species at least in comparing to extant Martes like a PCA on cranial and associated dental variable, just a suggestion), or to better describe the variability or rather, the position, of Circamustela hartmanni/orum in respect to other species of the genus. This observation could actually be made for anyone of the taxa described in the manuscript, which would probably discourage the authors in considering this suggestion, but I deem some analyses should indeed be done (maybe grouping family members in single scatterplot graph? I believe a solution can be found).

Considering the biochronological relevance of the species recovered in Hammerschmiede I would add a figure showing the chronology of the discussed species (as well as other relevant taxa of the same genera) to visually show the relevance of the finding here reported.

The “Ecomorphology”section of the discussion is a bit speculative, which is not a problem per se but it could be improved reducing the interpretations and planning a quantitative comparison between other Miocene small mammal associations of Europe.

Lastly, although I am not a Native English Speaker, I nevertheless found the English level to be appropriate and well-structured, and effective, I would say, in conveying the message.

In conclusion I deem the manuscript is almost ready for publication (this is why i would say minor revision needed) and, if the author care to take into consideration some of my suggestions, I think it will make a really valuable contribution to Miocene carnivorans in Europe.

6. PLOS authors have the option to publish the peer review history of their article (what does this mean?). If published, this will include your full peer review and any attached files.

Reviewer #1: **Yes: **Michael Morlo

Reviewer #2: No

---

## [Author Response · Author response to Decision Letter 0]

14 Apr 2022

We are thankful to the reviewers and the editorial board for their fruitful comments. A detailed list with the changes we have made has been attached to the revised version.

---

## [Editor Report · Decision Letter 1]

12 May 2022

The exceptionally high diversity of small carnivorans from the Late Miocene hominid locality of Hammerschmiede (Bavaria, Germany)

PONE-D-22-05374R1

Dear Dr. Kargopoulos,

We’re pleased to inform you that your manuscript has been judged scientifically suitable for publication and will be formally accepted for publication once it meets all outstanding technical requirements.

Kind regards,

Giorgio Carnevale, Ph.D

Academic Editor

PLOS ONE

---

## [Editor Report · Acceptance letter]

30 Jun 2022

PONE-D-22-05374R1 

The exceptionally high diversity of small carnivorans from the Late Miocene hominid locality of Hammerschmiede (Bavaria, Germany) 

Dear Dr. Kargopoulos:

I'm pleased to inform you that your manuscript has been deemed suitable for publication in PLOS ONE. Congratulations! Your manuscript is now with our production department. 

Kind regards, 

on behalf of

Dr. Giorgio Carnevale 

Academic Editor

PLOS ONE